# Towards Understanding the Mixture-of-Experts Layer in Deep Learning

**Zixiang Chen**
Department of Computer Science
University of California, Los Angeles
Los Angeles, CA 90095, USA
chenzx19@cs.ucla.edu

**Yihe Deng**
Department of Computer Science
University of California, Los Angeles
Los Angeles, CA 90095, USA
yihedeng@cs.ucla.edu

**Yue Wu**
Department of Computer Science
University of California, Los Angeles
Los Angeles, CA 90095, USA
ywu@cs.ucla.edu

**Quanquan Gu**
Department of Computer Science
University of California, Los Angeles
Los Angeles, CA 90095, USA
qgu@cs.ucla.edu

**Yuanzhi Li**
Machine Learning Department
Carnegie Mellon University
Pittsburgh, PA 15213, USA
yuanzhil@andrew.cmu.edu

## Abstract

The Mixture-of-Experts (MoE) layer, a sparsely-activated model controlled by a router, has achieved great success in deep learning. However, the understanding of such architecture remains elusive. In this paper, we formally study how the MoE layer improves the performance of neural network learning and why the mixture model will not collapse into a single model. Our empirical results suggest that the cluster structure of the underlying problem and the non-linearity of the expert are pivotal to the success of MoE. This motivates us to consider a challenging classification problem with intrinsic cluster structures. Theoretically, we proved that this problem is hard to solve by a single expert such as a two-layer convolutional neural network (CNN). Yet with the MoE layer with each expert being a two-layer CNN, the problem can be solved successfully. In particular, our theory shows that the router can learn the cluster-center features, which helps divide the input complex problem into simpler classification sub-problems that individual experts can conquer. To our knowledge, this is the first theoretical result toward formally understanding the mechanism of the MoE layer for deep learning.

## 1 Introduction

The Mixture-of-Expert (MoE) structure (Jacobs et al., 1991; Jordan and Jacobs, 1994) is a classic design that substantially scales up the model capacity and only introduces small computation overhead. In recent years, the MoE layer (Eigen et al., 2013; Shazeer et al., 2017), which is an extension of the MoE model to deep neural networks, has achieved remarkable success in deep learning. Generally speaking, an MoE layer contains many experts that share the same network architecture and are trained by the same algorithm, with a gating (or routing) function that routes individual inputs to a few experts among all the candidates. Through the sparse gating function, the router in the MoE layer

36th Conference on Neural Information Processing Systems (NeurIPS 2022).

can route each input to the top-$K(K \geq 2)$ best experts (Shazeer et al., 2017), or the single $(K = 1)$ best expert (Fedus et al., 2021). This routing scheme only costs the computation of $K$ experts for a new input, which enjoys fast inference time.

Despite the great empirical success of the MoE layer, the theoretical understanding of such architecture is still elusive. In practice, all experts have the same structure, initialized from the same weight distribution (Fedus et al., 2021) and are trained with the same optimization configuration. The router is also initialized to dispatch the data uniformly. It is unclear why the experts can diverge to different functions that are specialized to make predictions for different inputs, and why the router can automatically learn to dispatch data, especially when they are all trained using simple *local search algorithms* such as gradient descent. Therefore, we aim to answer the following questions:

> *Why do the experts in MoE diversify instead of collapsing into a single model? And how can the router learn to dispatch the data to the right expert?*

In this paper, in order to answer the above question, we consider the natural "mixture of classification" data distribution with cluster structure and theoretically study the behavior and benefit of the MoE layer. We focus on the simplest setting of the mixture of linear classification, where the data distribution has multiple clusters, and each cluster uses separate (linear) feature vectors to represent the labels. In detail, we consider the data generated as a combination of feature patches, cluster patches, and noise patches (See Definition 3.1 for more details). We study training an MoE layer based on the data generated from the "mixture of classification" distribution using gradient descent, where each expert is chosen to be a two-layer CNN. The main contributions of this paper are summarized as follows:

- We first prove a negative result (Theorem 4.1) that any single expert, such as two-layer CNNs with arbitrary activation function, cannot achieve a test accuracy of more than $87.5\%$ on our data distribution.
- Empirically, we found that the mixture of linear experts performs better than the single expert but is still significantly worse than the mixture of non-linear experts. Figure 1 provides such a result in a special case of our data distribution with four clusters. *Although a mixture of linear models can represent the labeling function of this data distribution with $100\%$ accuracy, it fails to learn so after training*. We can see that the underlying cluster structure cannot be recovered by the mixture of linear experts, and neither the router nor the experts are diversified enough after training. In contrast, the mixture of non-linear experts can correctly recover the cluster structure and diversify.
- Motivated by the negative result and the experiment on the toy data, we study a sparsely-gated MoE model with two-layer CNNs trained by gradient descent. We prove that this MoE model can achieve nearly $100\%$ test accuracy *efficiently* (Theorem 4.2).
- Along with the result on the test accuracy, we formally prove that each expert of the sparsely-gated MoE model will be specialized to a specific portion of the data (i.e., at least one cluster), which is determined by the initialization of the weights. In the meantime, the router can learn the cluster-center features and route the input data to the right experts.
- Finally, we also conduct extensive experiments on both synthetic and real datasets to corroborate our theory.

**Notation.** We use lower case letters, lower case bold face letters, and upper case bold face letters to denote scalars, vectors, and matrices respectively. We denote a union of disjoint sets $(A_i : i \in I)$ by $\sqcup_{i \in I} A_i$. For a vector $\mathbf{x}$, we use $\|\mathbf{x}\|_2$ to denote its Euclidean norm. For a matrix $\mathbf{W}$, we use $\|\mathbf{W}\|_F$ to denote its Frobenius norm. Given two sequences $\{x_n\}$ and $\{y_n\}$, we denote $x_n = \mathcal{O}(y_n)$ if $|x_n| \leq C_1|y_n|$ for some absolute positive constant $C_1$, $x_n = \Omega(y_n)$ if $|x_n| \geq C_2|y_n|$ for some absolute positive constant $C_2$, and $x_n = \Theta(y_n)$ if $C_3|y_n| \leq |x_n| \leq C_4|y_n|$ for some absolute constants $C_3, C_4 > 0$. We also use $\widetilde{\mathcal{O}}(\cdot)$ to hide logarithmic factors of $d$ in $\mathcal{O}(\cdot)$. Additionally, we denote $x_n = \text{poly}(y_n)$ if $x_n = \mathcal{O}(y_n^D)$ for some positive constant $D$, and $x_n = \text{polylog}(y_n)$ if $x_n = \text{poly}(\log(y_n))$. We also denote by $x_n = o(y_n)$ if $\lim_{n \to \infty} x_n/y_n = 0$. Finally we use $[N]$ to denote the index set $\{1, \ldots, N\}$.

## 2 Related Work

**Mixture of Experts Model.** The mixture of experts model (Jacobs et al., 1991; Jordan and Jacobs, 1994) has long been studied in the machine learning community. These MoE models are based

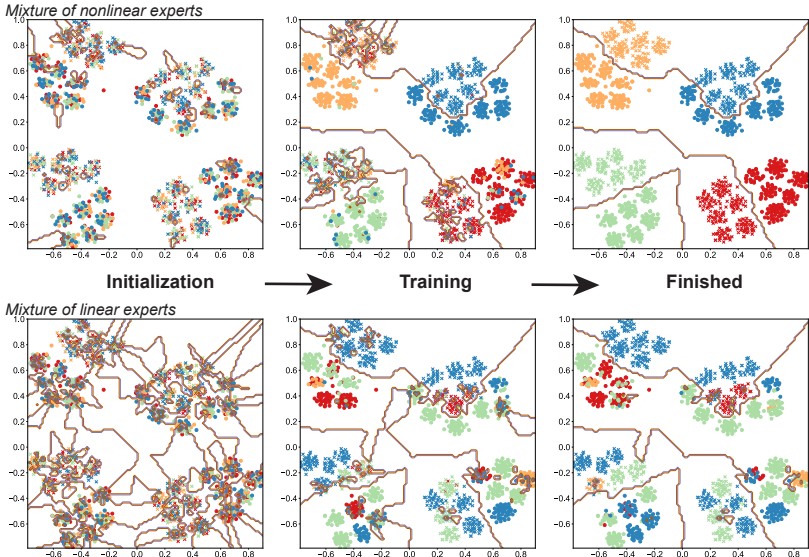

**Figure 1: Visualization of the training of MoE with nonlinear expert and linear expert**. Different colors denote router's dispatch to different experts. The lines denote the decision boundary of the MoE model. The data points are visualized on 2d space via t-SNE (Van der Maaten and Hinton, 2008). The MoE architecture follows section 3 where nonlinear experts use activation function $\sigma(z) = z^3$. For this visualization, we let the expert number $M = 4$ and cluster number $K = 4$. We generate $n = 1,600$ data points from the distribution illustrated in Section 3 with $\alpha \in (0.5, 2)$, $\beta \in (1, 2)$, $\gamma \in (1, 2)$, and $\sigma_p = 1$. More details of the visualization are discussed in Appendix A.

on various base expert models such as support vector machine (Collobert et al., 2002) , Gaussian processes (Tresp, 2001), or hidden Markov models (Jordan et al., 1997). In order to increase the model capacity to deal with the complex vision and speech data, Eigen et al. (2013) extended the MoE structure to the deep neural networks, and proposed a deep MoE model composed of multiple layers of routers and experts. Shazeer et al. (2017) simplified the MoE layer by making the output of the gating function sparse for each example, which greatly improves the training stability and reduces the computational cost. Since then, the MoE layer with different base neural network structures (Shazeer et al., 2017; Dauphin et al., 2017; Vaswani et al., 2017) has been proposed and achieved tremendous successes in a variety of language tasks. Very recently, Fedus et al. (2021) improved the performance of the MoE layer by routing one example to only a single expert instead of $K$ experts, which further reduces the routing computation while preserving the model quality.

**Mixture of Linear Regressions/Classifications.** In this paper, we consider a "mixture of classification" model. This type of models can be dated back to (De Veaux, 1989; Jordan and Jacobs, 1994; Faria and Soromenho, 2010) and has been applied to many tasks including object recognition (Quattoni et al., 2004) human action recognition (Wang and Mori, 2009), and machine translation (Liang et al., 2006). In order to learn the unknown parameters for mixture of linear regressions/classification model, (Anandkumar et al., 2012; Hsu et al., 2012; Chaganty and Liang, 2013; Anandkumar et al., 2014; Li and Liang, 2018) studies the method of moments and tensor factorization. Another line of work studies specific algorithms such as Expectation-Maximization (EM) algorithm (Khalili and Chen, 2007; Yi et al., 2014; Balakrishnan et al., 2017; Wang et al., 2015).

**Theoretical Understanding of Deep Learning.** In recent years, great efforts have been made to establish the theoretical foundation of deep learning. A series of studies have proved the convergence (Jacot et al., 2018; Li and Liang, 2018; Du et al., 2019; Allen-Zhu et al., 2019b; Zou et al., 2018) and generalization (Allen-Zhu et al., 2019a; Arora et al., 2019a,b; Cao and Gu, 2019) guarantees in the so-called "neural tangent kernel" (NTK) regime, where the parameters stay close to the initialization, and the neural network function is approximately linear in its parameters. A recent line of works (Allen-Zhu and Li, 2019; Bai and Lee, 2019; Allen-Zhu and Li, 2020a,b,c; Li et al., 2020; Cao et al., 2022; Zou et al., 2021; Wen and Li, 2021) studied the learning dynamic of neural networks beyond the NTK regime. It is worthwhile to mention that our analysis of the MoE model is also beyond the NTK regime.

# 3 Problem Setting and Preliminaries

We consider an MoE layer with each expert being a two-layer CNN trained by gradient descent (GD) over $n$ independent training examples $\{(\mathbf{x}_i, y_i)\}_{i=1}^n$ generated from a data distribution $\mathcal{D}$. In this section, we will first introduce our data model $\mathcal{D}$, and then explain our neural network model and the details of the training algorithm.

## 3.1 Data distribution

We consider a binary classification problem over $P$-patch inputs, where each patch has $d$ dimensions. In particular, each labeled data is represented by $(\mathbf{x}, y)$, where input $\mathbf{x} = (\mathbf{x}^{(1)}, \mathbf{x}^{(2)}, \ldots, \mathbf{x}^{(P)}) \in (\mathbb{R}^d)^P$ is a collection of $P$ patches and $y \in \{\pm 1\}$ is the data label. We consider data generated from $K$ clusters. Each cluster $k \in [K]$ has a label signal vector $\mathbf{v}_k$ and a cluster-center signal vector $\mathbf{c}_k$ with $\|\mathbf{v}_k\|_2 = \|\mathbf{c}_k\|_2 = 1$. For simplicity, we assume that all the signals $\{\mathbf{v}_k\}_{k \in [K]} \cup \{\mathbf{c}_k\}_{k \in [K]}$ are orthogonal with each other.

**Definition 3.1.** A data pair $(\mathbf{x}, y) \in (\mathbb{R}^d)^P \times \{\pm 1\}$ is generated from the distribution $\mathcal{D}$ as follows.

- Uniformly draw a pair $(k, k')$ with $k \neq k'$ from $\{1, \ldots, K\}$.
- Generate the label $y \in \{\pm 1\}$ uniformly, generate a Rademacher random variable $\epsilon \in \{\pm 1\}$.
- Independently generate random variables $\alpha, \beta, \gamma$ from distribution $\mathcal{D}_\alpha, \mathcal{D}_\beta, \mathcal{D}_\gamma$. In this paper, we assume there exists absolute constants $C_1, C_2$ such that almost surely $0 < C_1 \leq \alpha, \beta, \gamma \leq C_2$.
- Generate $\mathbf{x}$ as a collection of $P$ patches: $\mathbf{x} = (\mathbf{x}^{(1)}, \mathbf{x}^{(2)}, \ldots, \mathbf{x}^{(P)}) \in (\mathbb{R}^d)^P$, where
  - **Feature signal.** One and only one patch is given by $y\alpha\mathbf{v}_k$.
  - **Cluster-center signal.** One and only one patch is given by $\beta\mathbf{c}_k$.
  - **Feature noise.** One and only one patch is given by $\epsilon\gamma\mathbf{v}_{k'}$.
  - **Random noise.** The rest of the $P - 3$ patches are Gaussian noises that are independently drawn from $N(0, (\sigma_p^2/d) \cdot \mathbf{I}_d)$ where $\sigma_p$ is an absolute constant.

**How to learn this type of data?** Since the positions of signals and noises are not specified in Definition 3.1, it is natural to use the CNNs structure that applies the same function to each patch. We point out that the strength of the feature noises $\gamma$ can be as large as the strength of the feature signals $\alpha$. As we will see later in Theorem 4.1, this classification problem is hard to learn with a single expert, such as any two-layer CNNs (any activation function with any number of neurons). However, such a classification problem has an intrinsic clustering structure that may be utilized to achieve better performance. Examples can be divided into $K$ clusters $\cup_{k \in [K]} \Omega_k$ based on the cluster-center signals: an example $(\mathbf{x}, y) \in \Omega_k$ if and only if at least one patch of $\mathbf{x}$ aligns with $\mathbf{c}_k$. It is not difficult to show that the binary classification sub-problem over $\Omega_k$ can be easily solved by an individual expert. We expect the MoE can learn this data cluster structure from the cluster-center signals.

**Significance of our result.** Although this data can be learned by existing works on a mixture of linear classifiers with sophisticated algorithms (Anandkumar et al., 2012; Hsu et al., 2012; Chaganty and Liang, 2013), the focus of our paper is training a mixture of nonlinear neural networks, a more practical model used in real applications. When an MoE is trained by variants of gradient descent, we show that the experts *automatically learn to specialize on each cluster*, while the router *automatically learns to dispatch the data to the experts according to their specialty*. Although from a representation point of view, it is not hard to see that the concept class can be represented by MoEs, our result is very significant as we prove that gradient descent from random initialization can find a good MoE with non-linear experts efficiently. To make our results even more compelling, we empirically show that MoE with linear experts, despite also being able to represent the concept class, *cannot* be trained to find a good classifier efficiently.

## 3.2 Structure of the MoE layer

An MoE layer consists of a set of $M$ "expert networks" $f_1, \ldots, f_M$, and a gating network which is generally set to be linear (Shazeer et al., 2017; Fedus et al., 2021). Denote by $f_m(\mathbf{x}; \mathbf{W})$ the output of the $m$-th expert network with input $x$ and parameter $\mathbf{W}$. Define an $M$-dimensional vector $\mathbf{h}(\mathbf{x}; \boldsymbol{\Theta}) = \sum_{p \in [P]} \boldsymbol{\Theta}^\top \mathbf{x}^{(p)}$ as the output of the gating network parameterized by $\boldsymbol{\Theta} = [\boldsymbol{\theta}_1, \ldots, \boldsymbol{\theta}_M] \in \mathbb{R}^{d \times M}$. The output $F$ of the MoE layer can be written as follows:

$$F(\mathbf{x}; \boldsymbol{\Theta}, \mathbf{W}) = \sum_{m \in \mathcal{T}_\mathbf{x}} \pi_m(\mathbf{x}; \boldsymbol{\Theta}) f_m(\mathbf{x}; \mathbf{W}),$$

where $\mathcal{T}_{\mathbf{x}} \subseteq [M]$ is a set of selected indices and $\pi_m(\mathbf{x}; \boldsymbol{\Theta})$'s are route gate values given by

$$\pi_m(\mathbf{x}; \boldsymbol{\Theta}) = \frac{\exp(h_m(\mathbf{x}; \boldsymbol{\Theta}))}{\sum_{m'=1}^{M} \exp(h_{m'}(\mathbf{x}; \boldsymbol{\Theta}))}, \forall m \in [M].$$

**Expert Model.** In practice, one often uses nonlinear neural networks as experts in the MoE layer. In fact, we found that the non-linearity of the expert is essential for the success of the MoE layer (see Section 6). For $m$-th expert, we consider a convolution neural network as follows:

$$f_m(\mathbf{x}; \mathbf{W}) = \sum_{j \in [J]} \sum_{p=1}^{P} \sigma\big(\langle \mathbf{w}_{m,j}, \mathbf{x}^{(p)} \rangle\big), \tag{3.1}$$

where $\mathbf{w}_{m,j} \in \mathbb{R}^d$ is the weight vector of the $j$-th filter (i.e., neuron) in the $m$-th expert, $J$ is the number of filters (i.e., neurons). We denote $\mathbf{W}_m = [\mathbf{w}_{m,1}, \ldots, \mathbf{w}_{m,J}] \in \mathbb{R}^{d \times J}$ as the weight matrix of the $m$-th expert and further let $\mathbf{W} = \{\mathbf{W}_m\}_{m \in [M]}$ as the collection of expert weight matrices. For nonlinear CNN, we consider the cubic activation function $\sigma(z) = z^3$, which is one of the simplest nonlinear activation functions (Vecci et al., 1998). We also include the experiment for other activation functions such as RELU in Appendix Table 7.

**Top-1 Routing Model.** A simple choice of the selection set $\mathcal{T}_{\mathbf{x}}$ is the whole experts set $\mathcal{T}_{\mathbf{x}} = [M]$ (Jordan and Jacobs, 1994), which is the case for the so-called soft-routing model. However, it will be time consuming to use soft-routing in deep learning. In this paper, we consider "switch routing", which is introduced by Fedus et al. (2021) to make the gating network sparse and save the computation time. For each input $\mathbf{x}$, instead of using all the experts, we only pick one expert from $[M]$, i.e., $|\mathcal{T}_{\mathbf{x}}| = 1$. In particular, we choose $\mathcal{T}_{\mathbf{x}} = \operatorname{argmax}_m\{h_m(\mathbf{x}; \boldsymbol{\Theta})\}$.

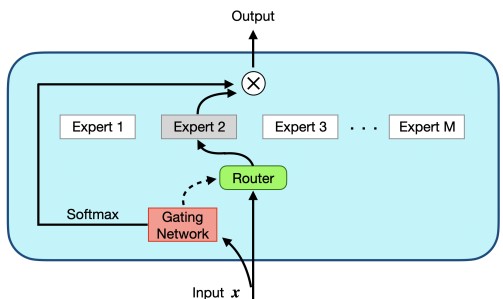

**Figure 2: Illustration of an MoE layer.** For each input $\mathbf{x}$, the router will only select one expert to perform computations. The choice is based on the output of the gating network (dotted line). The expert layer returns the output of the selected expert (gray box) multiplied by the route gate value (softmax of the gating function output).

**Algorithm 1** Gradient descent with random initialization

**Require:** Number of iterations $T$, expert learning rate $\eta$, router learning rate $\eta_r$, initialization scale $\sigma_0$, training set $S = \{(\mathbf{x}_i, y_i)\}_{i=1}^n$.
1: Generate each entry of $\mathbf{W}^{(0)}$ independently from $N(0, \sigma_0^2)$.
2: Initialize each entry of $\boldsymbol{\Theta}^{(0)}$ as zero.
3: **for** $t = 0, 2, \ldots, T - 1$ **do**
4:    Generate each entry of $\mathbf{r}^{(t)}$ independently from Unif[0,1].
5:    Update $\mathbf{W}^{(t+1)}$ as in (3.4).
6:    Update $\boldsymbol{\Theta}^{(t+1)}$ as in (3.5).
7: **end for**
8: **return** $(\boldsymbol{\Theta}^{(T)}, \mathbf{W}^{(T)})$.

### 3.3 Training Algorithm

Given the training data $S = \{(\mathbf{x}_i, y_i)\}_{i=1}^n$, we train $F$ with gradient descent to minimize the following empirical loss function:

$$\mathcal{L}(\boldsymbol{\Theta}, \mathbf{W}) = \frac{1}{n} \sum_{i=1}^{n} \ell\big(y_i F(\mathbf{x}_i; \boldsymbol{\Theta}, \mathbf{W})\big), \tag{3.2}$$

where $\ell$ is the logistic loss defined as $\ell(z) = \log(1 + \exp(-z))$. We initialize $\boldsymbol{\Theta}^{(0)}$ to be zero and initialize each entry of $\mathbf{W}^{(0)}$ by i.i.d $\mathcal{N}(0, \sigma_0^2)$. Zero initialization of the gating network is widely used in MoE training. As discussed in Shazeer et al. (2017), it can help avoid out-of-memory errors and initialize the network in a state of approximately equal expert load (see (5.1) for the definition of expert load).

Instead of directly using the gradient of empirical loss (3.2) to update weights, we add perturbation to the router and use the gradient of the perturbed empirical loss to update the weights. In particular, the training example $\mathbf{x}_i$ will be distributed to $\operatorname{argmax}_m\{h_m(\mathbf{x}_i; \boldsymbol{\Theta}^{(t)}) + r_{m,i}^{(t)}\}$ instead, where $\{r_{m,i}^{(t)}\}_{m \in [M], i \in [n]}$ are random noises. Adding noise term is a widely used training strategy

for sparsely-gated MoE layer (Shazeer et al., 2017; Fedus et al., 2021), which can encourage exploration across the experts and stabilize the MoE training. In this paper, we draw $\{r_{m,i}^{(t)}\}_{m\in[M],i\in[n]}$ independently from the uniform distribution $\text{Unif}[0,1]$ and denotes its collection as $\mathbf{r}^{(t)}$. Therefore, the perturbed empirical loss at iteration $t$ can be written as

$$\mathcal{L}^{(t)}(\boldsymbol{\Theta}^{(t)}, \mathbf{W}^{(t)}) = \frac{1}{n}\sum_{i=1}^{n}\ell\big(y_i\pi_{m_{i,t}}(\mathbf{x}_i; \boldsymbol{\Theta}^{(t)})f_{m_{i,t}}(\mathbf{x}_i; \mathbf{W}^{(t)})\big), \qquad (3.3)$$

where $m_{i,t} = \arg\max_m\{h_m(\mathbf{x}_i; \boldsymbol{\Theta}^{(t)}) + r_{m,i}^{(t)}\}$. Starting from the initialization $\mathbf{W}^{(0)}$, the gradient descent update rule for the experts is

$$\mathbf{W}_m^{(t+1)} = \mathbf{W}_m^{(t)} - \eta \cdot \nabla_{\mathbf{w}_m}\mathcal{L}^{(t)}(\boldsymbol{\Theta}^{(t)}, \mathbf{W}^{(t)})/\|\nabla_{\mathbf{w}_m}\mathcal{L}^{(t)}(\boldsymbol{\Theta}^{(t)}, \mathbf{W}^{(t)})\|_F, \forall m \in [M], \quad (3.4)$$

where $\eta > 0$ is the expert learning rate. Starting from the initialization $\boldsymbol{\Theta}^{(0)}$, the gradient update rule for the gating network is

$$\boldsymbol{\theta}_m^{(t+1)} = \boldsymbol{\theta}_m^{(t)} - \eta_r \cdot \nabla_{\boldsymbol{\theta}_m}\mathcal{L}^{(t)}(\boldsymbol{\Theta}^{(t)}, \mathbf{W}^{(t)}), \forall m \in [M], \qquad (3.5)$$

where $\eta_r > 0$ is the router learning rate. In practice, the experts are trained by Adam to make sure they have similar learning speeds. Here we use a normalized gradient which can be viewed as a simpler alternative to Adam (Jelassi et al., 2021).

## 4 Main Results

In this section, we will present our main results. We first provide a negative result for learning with a single expert.

**Theorem 4.1** (Single expert performs poorly). *Suppose $\mathcal{D}_\alpha = \mathcal{D}_\gamma$ in Definition 3.1, then any function with the form $F(\mathbf{x}) = \sum_{p=1}^{P}f(\mathbf{x}^{(p)})$ will get large test error $\mathbb{P}_{(\mathbf{x},y)\sim\mathcal{D}}\big(yF(\mathbf{x}) \leq 0\big) \geq 1/8$.*

Theorem 4.1 indicates that if the feature noise has the same strength as the feature signal i.e., $\mathcal{D}_\alpha = \mathcal{D}_\gamma$, any two-layer CNNs with the form $F(\mathbf{x}) = \sum_{j\in[J]}a_j\sum_{p\in[P]}\sigma(\mathbf{w}_j^\top\mathbf{x}^{(p)} + b_j)$ can't perform well on the classification problem defined in Definition 3.1 where $\sigma$ can be any activation function. Theorem 4.1 also shows that a simple ensemble of the experts may not improve the performance because the ensemble of the two-layer CNNs is still in the form of the function defined in Theorem 4.1.

As a comparison, the following theorem gives the learning guarantees for training an MoE layer that follows the structure defined in Section 3.2 with cubic activation function.

**Theorem 4.2** (Nonlinear MoE performs well). *Suppose the training data size $n = \Omega(d)$. Choose experts number $M = \Theta(K\log K\log\log d)$, filter size $J = \Theta(\log M\log\log d)$, initialization scale $\sigma_0 \in [d^{-1/3}, d^{-0.01}]$, learning rate $\eta = \widetilde{O}(\sigma_0), \eta_r = \Theta(M^2)\eta$. Then with probability at least $1 - o(1)$, Algorithm 1 is able to output $(\boldsymbol{\Theta}^{(T)}, \mathbf{W}^{(T)})$ within $T = \widetilde{O}(\eta^{-1})$ iterations such that the non-linear MoE defined in Section 3.2 satisfies that*

- *Training error is zero, i.e., $y_iF(\mathbf{x}_i; \boldsymbol{\Theta}^{(T)}, \mathbf{W}^{(T)}) > 0, \forall i \in [n]$.*
- *Test error is nearly zero, i.e., $\mathbb{P}_{(\mathbf{x},y)\sim\mathcal{D}}\big(yF(\mathbf{x}; \boldsymbol{\Theta}^{(T)}, \mathbf{W}^{(T)}) \leq 0\big) = o(1)$.*

*More importantly, the experts can be divided into a disjoint union of $K$ non-empty sets $[M] = \sqcup_{k\in[K]}\mathcal{M}_k$ and*

- *(Each expert is good on one cluster) Each expert $m \in \mathcal{M}_k$ performs good on the cluster $\Omega_k$, $\mathbb{P}_{(\mathbf{x},y)\sim\mathcal{D}}(yf_m(\mathbf{x}; \mathbf{W}^{(T)}) \leq 0|(\mathbf{x},y) \in \Omega_k) = o(1)$.*
- *(Router only distributes example to good expert) With probability at least $1 - o(1)$, an example $\mathbf{x} \in \Omega_k$ will be routed to one of the experts in $\mathcal{M}_k$.*

Theorem 4.2 shows that a non-linear MoE performs well on the classification problem in Definition 3.1. In addition, the router will learn the cluster structure and divide the problem into $K$ simpler subproblems, each of which is associated with one cluster. In particular, each cluster will be classified accurately by a subset of experts. On the other hand, each expert will perform well on at least one cluster.

Furthermore, together with Theorem 4.1, Theorem 4.2 suggests that there exist problem instances in Definition 3.1 (i.e., $\mathcal{D}_\alpha = \mathcal{D}_\gamma$) such that an MoE provably outperforms a single expert.

# 5 Overview of Key Techniques

A successful MoE layer needs to ensure that the router can learn the cluster-center features and divide the complex problem in Definition 3.1 into simpler linear classification sub-problems that individual experts can conquer. Finding such a gating network is difficult because this problem is highly non-convex. In the following, we will introduce the main difficulties in analyzing the MoE layer and the corresponding key techniques to overcome those barriers.

**Main Difficulty 1: Discontinuities in Routing.** Compared with the traditional soft-routing model, the sparse routing model saves computation and greatly reduces the inference time. However, this form of sparsity also causes discontinuities in routing (Shazeer et al., 2017). In fact, even a small perturbation of the gating network outputs $\mathbf{h}(\mathbf{x}; \boldsymbol{\Theta}) + \boldsymbol{\delta}$ may change the router behavior drastically if the second largest gating network output is close to the largest gating network output.

**Key Technique 1: Stability by Smoothing.** We point out that the noise term added to the gating network output ensures a smooth transition between different routing behavior, which makes the router more stable. This is proved in the following lemma.

**Lemma 5.1.** Let $\mathbf{h}, \widehat{\mathbf{h}} \in \mathbb{R}^M$ to be the output of the gating network and $\{r_m\}_{m=1}^M$ to be the noise independently drawn from Unif[0,1]. Denote $\mathbf{p}, \widehat{\mathbf{p}} \in \mathbb{R}^M$ to be the probability that experts get routed, i.e., $p_m = \mathbb{P}(\arg\max_{m' \in [M]}\{h_{m'} + r_{m'}\} = m)$, $\widehat{p}_m = \mathbb{P}(\arg\max_{m' \in [M]}\{\widehat{h}_{m'} + r_{m'}\} = m)$. Then we have that $\|\mathbf{p} - \widehat{\mathbf{p}}\|_\infty \le M^2 \|\mathbf{h} - \widehat{\mathbf{h}}\|_\infty$.

Lemma 5.1 implies that when the change of the gating network outputs at iteration $t$ and $t'$ is small, i.e., $\|\mathbf{h}(\mathbf{x}; \boldsymbol{\Theta}^{(t)}) - \mathbf{h}(\mathbf{x}; \boldsymbol{\Theta}^{(t')})\|_\infty$, the router behavior will be similar. So adding noise provides a smooth transition from time $t$ to $t'$. It is also worth noting that $\boldsymbol{\Theta}$ is zero initialized. So $\mathbf{h}(\mathbf{x}; \boldsymbol{\Theta}^{(0)}) = 0$ and thus each expert gets routed with the same probability $p_m = 1/M$ by symmetric property. Therefore, at the early of the training when $\|\mathbf{h}(\mathbf{x}; \boldsymbol{\Theta}^{(t)}) - \mathbf{h}(\mathbf{x}; \boldsymbol{\Theta}^{(0)})\|_\infty$ is small, router will almost uniformly pick one expert from $[M]$, which helps exploration across experts.

**Main Difficulty 2: No "Real" Expert.** At the beginning of the training, the gating network is zero, and the experts are randomly initialized. Thus it is hard for the router to learn the right features because all the experts look the same: they share the same network architecture and are trained by the same algorithm. The only difference is the initialization. Moreover, if the router makes a mistake at the beginning of the training, the experts may amplify the mistake because the experts will be trained based on mistakenly dispatched data.

**Key Technique 2: Experts from Exploration.** Motivated by the key technique 1, we introduce an exploration stage to the analysis of MoE layer during which the router almost uniformly picks one expert from $[M]$. This stage starts at $t = 0$ and ends at $T_1 = \lfloor \eta^{-1}\sigma_0^{0.5} \rfloor \ll T = \widetilde{O}(\eta^{-1})$ and the gating network remains nearly unchanged $\|\mathbf{h}(\mathbf{x}; \boldsymbol{\Theta}^{(t)}) - \mathbf{h}(\mathbf{x}; \boldsymbol{\Theta}^{(0)})\|_\infty = O(\sigma_0^{1.5})$. Because the experts are treated almost equally during exploration stage, we can show that the experts become specialized to some specific task only based on the initialization. In particular, the experts set $[M]$ can be divided into $K$ nonempty disjoint sets $[M] = \sqcup_k \mathcal{M}_k$, where $\mathcal{M}_k := \{m | \arg\max_{k' \in [K], j \in [J]} \langle \mathbf{v}_{k'}, \mathbf{w}_{m,j}^{(0)} \rangle = k\}$. For nonlinear MoE with cubic activation function, the following lemma further shows that experts in different set $\mathcal{M}_k$ will diverge at the end of the exploration stage.

**Lemma 5.2.** Under the same condition as in Theorem 4.2, with probability at least $1 - o(1)$, the following equations hold for all expert $m \in \mathcal{M}_k$,

$$\mathbb{P}_{(\mathbf{x},y)\sim\mathcal{D}}\big(yf_m(\mathbf{x}; \mathbf{W}^{(T_1)}) \le 0 \big| (\mathbf{x}, y) \in \Omega_k\big) = o(1),$$
$$\mathbb{P}_{(\mathbf{x},y)\sim\mathcal{D}}\big(yf_m(\mathbf{x}; \mathbf{W}^{(T_1)}) \le 0 \big| (\mathbf{x}, y) \in \Omega_{k'}\big) = \Omega(1/K), \forall k' \ne k.$$

Lemma 5.2 implies that, at the end of the exploration stage, the expert $m \in \mathcal{M}_k$ can achieve nearly zero test error on the cluster $\Omega_k$ but high test error on the other clusters $\Omega_{k'}, k' \ne k$.

**Main Difficulty 3: Expert Load Imbalance.** Given the training data set $S = \{(\mathbf{x}_i, y_i)\}_{i=1}^n$, the load of expert $m$ at iterate $t$ is defined as

$$\text{Load}_m^{(t)} = \sum_{i \in [n]} \mathbb{P}(m_{i,t} = m), \tag{5.1}$$

where $\mathbb{P}(m_{i,t} = m)$ is probability that the input $\mathbf{x}_i$ being routed to expert $m$ at iteration $t$. Eigen et al. (2013) first described the load imbalance issues in the training of the MoE layer. The gating

network may converge to a state where it always produces large $\text{Load}_m^{(t)}$ for the same few experts. This imbalance in expert load is self-reinforcing, as the favored experts are trained more rapidly and thus are selected even more frequently by the router (Shazeer et al., 2017; Fedus et al., 2021). Expert load imbalance issue not only causes memory and performance problems in practice, but also impedes the theoretical analysis of the expert training.

**Key Technique 3: Normalized Gradient Descent.** Lemma 5.2 shows that the experts will diverge into $\sqcup_{k \in [K]} \mathcal{M}_k$. Normalized gradient descent can help different experts in the same $\mathcal{M}_k$ being trained at the same speed regardless of the imbalance load caused by the router. Because the self-reinforcing circle no longer exists, the load imbalance issue will get mitigated. In particular, the router will treat different experts in the same $\mathcal{M}_k$ almost equally and dispatch almost the same amount of data to them during the early stage of training (See Section E.2 in Appendix for detail), which is enough for the router to learn the cluster-center features. However, we can't guarantee load balance for an arbitrary long training period if we only use normalized gradient descent. That's the reason Theorem 4.2 requires early stopping. This load imbalance issue can be further avoided by adding load balancing loss (Eigen et al., 2013; Shazeer et al., 2017; Fedus et al., 2021), or using advanced MoE layer structure such as BASE Layers (Lewis et al., 2021; Dua et al., 2021) and Hash Layers (Roller et al., 2021).

**Road Map:** Here we provide the road map of the proof of Theorem 4.2 and the full proof is presented in Appendix E. The training process can be decomposed into several stages. The first stage is called *Exploration stage*. During this stage, the experts will diverge into $K$ professional groups $\sqcup_{k=1}^K \mathcal{M}_k = [M]$. In particular, we will show that $\mathcal{M}_k$ is not empty for all $k \in [K]$. Besides, for all $m \in \mathcal{M}_k$, $f_m$ is a good classifier over $\Omega_k$. The second stage is called *router learning stage*. During this stage, the router will learn to dispatch $\mathbf{x} \in \Omega_k$ to one of the experts in $\mathcal{M}_k$. Finally, we will give the generalization analysis for the MoEs from the previous two stages.

# 6 Experiments

In this section, we conduct experiments to validate our theory. The code and data for our experiments can be found on Github [1].

Setting 1: $\alpha \in (0.5, 2), \beta \in (1, 2), \gamma \in (0.5, 3), \sigma_p = 1$

|  | Test accuracy (%) | Dispatch Entropy |
|---|---|---|
| Single (linear) | 68.71 | NA |
| Single (nonlinear) | 79.48 | NA |
| MoE (linear) | $92.99 \pm 2.11$ | $1.300 \pm 0.044$ |
| MoE (nonlinear) | $\mathbf{99.46 \pm 0.55}$ | $\mathbf{0.098 \pm 0.087}$ |

Setting 2: $\alpha \in (0.5, 2), \beta \in (1, 2), \gamma \in (0.5, 3), \sigma_p = 2$

|  | Test accuracy (%) | Dispatch Entropy |
|---|---|---|
| Single (linear) | 60.59 | NA |
| Single (nonlinear) | 72.29 | NA |
| MoE (linear) | $88.48 \pm 1.96$ | $1.294 \pm 0.036$ |
| MoE (nonlinear) | $\mathbf{98.09 \pm 1.27}$ | $\mathbf{0.171 \pm 0.103}$ |

**Table 1: Comparison between MoE (linear) and MoE (nonlinear)** in our setting. We report results of top-1 gating with noise for both linear and nonlinear models. Over ten random experiments, we report the average value $\pm$ standard deviation for both test accuracy and dispatch entropy.

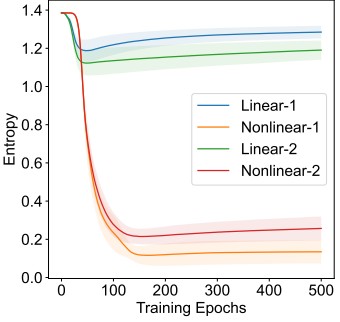

**Figure 3: Illustration of router dispatch entropy.** We demonstrate the change of entropy of MoE during training on the synthetic data. MoE (linear)-1 and MoE (nonlinear)-1 refer to Setting 1 in Table 1. MoE (linear)-2 and MoE (nonlinear)-2 refer to Setting 2 in Table 1.

## 6.1 Synthetic-data Experiments

**Datasets.** We generate $16,000$ training examples and $16,000$ test examples from the data distribution defined in Definition 3.1 with cluster number $K = 4$, patch number $P = 4$ and dimension $d = 50$. We randomly shuffle the order of the patches of $\mathbf{x}$ after we generate data $(\mathbf{x}, y)$. We consider two

---

[1]https://github.com/uclaml/MoE

**Table 2:** Comparison between MoE and single model on CIFAR-10 and CIFAR-10-Rotate datasets. We report the average test accuracy over 10 random experiments $\pm$ the standard deviation.

|  |  | CIFAR-10 (%) | CIFAR-10-Rotate (%) |
|---|---|---|---|
| CNN | Single | $80.68 \pm 0.45$ | $76.78 \pm 1.79$ |
|  | MoE | $80.31 \pm 0.62$ | $\mathbf{79.60 \pm 1.25}$ |
| MobileNetV2 | Single | $92.45 \pm 0.25$ | $85.76 \pm 2.91$ |
|  | MoE | $92.23 \pm 0.72$ | $\mathbf{89.85 \pm 2.54}$ |
| ResNet18 | Single | $95.51 \pm 0.31$ | $88.23 \pm 0.96$ |
|  | MoE | $95.32 \pm 0.68$ | $\mathbf{92.60 \pm 2.01}$ |

parameter settings: 1. $\alpha \sim \text{Uniform}(0.5, 2)$, $\beta \sim \text{Uniform}(1, 2)$, $\gamma \sim \text{Uniform}(0.5, 3)$ and $\sigma_p = 1$; 2. $\alpha \sim \text{Uniform}(0.5, 2)$, $\beta \sim \text{Uniform}(1, 2)$, $\gamma \sim \text{Uniform}(0.5, 3)$ and $\sigma_p = 2$. Note that Theorem 4.1 shows that when $\alpha$ and $\gamma$ follow the same distribution, neither single linear expert or single nonlinear expert can give good performance. Here we consider a more general and difficult setting when $\alpha$ and $\gamma$ are from different distributions.

**Models.** We consider the performances of single linear CNN, single nonlinear CNN, linear MoE, and nonlinear MoE. The single nonlinear CNN architecture follows (3.1) with cubic activation function, while single linear CNN follows (3.1) with identity activation function. For both linear and nonlinear MoEs, we consider a mixture of 8 experts with each expert being a single linear CNN or a single nonlinear CNN. Finally, we train single models with gradient descent and train the MoEs with Algorithm 1. We run 10 random experiments and report the average accuracy with standard deviation.

**Evaluation.** To evaluate how well the router learned the underlying cluster structure of the data, we define the entropy of the router's dispatch as follows. Denote by $n_{k,m}$ the number of data in cluster $K$ that are dispatched to expert $m$. The total number of data dispatched to expert $m$ is $n_m = \sum_{k=1}^{K} n_{k,m}$ and the total number of data is $n = \sum_{k=1}^{K} \sum_{m=1}^{M} n_{k,m}$. The dispatch entropy is then defined as

$$\text{entropy} = -\sum_{m=1, n_m \neq 0}^{M} \frac{n_m}{n} \sum_{k=1}^{K} \frac{n_{k,m}}{n_m} \cdot \log\left(\frac{n_{k,m}}{n_m}\right). \tag{6.1}$$

When each expert receives the data from at most one cluster, the dispatch entropy will be zero. And a uniform dispatch will result in the maximum dispatch entropy.

As shown in Table 1, the linear MoE does not perform as well as the nonlinear MoE in Setting 1, with around 6% less test accuracy and much higher variance. With stronger random noise (Setting 2), the difference between the nonlinear MoE and linear MoE becomes even more significant. We also observe that the final dispatch entropy of nonlinear MoE is nearly zero while that of the linear MoE is large. In Figure 3, we further demonstrate the change of dispatch entropy during the training process. The dispatch entropy of nonlinear MoE significantly decreases, while that of linear MoE remains large. Such a phenomenon indicates that the nonlinear MoE can successfully learn the underlying cluster structure of the data while the linear MoE fails to do so.

## 6.2 Real-data Experiments

We further conduct experiments on real image datasets and demonstrate the importance of the clustering data structure to the MoE layer in deep neural networks.

**Datasets.** We consider the **CIFAR-10** dataset (Krizhevsky, 2009) and the 10-class classification task. Furthermore, we create a **CIFAR-10-Rotate** dataset that has a strong underlying cluster structure that is independent of its labeling function. Specifically, we rotate the images by 30 degrees and merge the rotated dataset with the original one. The task is to predict if the image is rotated, which is a binary classification problem. We deem that some of the classes in CIFAR-10 form underlying clusters in CIFAR-10-Rotate. In Appendix A, we explain in detail how we generate CIFAR-10-Rotate and present some specific examples.

**Models.** For the MoE, we consider a mixture of 4 experts with a linear gating network. For the expert/single model architectures, we consider a CNN with 2 convolutional layers (architecture details are illustrated in Appendix A.) For a more thorough evaluation, we also consider expert/single models

with architecture including **MobileNetV2** (Sandler et al., 2018) and **ResNet18** (He et al., 2016). The training process of MoE also follows Algorithm 1.

The experiment results are shown in Table 2, where we compare single and mixture models of different architectures over CIFAR-10 and CIFAR-10-Rotate datasets. We observe that the improvement of MoEs over single models differs largely on the different datasets. On CIFAR-10, the performance of MoEs is very close to the single models. However, on the CIFAR-10-Rotate dataset, we can observe a significant performance improvement from single models to MoEs. Such results indicate the advantage of MoE over single models depends on the task and the cluster structure of the data.

**Visualization.** In Figure 4, we visualize the latent embedding learned by MoEs (ResNet18) for the 10-class classification task in CIFAR-10 as well as the binary classification task in CIFAR-10-Rotate. We visualize the data with the same label $y$ to see if cluster structures exist within each class. For CIFAR-10, we choose $y = 1$ ("car"), and plot the latent embedding of the data using t-SNE on the left sub-figure, which does not show an salient cluster structure. For CIFAR-10-Rotate, we choose $y = 1$ ("rotated") and visualize the data using t-SNE in the middle sub-figure. Here, we can observe a clear clustering structure even though the class signal is not provided during training. We take a step further to investigate what is in each cluster in the right sub-figure. We can observe that most of the examples in the "frog" class fall into one cluster, while examples of "ship" class mostly fall into the other cluster.

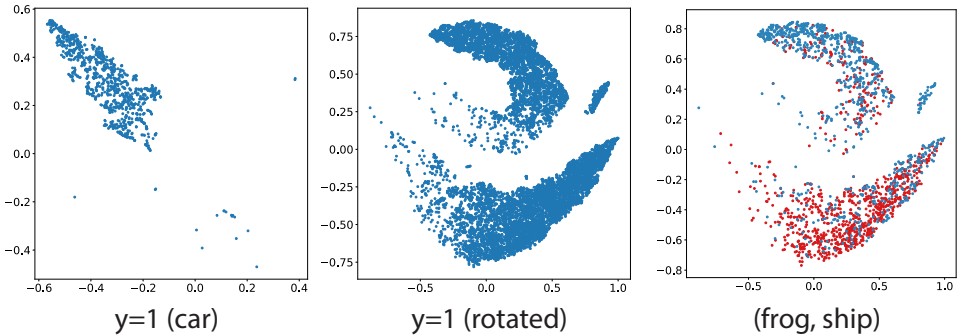

**Figure 4:** Visualization of the latent embedding on CIFAR-10 and CIFAR-10-Rotate with chosen label $y$. The left sub-figure denotes the visualization of CIFAR-10 when label $y$ is chosen to be 1 (car). The central sub-figure represents the visualization of CIFAR-10-Rotate when label $y$ is chosen to be 1 (rotated). On the right sub-figure, red denotes that the data is from the ship class, and blue denotes that the data is from the frog class.

## 7   Conclusion and Future Work

In this work, we formally study the mechanism of the Mixture of Experts (MoE) layer for deep learning. To our knowledge, we provide the first theoretical result toward understanding how the MoE layer works in deep learning. Our empirical evidence reveals that the cluster structure of the data plays an important role in the success of the MoE layer. Motivated by these empirical observations, we study a data distribution with cluster structure and show that Mixture-of-Experts provably improves the test accuracy of a single expert of two-layer CNNs.

There are several important future directions. First, our current results are for CNNs. It is interesting to extend our results to other neural network architectures, such as transformers. Second, our data distribution is motivated by the classification problem of image data. We plan to extend our analysis to other types of data (e.g., natural language data).

## Acknowledgments and Disclosure of Funding

We thank the anonymous reviewers and area chair for their helpful comments. ZC, YD, YW and QG are supported in part by the National Science Foundation CAREER Award 1906169, BIGDATA IIS-1855099, and the Sloan Research Fellowship. YL is supported in part by the NSF RI2007517. The views and conclusions contained in this paper are those of the authors and should not be interpreted as representing any funding agencies.

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
