# A  Experiment Details

## A.1  Visualization

In the visualization of Figure 1, MoE (linear) and MoE (nonlinear) are trained according to Algorithm 1 by normalized gradient descent with learning rate $0.001$ and gradient descent with learning rate $0.1$. According to Definition 3.1, we set $K = 4$, $P = 4$ and $d = 50$ and choose $\alpha \in (0.5, 2)$, $\beta \in (1, 2)$, $\gamma \in (1, 2)$ and $\sigma_p = 1$, and generate $3,200$ data examples. We consider mixture of $M = 4$ experts for both MoE (linear) and MoE (nonlinear). For each expert, we set the number of neurons/filters $J = 16$. We train MoEs on $1,600$ data examples and visualize classification result and decision boundary on the remaining $1,600$ examples. The data examples are visualized via t-SNE (Van der Maaten and Hinton, 2008). When visualizing the data points and decision boundary on the 2d space, we increase the magnitude of random noise patch by $3$ so that the positive/negative examples and decision boundaries can be better viewed.

## A.2  Synthetic-data Experiments

**Synthetic-data experiment setup.**  For the experiments on synthetic data, we generate the data according to Definition 3.1 with $K = 4$, $P = 4$ and $d = 50$. We consider four parameter settings:

- $\alpha \sim \text{Uniform}(0.5, 2)$, $\beta \sim \text{Uniform}(1, 2)$, $\gamma \sim \text{Uniform}(0.5, 3)$ and $\sigma_p = 1$;

- $\alpha \sim \text{Uniform}(0.5, 2)$, $\beta \sim \text{Uniform}(1, 2)$, $\gamma \sim \text{Uniform}(0.5, 3)$ and $\sigma_p = 2$;

- $\alpha \sim \text{Uniform}(0.5, 2)$, $\beta \sim \text{Uniform}(1, 2)$, $\gamma \sim \text{Uniform}(0.5, 2)$ and $\sigma_p = 1$;

- $\alpha \sim \text{Uniform}(0.5, 2)$, $\beta \sim \text{Uniform}(1, 2)$, $\gamma \sim \text{Uniform}(0.5, 2)$ and $\sigma_p = 2$.

We consider mixture of $M = 8$ experts for all MoEs and $J = 16$ neurons/filters for all experts. For single models, we consider $J = 128$ neurons/filters. We train MoEs using Algorithm 1. Specifically, we train the experts by normalized gradient descent with learning rate $0.001$ and the gating network by gradient descent with learning rate $0.1$. We train single linear/nonlinear models by Adam (Kingma and Ba, 2014) to achieve the best performance, with learning rate $0.01$ and weight decay 5e-4 for single nonlinear model and learning rate $0.003$ and weight decay $5e - 4$ for single linear model.

**Synthetic-data experiment results.**  In Table 3, we present the empirical results of single linear CNN, single nonlinear CNN, linear MoE, and nonlinear MoE under settings 3 and 4, where $\alpha$ and $\gamma$ follow the same distribution as we assumed in theoretical analysis. Furthermore, we report the total number of filters for both single CNNs and a mixture of CNNs, where the filter size (equal to 50) is the same for all single models and experts. For linear and nonlinear MoE, there are 16 filters for each of the 8 experts, and therefore 128 filters in total. Note that in the synthetic-data experiment in the main paper, we let the number of filters of single models be the same as MoEs (128). Here, we additionally report the performances of single models with 512 filters, and see if increasing the model size of single models can beat MoE. From Table 3, we observe that: 1. single models perform poorly in all settings; 2. linear MoEs do not perform as well as nonlinear MoEs. Specifically, the final dispatch entropy of nonlinear MoEs is nearly zero while the dispatch entropy of linear MoEs is consistently larger under settings 1-4. This indicates that nonlinear MoEs successfully uncover the underlying cluster structure while linear MoEs fail to do so. In addition, we can see that even larger single models cannot beat linear MoEs or nonlinear MoEs. This is consistent with Theorem 4.1, where a single model fails under such data distribution regardless of its model size. Notably, by comparing the results in Table 1 and Table 3, we can see that a single nonlinear model suffers from overfitting as we increase the number of filters.

**Router dispatch examples.**  We demonstrate specific examples of router dispatch for MoE (nonlinear) and MoE (linear). The examples of initial and final router dispatch for MoE (nonlinear) are shown in Table 4 and Table 5. Under the dispatch for nonlinear MoE, each expert is given either no data or data that comes from one cluster only. The entropy of such dispatch is thus 0. The test accuracy of MoE trained under such a dispatch is either $100\%$ or very close to $100\%$, as the expert can be easily trained on the data from one cluster only. An example of the final dispatch for MoE (linear) is shown in Table 6, where clusters are not well separated and an expert gets data from different clusters. The test accuracy under such dispatch is lower ($90.61\%$).

**Table 3: Comparison between MoE (linear) and MoE (nonlinear)** in our setting. We report results of top-1 gating with noise for both linear and nonlinear models. Over ten random experiments, we report the average value $\pm$ standard deviation for both test accuracy and dispatch entropy.

Setting 1: $\alpha \in (0.5, 2), \beta \in (1, 2), \gamma \in (0.5, 3), \sigma_p = 1$

|  | Test accuracy (%) | Dispatch Entropy | Number of Filters |
|---|---|---|---|
| Single (linear) | 68.71 | NA | 128 |
| Single (linear) | 67.63 | NA | 512 |
| Single (nonlinear) | 79.48 | NA | 128 |
| Single (nonlinear) | 78.18 | NA | 512 |
| MoE (linear) | $92.99 \pm 2.11$ | $1.300 \pm 0.044$ | 128 (16*8) |
| MoE (nonlinear) | $\mathbf{99.46 \pm 0.55}$ | $\mathbf{0.098 \pm 0.087}$ | 128 (16*8) |

Setting 2: $\alpha \in (0.5, 2), \beta \in (1, 2), \gamma \in (0.5, 3), \sigma_p = 2$

|  | Test accuracy (%) | Dispatch Entropy | Number of Filters |
|---|---|---|---|
| Single (linear) | 60.59 | NA | 128 |
| Single (linear) | 63.04 | NA | 512 |
| Single (nonlinear) | 72.29 | NA | 128 |
| Single (nonlinear) | 52.09 | NA | 512 |
| MoE (linear) | $88.48 \pm 1.96$ | $1.294 \pm 0.036$ | 128 (16*8) |
| MoE (nonlinear) | $\mathbf{98.09 \pm 1.27}$ | $\mathbf{0.171 \pm 0.103}$ | 128 (16*8) |

Setting 3: $\alpha \in (0.5, 2), \beta \in (1, 2), \gamma \in (0.5, 2), \sigma_p = 1$

|  | Test accuracy (%) | Dispatch Entropy | Number of Filters |
|---|---|---|---|
| Single (linear) | 74.81 | NA | 128 |
| Single (linear) | 74.54 | NA | 512 |
| Single (nonlinear) | 72.69 | NA | 128 |
| Single (nonlinear) | 67.78 | NA | 512 |
| MoE (linear) | $95.93 \pm 1.34$ | $1.160 \pm 0.100$ | 128 (16*8) |
| MoE (nonlinear) | $\mathbf{99.99 \pm 0.02}$ | $\mathbf{0.008 \pm 0.011}$ | 128 (16*8) |

Setting 4: $\alpha \in (0.5, 2), \beta \in (1, 2), \gamma \in (0.5, 2), \sigma_p = 2$

|  | Test accuracy (%) | Dispatch Entropy | Number of Filters |
|---|---|---|---|
| Single (linear) | 74.63 | NA | 128 |
| Single (linear) | 72.98 | NA | 512 |
| Single (nonlinear) | 68.60 | NA | 128 |
| Single (nonlinear) | 61.65 | NA | 512 |
| MoE (linear) | $93.30 \pm 1.48$ | $1.160 \pm 0.155$ | 128 (16*8) |
| MoE (nonlinear) | $\mathbf{98.92 \pm 1.18}$ | $\mathbf{0.089 \pm 0.120}$ | 128 (16*8) |

**Table 4:** Dispatch details of MoE (nonlinear) with test accuracy $100\%$.

| Expert number | 1 | 2 | 3 | 4 | 5 | 6 | 7 | 8 |
|---|---|---|---|---|---|---|---|---|
| Initial dispatch | 1921 | 2032 | 1963 | 1969 | 2075 | 1980 | 2027 | 2033 |
| Final dispatch | 0 | 3979 | 4009 | 0 | 0 | 3971 | 0 | 4041 |
| Cluster 1 | 0 | 0 | 0 | 0 | 0 | 3971 | 0 | 0 |
| Cluster 2 | 0 | 0 | 4009 | 0 | 0 | 0 | 0 | 0 |
| Cluster 3 | 0 | 0 | 0 | 0 | 0 | 0 | 0 | 4041 |
| Cluster 4 | 0 | 3979 | 0 | 0 | 0 | 0 | 0 | 0 |

**Table 5:** Dispatch details of MoE (nonlinear) with test accuracy 99.95%.

| Expert number | 1 | 2 | 3 | 4 | 5 | 6 | 7 | 8 |
|---|---|---|---|---|---|---|---|---|
| Initial dispatch | 1978 | 2028 | 2018 | 1968 | 2000 | 2046 | 2000 | 1962 |
| Final dispatch | 3987 | 4 | 3975 | 6 | 0 | 1308 | 4009 | 2711 |
| Cluster 1 | 0 | 0 | 3971 | 0 | 0 | 0 | 0 | 0 |
| Cluster 2 | 0 | 0 | 0 | 0 | 0 | 4 | 4005 | 0 |
| Cluster 3 | 8 | 4 | 4 | 6 | 0 | 1304 | 4 | 2711 |
| Cluster 4 | 3979 | 0 | 0 | 0 | 0 | 0 | 0 | 0 |

**Table 6:** Dispatch details of MoE (linear) with test accuracy 90.61%.

| Expert number | 1 | 2 | 3 | 4 | 5 | 6 | 7 | 8 |
|---|---|---|---|---|---|---|---|---|
| Initial dispatch | 1969 | 2037 | 1983 | 2007 | 1949 | 1905 | 2053 | 2097 |
| Final dispatch | 136 | 2708 | 6969 | 5311 | 27 | 87 | 4 | 758 |
| Cluster 1 | 0 | 630 | 1629 | 1298 | 27 | 87 | 4 | 296 |
| Cluster 2 | 136 | 1107 | 1884 | 651 | 0 | 0 | 0 | 231 |
| Cluster 3 | 0 | 594 | 1976 | 1471 | 0 | 0 | 0 | 0 |
| Cluster 4 | 0 | 377 | 1480 | 1891 | 0 | 0 | 0 | 231 |

**MoE during training.** We further provide figures that illustrate the feature learning and center learning process of each expert $\mathbf{W}_m = [\mathbf{w}_{m,1}, \ldots, \mathbf{w}_{m,J}]$ and the router $\mathbf{\Theta} = [\boldsymbol{\theta}_1, \ldots, \boldsymbol{\theta}_M]$, with $J$ as the number of filters/neurons and $M$ as the number of experts. We observe the feature learning process (change of $\max_j \langle \mathbf{w}_{m,j}, \mathbf{v}_k \rangle$) and center learning process (change of $\max_j \langle \mathbf{w}_{m,j}, \mathbf{c}_k \rangle$) of each expert $\mathbf{w}_m$ for each feature signal $\mathbf{v}_k$ and center signal $\mathbf{c}_k$. Similarly, for the weight of the router $\boldsymbol{\theta}_m$, we observe the the feature learning process (change of $\langle \boldsymbol{\theta}_m, \mathbf{v}_k \rangle$) and center learning process (change of $\langle \boldsymbol{\theta}_m, \mathbf{c}_k \rangle$) for each feature signal $\mathbf{v}_k$ and center signal $\mathbf{c}_k$. In Figure 5, we demonstrate the training process of MoE (nonlinear), and in Figure 6, we demonstrate the training process of MoE (linear). Each colored line denotes a value of $k$. The data is the same as setting 1 in Table 1, with $\alpha \in (0.5, 2)$, $\beta \in (1, 2)$, $\gamma \in (0.5, 3)$ and $\sigma_p = 1$. We can observe that, in the top left sub-figure of Figure 5 for MoE (nonlinear), feature learning ($\max_j \langle \mathbf{w}_{m,j}, \mathbf{v}_k \rangle$) exhibit a property that each expert picks up one feature signal quickly. Similarly, as shown in the bottom right sub-figure, the router picks up the corresponding center signal. Meanwhile, the nonlinear experts almost do not learn center signals and the magnitude of the inner products between router weight and feature signals remain small. However, for MoE (linear), as shown in the top two sub-figures of Figure 6, an expert does not learn a specific feature signal, but instead learns multiple feature and center signals. Moreover, as demonstrated in the bottom sub-figures of Figure 6, the magnitude of the inner products between router weight and feature signals can be even larger than the inner products between router weight and center signals.

**Verification of Theorem 4.1.** In Table 7, we provide the performances of single models with different activation functions under setting 3, where $\alpha, \gamma \in (1, 2)$ follow the same distribution. In Table 8, we further report the performances of single models with different activation functions under setting 1 and setting 2. Empirically, even when $\alpha$ and $\gamma$ do not share the same distribution, single models still fail. Note that, for Tables 7 and 8, the numbers of filters for single models are 128.

**Load balancing loss.** In Table 9, we present the results of linear MoE with load balancing loss and directly compare it with nonlinear MoE without load balancing loss. Load balancing loss guarantees that the experts receive similar amount of data and prevents MoE from activating only one or few experts. However, on the data distribution that we study, load balancing loss is not the key to the success of MoE: the single experts cannot perform well on the entire data distribution and must diverge to learn different labeling functions with respect to each cluster.

**Initialization and Expert Divergence.** In Table 10, we consider nonlinear MoE with load balancing loss and the same initialization for all the experts. The synthetic data used in this experiments is the same as setting 1 in Table 3. Recall that the data distribution we study cannot be learned by any single model: experts must diverge to learn different labeling functions. We observe from Table 10

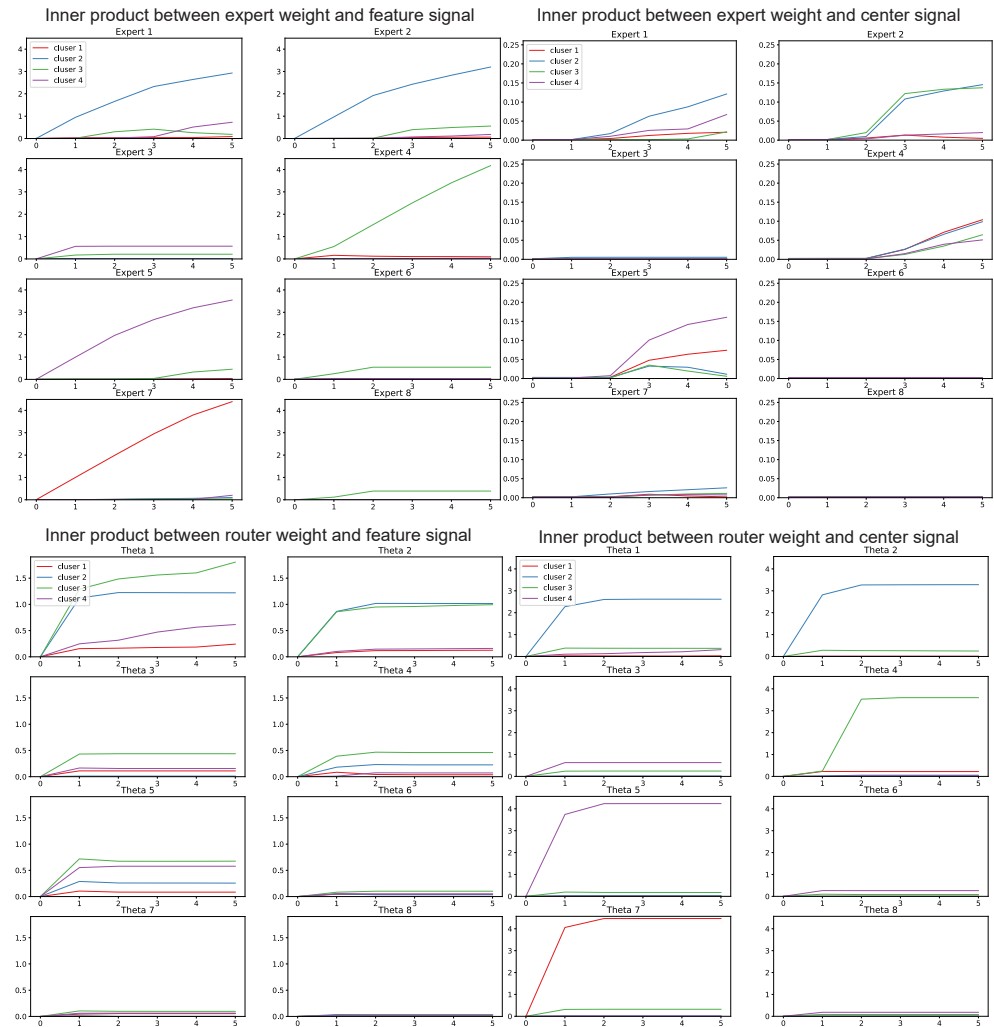

**Figure 5: Mixture of nonlinear experts.** Upper: visualization of the feature learning ($\max_j \langle \mathbf{w}_{m,j}, \mathbf{v}_k \rangle$) and center learning ($\max_j \langle \mathbf{w}_{m,j}, \mathbf{c}_k \rangle$) of each expert $\mathbf{w}_m$ for each feature $\mathbf{v}_k$ and cluster signal $\mathbf{c}_k$. Lower: visualization of the feature learning ($\langle \boldsymbol{\theta}_m, \mathbf{v}_k \rangle$) and center learning ($\langle \boldsymbol{\theta}_m, \mathbf{c}_k \rangle$) of the router weight $\boldsymbol{\theta}_m$ for each feature signal $\mathbf{v}_k$ and cluster signal $\mathbf{c}_k$.

**Table 7: Verification of Theorem 4.1 (single expert performs poorly).** Test accuracy of single linear/nonlinear models with different activation functions. Data is generated according to Definition 3.1 with $\alpha, \gamma \in (1, 2)$, $\beta \in (1, 2)$ and $\sigma_p = 1$.

| Activation | Optimal Accuracy (%) | Test Accuracy (%) |
|---|---|---|
| Linear | 87.50% | 74.81% |
| Cubic | 87.50% | 72.69% |
| Relu | 87.50% | 73.45% |
| Celu | 87.50% | 76.91% |
| Gelu | 87.50% | 74.01% |
| Tanh | 87.50% | 74.76% |

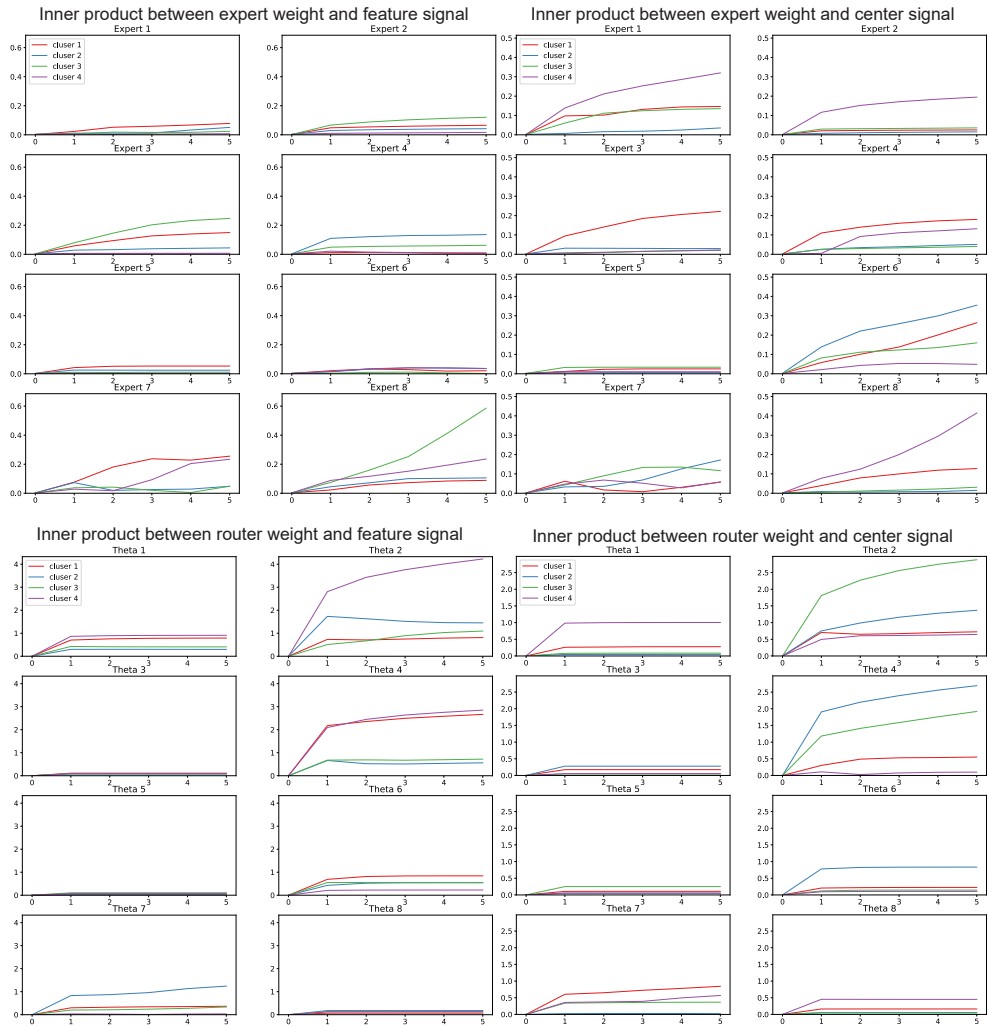

**Figure 6: Mixture of linear experts.** Upper: visualization of the feature learning $(\max_j \langle \mathbf{w}_{m,j}, \mathbf{v}_k \rangle)$ and center learning $(\max_j \langle \mathbf{w}_{m,j}, \mathbf{c}_k \rangle)$ of each expert $\mathbf{w}_m$ for each feature $\mathbf{v}_k$ and cluster signal $\mathbf{c}_k$. Lower: visualization of the feature learning $(\langle \boldsymbol{\theta}_m, \mathbf{v}_k \rangle)$ and center learning $(\langle \boldsymbol{\theta}_m, \mathbf{c}_k \rangle)$ of the router weight $\boldsymbol{\theta}_m$ for each feature signal $\mathbf{v}_k$ and cluster signal $\mathbf{c}_k$.

**Table 8: Single expert performs poorly (setting 1&2).** Test accuracy of single linear/nonlinear models with different activation functions. Data is generated according to Definition 3.1 with $\alpha \in (0.5, 2)$, $\beta \in (1, 2)$, $\gamma \in (0.5, 3)$, $\sigma_p = 1$ for setting 1. And we have $\alpha \in (0.5, 2)$, $\beta \in (1, 2)$, $\gamma \in (0.5, 3)$, $\sigma_p = 1$ for setting 2.

| Activation | Setting 1 | Setting 2 |
|------------|-----------|-----------|
| Linear | 68.71% | 60.59% |
| Cubic | 79.48% | 72.29% |
| Relu | 72.28% | 80.12% |
| Celu | 81.75% | 78.99% |
| Gelu | 79.04% | 82.01% |
| Tanh | 81.72% | 81.03% |

**Table 9: Load balancing loss.** We report the results for linear MoE with load balancing loss and compare them with our previous results on nonlinear MoE without load balancing loss. Over ten random experiments, we report the average test accuracy (%) $\pm$ standard deviation. Setting 1-4 follows the data distribution introduced above.

|  | Linear MoE with Load Balancing | Nonlinear MoE without Load Balancing |
|---|---|---|
| Setting 1 | $93.81 \pm 1.02$ | $\mathbf{99.46 \pm 0.55}$ |
| Setting 2 | $89.20 \pm 2.20$ | $\mathbf{98.09 \pm 1.27}$ |
| Setting 3 | $95.12 \pm 0.58$ | $\mathbf{99.99 \pm 0.02}$ |
| Setting 4 | $92.50 \pm 1.55$ | $\mathbf{98.92 \pm 1.18}$ |

**Table 10:** Nonlinear MoE with the **same initialization** and load balancing loss. The synthetic data is from setting 1. We report the average value $\pm$ standard deviation over 10 runs for both test accuracy and dispatch entropy.

|  | Number of experts $M = 8$ | | Number of experts $M = 32$ | |
|---|---|---|---|---|
| Load Balancing Coeff | Accuracy (%) | Dispatch Entropy | Accuracy (%) | Dispatch Entropy |
| 0.1 | $72.18 \pm 1.16$ | $1.358 \pm 0.010$ | $70.88 \pm 0.60$ | $1.381 \pm 0.002$ |
| 0.03 | $79.97 \pm 1.61$ | $1.237 \pm 0.041$ | $77.02 \pm 0.51$ | $1.252 \pm 0.010$ |
| 0.01 | $78.59 \pm 2.19$ | $1.252 \pm 0.048$ | $79.15 \pm 0.87$ | $1.221 \pm 0.014$ |

that, with the same initialization, nonlinear MoEs exhibit performances that are very similar to a single nonlinear expert.

**Load Balancing Loss and Normalized Gradient Descent.** In Table 11, we report the average test accuracy for nonlinear MoE with regard to using load balancing loss and/or normalized gradient descent. The synthetic data is the same as in setting 1 and we choose number of experts $M = 32$. All the experts are randomly initialized. We observe that using normalized gradient descent or load balancing loss (or both) can lead to successful learning of the data distribution. However, without using normalized gradient or load balancing loss will result in failure of learning the data distribution.

### A.3   Experiments on Image Data

**Datasets.** We consider CIFAR-10 (Krizhevsky, 2009) with the 10-class classification task, which contains $50,000$ training examples and $10,000$ testing examples. For CIFAR-10-Rotate, we design a binary classification task by copying and rotating all images by 30 degree and let the model predict if an image is rotated. In Figure 7, we demonstrate the positive and negative examples of CIFAR-10-Rotate. Specifically, we crop the rotated images to $(24, 24)$, and resize to $(32, 32)$ for model architectures that are designed on image size $(32, 32)$. And we further apply random Gaussian noise to all images to avoid the models taking advantage of image resolutions.

**Models.** For the simple CNN model, we consider CNN with 2 convolutional layers, both with kernel size 3 and ReLU activation followed by max pooling with size 2 and a fully connected layer. The number of filters of each convolutional layer is respectively 64, 128.

**CIFAR-10 Setup.** For real-data experiments on CIFAR-10, we apply the commonly used transforms on CIFAR-10 before each forward pass: random horizontal flips and random crops (padding the

**Table 11: Ablation study of normalized gradient descent and load balancing loss.** We report the average test accuracy $\pm$ standard deviation over 10 runs for nonlinear MoE. We consider the following four configurations: 1. normalized GD with load balancing; 2. GD with load balancing; 3. normalized GD without load balancing; 4 GD without load balancing.

| Number of experts $M = 32$ | | |
|---|---|---|
|  | Normalized GD | GD |
| With Load Balancing | $99.01 \pm 0.97$ | $98.64 \pm 0.34$ |
| Without Load Balancing | $99.47 \pm 0.48$ | $79.53 \pm 1.41$ |

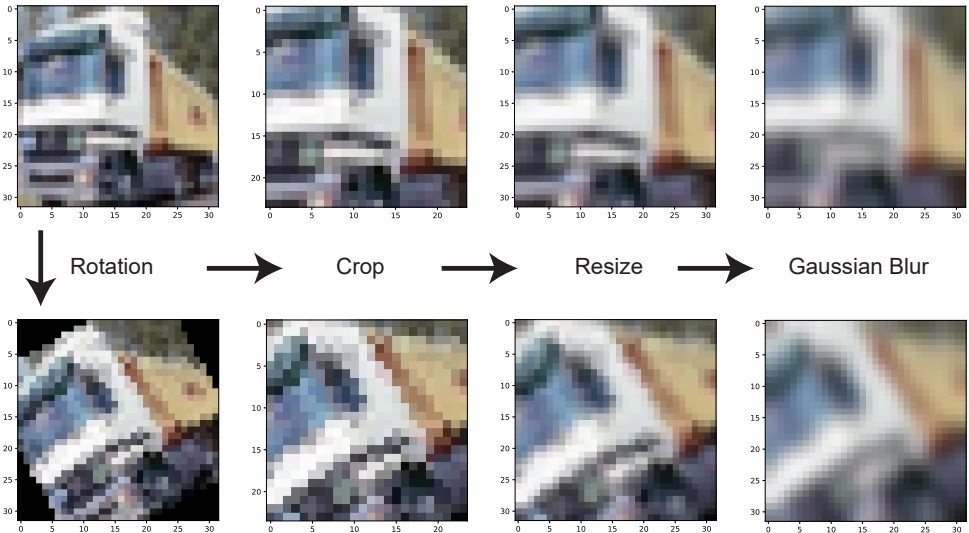

**Figure 7: Examples of the CIFAR-10-Rotate dataset.** Both the original image and the rotated image are processed in the same way, where we crop the image to $(24, 24)$, resize to $(32, 32)$ and apply random Gaussian blur.

**Table 12:** The test accuracy of the single classifier vs. MoE classifier.

|  | Single | MoE |
|---|---|---|
| Accuracy | 74.13% | 76.22% |

images on all sides with 4 pixels and randomly cropping to $(32, 32)$). And as conventionally, we normalize the data by channel. We train the single CNN model by SGD with learning rate $0.01$, momentum $0.9$ and weight decay 5e-4. And we train single MobileNetV2 and single ResNet18 by SGD with learning rate $0.1$, momentum $0.9$ and weight decay 5e-4 to achieve the best performances. We train MoEs according to Algorithm 1. Specifically, for MoE (ResNet18) and MoE (MobileNetV2), we use normalized gradient descent with learning rate $0.1$ and SGD with learning rate 1e-4, both with momentum $0.9$ and weight decay of 5e-4. For MoE (CNN), we use normalized gradient descent with learning rate $0.01$ and SGD with learning rate 1e-4, both with momentum $0.9$ and weight decay of 5e-4. We consider top-1 gating with noise and load balancing loss for MoE, where the multiplicative coefficient of load balancing loss is set at 1e-3. All models are trained for 200 epochs to achieve convergence.

**CIFAR-10-Rotate Setup.** For experiments on CIFAR10-Rotate, the data is normalized by channel as the same as in CIFAR-10 before each forward pass. We train the single CNN, single MobileNetV2 and single ResNet18 by SGD with learning rate $0.01$, momentum $0.9$ and weight decay 5e-4 to achieve the best performances. And we train MoEs by Algorithm 1 with normalized gradient descent learning rate $0.01$ on the experts and with SGD of learning rate 1e-4 on the gating networks, both with momentum $0.9$ and weight decay of 5e-4. We consider top-1 gating with noise and load balancing loss for MoE, where the multiplicative coefficient for load balancing loss is set at 1e-3. All models are trained for 50 epochs to achieve convergence.

### A.4 Experiments on Language Data

Here we provide a simple example of how MoE would work for multilingual tasks. We gather multilingual sentiment analysis data from the source of English (Sentiment140 (Go et al., 2009)) which is randomly sub-sampled to $200,000$ examples, Russian (RuReviews (Smetanin and Komarov, 2019)) which contains $90,000$ examples, and French (Blard, 2020) which contains $200,000$ examples. We randomly split the dataset into $80\%$ training data and $20\%$ test data. We use a pre-trained BERT multilingual base model (Devlin et al., 2018) to generate text embedding for each text. For the single model, we train an 1-layer neural network with cubic activation. For MoE, we let $M = 4$ with each

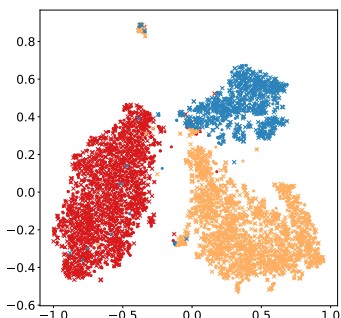

**Figure 8:** The distribution of text embedding of the multilingual sentiment analysis dataset. The embedding is generated by the pre-trained BERT multilingual base model and visualized on 2d space using t-SNE. Each color denotes a linguistic source, including English, French, and Russian.

**Table 13:** The final router dispatch details with regard to the linguistic source of the test data.

|         | Expert 1    | Expert 2   | Expert 3    | Expert 4    |
|---------|-------------|------------|-------------|-------------|
| English | $1,374$     | $3,745$    | $2,999$     | $\mathbf{31,882}$ |
| French  | $\mathbf{23,470}$ | $3,335$ | $\mathbf{13,182}$ | $13$ |
| Russian | $833$       | $\mathbf{9,405}$ | $7,723$ | $39$ |

expert sharing the same architecture as the single model. In Figure 8, we show the visualization of the text embeddings in the 2d space via t-SNE, where each color denotes a linguistic source. Here, $\cdot$ represents a positive example and $\times$ represents a negative example. We can observe that data from different linguistic sources naturally form different clusters.

In Table 12, we demonstrate the test accuracy of the single classifier and MoE on the multilingual sentiment analysis dataset, where we can observe an performance improvement of MoE over single model. And in Table 13, we show the final router dispatch details of MoE to each expert with regard to the language of the text. Notably, MoE learned to distribute examples largely according to the language.

# B  Proof of Theorem 4.1

Because we are using CNNs as experts, different ordering of the patches won't affect the value of $F(\mathbf{x})$. So for $(\mathbf{x}, y)$ drawn from $\mathcal{D}$ in Definition 3.1, we can assume that the first patch $\mathbf{x}^{(1)}$ is feature signal, the second patch $\mathbf{x}^{(2)}$ is cluster-center signal, the third patch $\mathbf{x}^{(3)}$ is feature noise. The other patches $\mathbf{x}^{(p)}, p \geq 4$ are random noises. Therefore, we can rewrite $\mathbf{x} = [\alpha y\mathbf{v}_k, \beta\mathbf{c}_k, \gamma\epsilon\mathbf{v}_{k'}, \boldsymbol{\xi}]$, where $\boldsymbol{\xi} = [\boldsymbol{\xi}_4, \ldots, \boldsymbol{\xi}_P]$ is a Gaussian matrix of size $\mathbb{R}^{d \times (P-3)}$.

*Proof of Theorem 4.1.* Conditioned on the event that $y = -\epsilon$, points $([\alpha y\mathbf{v}_k, \beta\mathbf{c}_k, -\gamma y\mathbf{v}_{k'}, \boldsymbol{\xi}], y)$, $([-\alpha y\mathbf{v}_k, \beta\mathbf{c}_k, \gamma y\mathbf{v}_{k'}, \boldsymbol{\xi}], -y)$, $([\gamma y\mathbf{v}_{k'}, \beta\mathbf{c}_{k'}, -\alpha y\mathbf{v}_k, \boldsymbol{\xi}], y)$, $([-\gamma y\mathbf{v}_{k'}, \beta\mathbf{c}_{k'}, \alpha y\mathbf{v}_k, \boldsymbol{\xi}], -y)$ follow the same distribution because $\gamma$ and $\alpha$ follow the same distribution, and $y$ and $-y$ follow the same distribution. Therefore, we have

$$4\mathbb{P}\big(yF(\mathbf{x}) \leq 0 | \epsilon = -y\big)$$

$$= \mathbb{E}\bigg[\underbrace{\mathbb{1}\big(yF([\alpha y\mathbf{v}_k, \beta\mathbf{c}_k, -\gamma y\mathbf{v}_{k'}, \boldsymbol{\xi}]) \leq 0\big)}_{I_1} + \underbrace{\mathbb{1}\big(-yF([-\alpha y\mathbf{v}_k, \beta\mathbf{c}_k, \gamma y\mathbf{v}_{k'}, \boldsymbol{\xi}]) \leq 0\big)}_{I_2}$$

$$+ \underbrace{\mathbb{1}\big(yF([\gamma y\mathbf{v}_{k'}, \beta\mathbf{c}_{k'}, -\alpha y\mathbf{v}_k, \boldsymbol{\xi}]) \leq 0\big)}_{I_3} + \underbrace{\mathbb{1}\big(-yF([-\gamma y\mathbf{v}_{k'}, \beta\mathbf{c}_{k'}, \alpha y\mathbf{v}_k, \boldsymbol{\xi}]) \leq 0\big)}_{I_4}\bigg].$$

It is easy to verify the following fact

$$\Big(yF([\alpha y\mathbf{v}_k, \beta\mathbf{c}_k, -\gamma y\mathbf{v}_{k'}, \boldsymbol{\xi}])\Big) + \Big(-yF([-\alpha y\mathbf{v}_k, \beta\mathbf{c}_k, \gamma y\mathbf{v}_{k'}, \boldsymbol{\xi}])\Big)$$

$$+ \Big(yF([\gamma y\mathbf{v}_{k'}, \beta\mathbf{c}_{k'}, -\alpha y\mathbf{v}_k, \boldsymbol{\xi}])\Big) + \Big(-yF([-\gamma y\mathbf{v}_{k'}, \beta\mathbf{c}_{k'}, \alpha y\mathbf{v}_k, \boldsymbol{\xi}])\Big)$$

$$= \Big(yf(\alpha y\mathbf{v}_k) + yf(\beta\mathbf{c}_k) + yf(-\gamma y\mathbf{v}_{k'}) + \sum_{p=4}^{P} yf(\boldsymbol{\xi}_p)\Big)$$

$$+ \Big(-yf(-\alpha y\mathbf{v}_k) - yf(\beta\mathbf{c}_k) - yf(\gamma y\mathbf{v}_{k'}) - \sum_{p=4}^{P} yf(\boldsymbol{\xi}_p)\Big)$$

$$+ \Big(yf(\gamma y\mathbf{v}_{k'}) + yf(\beta\mathbf{c}_{k'}) + yf(-\alpha y\mathbf{v}_k) + \sum_{p=4}^{P} yf(\boldsymbol{\xi}_p)\Big)$$

$$+ \Big(-yf(-\gamma y\mathbf{v}_{k'}) - yf(\beta\mathbf{c}_{k'}) - yf(\alpha y\mathbf{v}_k) - \sum_{p=4}^{P} yf(\boldsymbol{\xi}_p)\Big)$$

$$= 0.$$

By pigeonhole principle, at least one of $I_1, I_2, I_3, I_4$ is non-zero. This further implies that $4\mathbb{P}\big(yF(\mathbf{x}) \leq 0|\epsilon = -y\big) \geq 1$. Applying $\mathbb{P}(\epsilon = -y) = 1/2$, we have that

$$\mathbb{P}\big(yF(\mathbf{x}) \leq 0\big) \geq \mathbb{P}\big(yF(\mathbf{x}) \leq 0)|\epsilon = -y\big)\mathbb{P}(\epsilon = -y) \geq 1/8,$$

which completes the proof. $\qquad\square$

## C Smoothed Router

In this section, we will show that the noise term provides a smooth transition between different routing behavior. All the results in this section is independent from our NN structure and its initialization. We first present a general version of Lemma 5.1 with its proof.

**Lemma C.1** (Extension of Lemma 5.1). Let $\mathbf{h}, \widehat{\mathbf{h}} \in \mathbb{R}^M$ to be the output of the gating network and $\{r_m\}_{m=1}^{M}$ to be the noise independently drawn from $\mathcal{D}_r$. Denote $\mathbf{p}, \widehat{\mathbf{p}} \in \mathbb{R}^M$ to be the probability that experts get routed, i.e., $p_m = \mathbb{P}(\mathrm{argmax}_{m'\in[M]}\{h_{m'} + r_{m'}\} = m)$, $\widehat{p}_m = \mathbb{P}(\mathrm{argmax}_{m'\in[M]}\{\widehat{h}_{m'} + r_{m'}\} = m)$. Suppose the probability density function of $\mathcal{D}_r$ is bounded by $\kappa$, Then we have that $\|\mathbf{p} - \widehat{\mathbf{p}}\|_\infty \leq (\kappa M^2) \cdot \|\mathbf{h} - \widehat{\mathbf{h}}\|_\infty$.

*Proof.* Given random variable $\{r_m\}_{m=1}^{M}$, let us first consider the event that $\mathrm{argmax}_m\{h_m + r_m\} \neq \mathrm{argmax}_m\{\widehat{h}_m + r_m\}$. Let $m_1 = \mathrm{argmax}_m\{h_m + r_m\}$ and $m_2 = \mathrm{argmax}_m\{\widehat{h}_m + r_m\}$, then we have that

$$h_{m_1} + r_{m_1} \geq h_{m_2} + r_{m_2}, \widehat{h}_{m_2} + r_{m_2} \geq \widehat{h}_{m_1} + r_{m_1},$$

which implies that

$$\widehat{h}_{m_2} - \widehat{h}_{m_1} \geq r_{m_1} - r_{m_2} \geq h_{m_2} - h_{m_1}. \tag{C.1}$$

Define $C(m_1, m_2) = (\widehat{h}_{m_2} - \widehat{h}_{m_1} + h_{m_2} - h_{m_1})/2$, then (C.1) implies that

$$|r_{m_1} - r_{m_2} - C(m_1, m_2)| \leq |\widehat{h}_{m_2} - \widehat{h}_{m_1} - h_{m_2} + h_{m_1}|/2 \leq \|\widehat{\mathbf{h}} - \mathbf{h}\|_\infty. \tag{C.2}$$

Therefore, we have that,

$$\mathbb{P}(\mathrm{argmax}_m\{h_m + r_m\} \neq \mathrm{argmax}_m\{\widehat{h}_m + r_m\})$$

$$\leq \mathbb{P}(\exists m_1 \neq m_2 \in [M], \text{ s.t. } |r_{m_1} - r_{m_2} - C(m_1, m_2)| \leq \|\widehat{\mathbf{h}} - \mathbf{h}\|_\infty)$$

$$\leq \sum_{m_1 < m_2} \mathbb{P}\big(|r_{m_1} - r_{m_2} - C(m_1, m_2)| \leq \|\widehat{\mathbf{h}} - \mathbf{h}\|_\infty\big)$$

$$= \sum_{m_1 < m_2} \mathbb{E}\Big[\mathbb{P}\big(r_{m_2} + C(m_1, m_2) - \|\widehat{\mathbf{h}} - \mathbf{h}\|_\infty \leq r_{m_1} \leq r_{m_2} + C(m_1, m_2) + \|\widehat{\mathbf{h}} - \mathbf{h}\|_\infty\big)\Big|r_{m_2}\Big]$$

$$\leq (\kappa M^2) \cdot \|\widehat{\mathbf{h}} - \mathbf{h}\|_\infty,$$

where the first inequality is by (C.2), the second inequality is by union bound and the last inequality is due to the fact that the probability density function of $r_{m_1}$ is bounded by $\kappa$. Then we have that for $i \in [M]$,

$$
\begin{aligned}
|p_i - \widehat{p}_i| &\leq \left| \mathbb{E}\left[ \mathbb{1}\left( \operatorname*{argmax}_m \{\widehat{h}_m + r_m\} = i \right) - \mathbb{1}\left( \operatorname*{argmax}_m \{h_m + r_m\} = i \right) \right] \right| \\
&\leq \mathbb{E}\left| \mathbb{1}\left( \operatorname*{argmax}_m \{\widehat{h}_m + r_m\} = i \right) - \mathbb{1}\left( \operatorname*{argmax}_m \{h_m + r_m\} = i \right) \right| \\
&\leq \mathbb{P}\left( \operatorname*{argmax}_m \{\widehat{h}_m + r_m\} \neq \operatorname*{argmax}_m \{h_m + r_m\} \right) \\
&\leq (\kappa M^2) \cdot \|\widehat{\mathbf{h}} - \mathbf{h}\|_\infty,
\end{aligned}
$$

which completes the proof. $\qquad\square$

**Remark C.2.** A widely used choice of $\mathcal{D}_r$ in Lemma C.1 is uniform noise Unif[a, b], in which case the density function can be upper bounded by $1/(b-a)$. Another widely used choice of $\mathcal{D}_r$ is Gaussian noise $\mathcal{N}(0, \sigma_r^2)$, in which case the density function can be upper bounded by $1/(\sigma_r \sqrt{2\pi})$. Increase the range of uniform noise or increase the variance of the Gaussian noise will result in a smaller density function upper bound and a smoother behavior of routing. In our paper, we consider unif[0,1] for simplicity, in which case the the density function can be upper bounded by 1 ($\kappa = 1$).

The following Lemma shows that when two gate network outputs are close, the router will distribute the examples to those corresponding experts with nearly the same probability.

**Lemma C.3.** Let $\mathbf{h} \in \mathbb{R}^M$ be the output of the gating network and $\{r_m\}_{m=1}^M$ be the noise independently drawn from Unif[0,1]. Denote the probability that experts get routed by $\mathbf{p}$, i.e., $p_m = \mathbb{P}(\operatorname{argmax}_{m'}\{h_{m'} + r_{m'}\} = m)$. Then we have that

$$|p_m - p_{m'}| \leq M^2 |h_m - h_{m'}|.$$

*Proof.* Construct $\widehat{\mathbf{h}}$ as copy of $\mathbf{h}$ and permute its $m, m'$-th element. Denote the corresponding probability vector as $\widehat{\mathbf{p}}$. Then it is obviously that $|p_m - p_{m'}| = \|\mathbf{p} - \widehat{\mathbf{p}}\|_\infty$ and $|h_m - h_{m'}| = \|\widehat{\mathbf{h}} - \mathbf{h}\|_\infty$. Applying Lemma 5.1 completes the proof. $\qquad\square$

The following lemma shows that the router won't route examples to the experts with small gating network outputs, which saves computation and improves the performance.

**Lemma C.4.** Suppose the noise $\{r_m\}_{m=1}^M$ are independently drawn from Unif[0,1] and $h_m(\mathbf{x}; \boldsymbol{\Theta}) \leq \max_{m'} h_{m'}(\mathbf{x}; \boldsymbol{\Theta}) - 1$, example $\mathbf{x}$ will not get routed to expert $m$.

*Proof.* Because $h_m(\mathbf{x}; \boldsymbol{\Theta}) \leq \max_{m'} h_{m'}(\mathbf{x}; \boldsymbol{\Theta}) - 1$ implies that for any Uniform noise $\{r_{m'}\}_{m' \in [M]}$ we have that

$$h_m(\mathbf{x}; \boldsymbol{\Theta}) + r_m \leq \max_{m'} h_{m'}(\mathbf{x}; \boldsymbol{\Theta}) \leq \max_{m'}\{h_{m'}(\mathbf{x}; \boldsymbol{\Theta}) + r_{m'}\},$$

where the first inequality is by $r_m \leq 1$, the second inequality is by $r_{m'} \geq 0, \forall m' \in [M]$. $\qquad\square$

# D  Initialization of the Model

Before we look into the detailed proof of Theorem 4.2, let us first discuss some basic properties of the data distribution and our MoE model. For simplicity of notation, we simplify $(\mathbf{x}_i, y_i) \in \Omega_k$ as $i \in \Omega_k$.

**Training Data Set Property.** Because we are using CNNs as experts, different ordering of the patches won't affect the value of $F(\mathbf{x})$. So for $(\mathbf{x}, y)$ drawn from $\mathcal{D}$ in Definition 3.1, we can assume that the first patch $\mathbf{x}^{(1)}$ is feature signal, the second patch $\mathbf{x}^{(2)}$ is cluster-center signal, the third patch $\mathbf{x}^{(3)}$ is feature noise. The other patches $\mathbf{x}^{(p)}, p \geq 4$ are random noises. Therefore, we can rewrite $\mathbf{x} = [\alpha y \mathbf{v}_k, \beta \mathbf{c}_k, \gamma \epsilon \mathbf{v}_{k'}, \boldsymbol{\xi}]$, where $\boldsymbol{\xi} = [\boldsymbol{\xi}_4, \dots, \boldsymbol{\xi}_P]$ is a Gaussian matrix of size $\mathbb{R}^{d \times (P-3)}$. According to the type of the feature noise, we further divide $\Omega_k$ into $\Omega_k = \cup \Omega_{k,k'}$ based on the feature noise, i.e. $\mathbf{x} \in \Omega_{k,k'}$ if $\mathbf{x} = [\alpha y \mathbf{v}_k, \beta \mathbf{c}_k, \gamma \epsilon \mathbf{v}_{k'}, \boldsymbol{\xi}]$. To better characterize the router training,

we need to break down $\Omega_{k,k'}$ into $\Omega_{k,k'}^+$ and $\Omega_{k,k'}^-$. Denote by $\Omega_{k,k'}^+$ the set that $\{y_i = \epsilon_i | i \in \Omega_{k,k'}\}$, by $\Omega_{k,k'}^-$ the set that $\{y_i = -\epsilon_i | i \in \Omega_{k,k'}\}$.

**Lemma D.1.** With probability at least $1 - \delta$, the following properties hold for all $k \in [K]$,

$$\sum_{i \in \Omega_k} y_i \beta_i^3 = \widetilde{O}(\sqrt{n}), \sum_{i \in \Omega_k} \alpha_i^3 = \mathbb{E}[\alpha^3] \cdot n/K + \widetilde{O}(\sqrt{n}), \sum_{i \in \Omega_k} y_i \epsilon_i \gamma_i^3 = \widetilde{O}(\sqrt{n}), \quad \text{(D.1)}$$

$$\sum_{i \in \Omega_{k,k'}^+} y_i \alpha_i = \widetilde{O}(\sqrt{n}), \sum_{i \in \Omega_{k,k'}^-} y_i \alpha_i = \widetilde{O}(\sqrt{n}), \sum_{i \in \Omega_{k,k'}^+} \epsilon_i \gamma_i = \widetilde{O}(\sqrt{n}), \quad \text{(D.2)}$$

$$\sum_{i \in \Omega_{k,k'}^-} \epsilon_i \gamma_i = \widetilde{O}(\sqrt{n}), \sum_{i \in \Omega_k} \beta_i = \mathbb{E}[\beta] \cdot n/K + \widetilde{O}(\sqrt{n}). \quad \text{(D.3)}$$

*Proof.* Fix $k \in [K]$, by Hoeffding's inequality we have that with probability at least $1 - \delta/8K$,

$$\sum_{i \in \Omega_k} y_i \beta_i^3 = \sum_{i=1}^n y_i \beta_i^3 \mathbb{1}\left((\mathbf{x}_i, y_i) \in \Omega_k\right) = \widetilde{O}(\sqrt{n}),$$

where the last equality is by the fact that the expectation of $y \beta^3 \mathbb{1}\left((\mathbf{x}, y) \in \Omega_k\right)$ is zero. Fix $k \in [K]$, by Hoeffding's inequality we have that with probability at least $1 - \delta/8K$,

$$\sum_{i \in \Omega_k} \alpha_i^3 = \sum_{i=1}^n \alpha_i^3 \mathbb{1}\left((\mathbf{x}_i, y_i) \in \Omega_k\right) = \frac{n\mathbb{E}[\alpha^3]}{K} + \widetilde{O}(\sqrt{n}),$$

where the last equality is by the fact that the expectation of $\alpha^3 \mathbb{1}\left((\mathbf{x}, y) \in \Omega_k\right)$ is $\mathbb{E}[\alpha^3]/K$. Fix $k \in [K]$, by Hoeffding's inequality we have that with probability at least $1 - \delta/8K$,

$$\sum_{i \in \Omega_k} y_i \epsilon_i \gamma_i^3 = \sum_{i=1}^n y_i \epsilon_i \gamma_i^3 \mathbb{1}\left((\mathbf{x}_i, y_i) \in \Omega_k\right) = \widetilde{O}(\sqrt{n}),$$

where the last equality is by the fact that the expectation of $y\epsilon \gamma^3 \mathbb{1}\left((\mathbf{x}, y) \in \Omega_k\right)$ is zero. Now we have proved the bounds in (D.1). We can get other bounds in (D.2) and (D.3) similarly. Applying union bound over $[K]$ completes the proof. $\qquad\square$

**Lemma D.2.** Suppose that $d = \Omega(\log(4nP/\delta))$, with probability at least $1 - \delta$, the following inequalities hold for all $i \in [n], k \in [K], p \geq 4$,

- $\|\boldsymbol{\xi}_{i,p}\|_2 = O(1)$,
- $\langle \mathbf{v}_k, \boldsymbol{\xi}_{i,p} \rangle \leq \widetilde{O}(d^{-1/2})$, $\langle \mathbf{c}_k, \boldsymbol{\xi}_{i,p} \rangle \leq \widetilde{O}(d^{-1/2})$, $\langle \boldsymbol{\xi}_{i,p}, \boldsymbol{\xi}_{i',p'} \rangle \leq \widetilde{O}(d^{-1/2})$, $\forall (i', p') \neq (i, p)$.

*Proof of Lemma D.2.* By Bernstein's inequality, with probability at least $1 - \delta/(2nP)$ we have

$$\left| \|\boldsymbol{\xi}_{i,p}\|_2^2 - \sigma_p^2 \right| \leq O(\sigma_p^2 \sqrt{d^{-1} \log(4nP/\delta)}).$$

Therefore, as long as $d = \Omega(\log(4nP/\delta))$, we have $\|\boldsymbol{\xi}_{i,p}\|_2^2 \leq 2$. Moreover, clearly $\langle \boldsymbol{\xi}_{i,p}, \boldsymbol{\xi}_{i',p'} \rangle$ has mean zero, $\forall (i, p) \neq (i', p')$. Then by Bernstein's inequality, with probability at least $1 - \delta/(6n^2P^2)$ we have

$$|\langle \boldsymbol{\xi}_{i,p}, \boldsymbol{\xi}_{i',p'} \rangle| \leq 2\sigma_p^2 \sqrt{d^{-1} \log(12n^2P^2/\delta)}.$$

Similarly, $\langle \mathbf{v}_k, \boldsymbol{\xi}_{i,p} \rangle$ and $\langle \mathbf{c}_k, \boldsymbol{\xi}_{i,p} \rangle$ have mean zero. Then by Bernstein's inequality, with probability at least $1 - \delta/(3nPK)$ we have

$$|\langle \boldsymbol{\xi}_{i,p}, \mathbf{v}_k \rangle| \leq 2\sigma_p \sqrt{d^{-1} \log(6nPK/\delta)}, |\langle \boldsymbol{\xi}_{i,p}, \mathbf{c}_k \rangle| \leq 2\sigma_p \sqrt{d^{-1} \log(6nPK/\delta)}.$$

Applying a union bound completes the proof. $\qquad\square$

**MoE Initialization Property.**

We divide the experts into $K$ sets based on the initialization.

**Definition D.3.** Fix expert $m \in [M]$, denote $(k_m^*, j_m^*) = \text{argmax}_{j,k} \langle \mathbf{v}_k, \mathbf{w}_{m,j}^{(0)} \rangle$. Fix cluster $k \in [K]$, denote the profession experts set as $\mathcal{M}_k = \{m | k_m^* = k\}$.

**Lemma D.4.** For $M \geq \Theta(K \log(K/\delta))$, $J \geq \Theta(\log(M/\delta))$, the following inequalities hold with probability at least $1 - \delta$.

- $\max_{(j,k) \neq (j_m^*, k_m^*)} \langle \mathbf{w}_{m,j}^{(0)}, \mathbf{v}_k \rangle \leq \left(1 - \delta/\left(3MJ^2K^2\right)\right) \langle \mathbf{w}_{m,j_m^*}^{(0)}, \mathbf{v}_{k_m^*} \rangle$ for all $m \in [M]$

- $\langle \mathbf{w}_{m,j_m^*}^{(0)}, \mathbf{v}_{k_m^*} \rangle \geq 0.01\sigma_0$ for all $m \in [M]$.

- $|\mathcal{M}_k| \geq 1$ for all $k \in [K]$.

*Proof.* Recall that $\mathbf{w}_{m,j} \sim \mathcal{N}(0, \sigma_0^2 I_d)$. Notice that signals $\mathbf{v}_1, \ldots, \mathbf{v}_K$ are orthogonal. Given fixed $m \in [M]$, we have that $\{\langle \mathbf{w}_{m,j}^{(0)}, \mathbf{v}_k \rangle | j \in [J], k \in [K]\}$ are independent and individually draw from $\mathcal{N}(0, \sigma_0^2)$ we have that

$$\mathbb{P}(\langle \mathbf{w}_{m,j}^{(0)}, \mathbf{v}_k \rangle < 0.01\sigma_0) < 0.9.$$

Therefore, we have that

$$\mathbb{P}(\max_{j,k} \langle \mathbf{w}_{m,j}^{(0)}, \mathbf{v}_k \rangle < 0.01\sigma_0) < 0.9^{KJ}.$$

Therefore, as long as $J \geq \Theta(K^{-1} \log(M/\delta))$, fix $m \in [M]$ we can guarantee that with probability at least $1 - \delta/(3M)$,

$$\max_{j,k} \langle \mathbf{w}_{m,j}^{(0)}, \mathbf{v}_k \rangle > 0.01\sigma_0.$$

Take $G = \delta/(3MJ^2K^2)$, by Lemma F.1 we have that with probability at least $1 - \delta/(3M)$,

$$\max_{(j,k) \neq (j_m^*, k_m^*)} \langle \mathbf{w}_{m,j}^{(0)}, \mathbf{v}_k \rangle \leq (1 - G) \langle \mathbf{w}_{m,j_m^*}^{(0)}, \mathbf{v}_{k_m^*} \rangle.$$

By the symmetric property, we have that for all $k \in [K], m \in [M]$,

$$\mathbb{P}(k = k_m^*) = K^{-1}.$$

Therefore, the probability that $|\mathcal{M}_k|$ at least include one element is as follows,

$$\mathbb{P}(|\mathcal{M}_k| \geq 1) \geq 1 - (1 - K^{-1})^M.$$

By union bound we get that

$$\mathbb{P}(|\mathcal{M}_k| \geq 1, \forall k) \geq 1 - K(1 - K^{-1})^M \geq 1 - K \exp(-M/K) \geq 1 - \delta/3,$$

where the last inequality is by condition $M \geq K \log(3K/\delta)$. Therefore, with probability at least $1 - \delta/3$, $|\mathcal{M}_k| \geq 1, \forall k$.

Applying Union bound, we have that with probability at least $1 - \delta$,

$$\max_{(j,k) \neq (j_m^*, k_m^*)} \langle \mathbf{w}_{m,j}^{(0)}, \mathbf{v}_k \rangle \leq \left(1 - \delta/\left(3MJ^2K^2\right)\right) \langle \mathbf{w}_{m,j_m^*}^{(0)}, \mathbf{v}_{k_m^*} \rangle,$$

$$\langle \mathbf{w}_{m,j_m^*}^{(0)}, \mathbf{v}_{k_m^*} \rangle \geq 0.01\sigma_0, \forall m \in [M],$$

$$|\mathcal{M}_k| \geq 1, \forall k \in [K].$$

$\square$

**Lemma D.5.** Suppose the conclusions in Lemma D.2 hold, then with probability at least $1 - \delta$ we have that $|\langle \mathbf{w}_{m,j}^{(0)}, \mathbf{v} \rangle| \leq \widetilde{O}(\sigma_0)$ for all $\mathbf{v} \in \{\mathbf{v}_k\}_{k \in [K]} \cup \{\mathbf{c}_k\}_{k \in [K]} \cup \{\boldsymbol{\xi}_{i,p}\}_{i \in [n], p \in [P-3]}, m \in [M], j \in [J]$.

*Proof.* Fix $\mathbf{v} \in \{\mathbf{v}_k\}_{k \in [K]} \cup \{\mathbf{c}_k\}_{k \in [K]} \cup \{\boldsymbol{\xi}_{i,p}\}_{i \in [n], p \in [P-3]}, m \in [M], j \in [J]$, we have that $\langle \mathbf{w}_{m,j}^{(0)}, \mathbf{v} \rangle \sim \mathcal{N}(0, \sigma_0^2 \|\mathbf{v}\|_2^2)$ and $\|\mathbf{v}\|_2 = O(1)$. Therefore, with probability at least $1 - \delta/(nPMJ)$ we have that $|\langle \mathbf{w}_{m,j}^{(0)}, \mathbf{v} \rangle| \leq \widetilde{O}(\sigma_0)$. Applying union bound completes the proof. $\square$

# E Proof of Theorem 4.2

In this section we always assume that the conditions in Theorem 4.2 holds. It is easy to show that all the conclusions in this section D hold with probability at least $1 - O(1/\log d)$. The results in this section hold when all the conclusions in Section D hold. For simplicity of notation, we simplify $(\mathbf{x}_i, y_i) \in \Omega_{k,k'}$ as $i \in \Omega_{k,k'}$, and $\ell'(y_i \pi_{m_{i,t}}(\mathbf{x}_i; \mathbf{\Theta}^{(t)}) f_{m_{i,t}}(\mathbf{x}_i; \mathbf{W}^{(t)}))$ as $\ell'_{i,t}$.

Recall that at iteration $t$, data $\mathbf{x}_i$ is routed to the expert $m_{i,t}$. Here $m_{i,t}$ should be interpreted as a random variable. The gradient of MoE model at iteration $t$ can thus be computed as follows

$$\nabla_{\boldsymbol{\theta}_m} \mathcal{L}^{(t)} = \frac{1}{n} \sum_{i,p} \mathbb{1}(m_{i,t} = m)\ell'_{i,t}\pi_{m_{i,t}}(\mathbf{x}_i; \mathbf{\Theta}^{(t)})(1 - \pi_{m_{i,t}}(\mathbf{x}_i; \mathbf{\Theta}^{(t)}))y_i f_{m_{i,t}}(\mathbf{x}_i; \mathbf{W}^{(t)})\mathbf{x}_i^{(p)}$$

$$- \frac{1}{n} \sum_{i,p} \mathbb{1}(m_{i,t} \neq m)\ell'_{i,t}\pi_{m_{i,t}}(\mathbf{x}_i; \mathbf{\Theta}^{(t)})\pi_m(\mathbf{x}_i; \mathbf{\Theta}^{(t)})y_i f_{m_{i,t}}(\mathbf{x}_i; \mathbf{W}^{(t)})\mathbf{x}_i^{(p)}$$

$$= \frac{1}{n} \sum_{i,p} \mathbb{1}(m_{i,t} = m)\ell'_{i,t}\pi_{m_{i,t}}(\mathbf{x}_i; \mathbf{\Theta}^{(t)})y_i f_{m_{i,t}}(\mathbf{x}_i; \mathbf{W}^{(t)})\mathbf{x}_i^{(p)}$$

$$- \frac{1}{n} \sum_{i,p} \ell'_{i,t}\pi_{m_{i,t}}(\mathbf{x}_i; \mathbf{\Theta}^{(t)})\pi_m(\mathbf{x}_i; \mathbf{\Theta}^{(t)})y_i f_{m_{i,t}}(\mathbf{x}_i; \mathbf{W}^{(t)})\mathbf{x}_i^{(p)}, \tag{E.1}$$

$$\nabla_{\mathbf{w}_{m,j}} \mathcal{L}^{(t)} = \frac{1}{n} \sum_{i,p} \mathbb{1}(m_{i,t} = m)\ell'_{i,t}\pi_m(\mathbf{x}_i; \mathbf{\Theta}^{(t)})y_i \sigma'(\langle \mathbf{w}_{m,j}^{(t)}, \mathbf{x}_i^{(p)} \rangle)\mathbf{x}_i^{(p)}. \tag{E.2}$$

Following lemma shows implicit regularity in the gating network training.

**Lemma E.1.** For all $t \geq 0$, we have that $\sum_{m=1}^{M} \nabla_{\boldsymbol{\theta}_m} \mathcal{L}^{(t)} = \mathbf{0}$ and thus $\sum_m \boldsymbol{\theta}_m^{(t)} = \sum_m \boldsymbol{\theta}_m^{(0)}$. In particular, when $\mathbf{\Theta}$ is zero initialized, then $\sum_m \boldsymbol{\theta}_m^{(t)} = 0$

*Proof.* We first write out the gradient of $\boldsymbol{\theta}_m$ for all $m \in [M]$,

$$\nabla_{\boldsymbol{\theta}_m} \mathcal{L}^{(t)} = \frac{1}{n} \sum_{i \in [n], p \in [P]} \mathbb{1}(m_{i,t} = m)\ell'_{i,t}\pi_{m_{i,t}}(\mathbf{x}_i; \mathbf{\Theta}^{(t)})y_i f_{m_{i,t}}(\mathbf{x}_i; \mathbf{W}^{(t)})\mathbf{x}_i^{(p)}$$

$$- \frac{1}{n} \sum_{i \in [n], p \in [P]} \ell'_{i,t}\pi_{m_{i,t}}(\mathbf{x}_i; \mathbf{\Theta}^{(t)})\pi_m(\mathbf{x}_i; \mathbf{\Theta}^{(t)})y_i f_{m_{i,t}}(\mathbf{x}_i; \mathbf{W}^{(t)})\mathbf{x}_i^{(p)}.$$

Take summation from $m = 1$ to $m = M$, then we have

$$\sum_{m=1}^{M} \nabla_{\boldsymbol{\theta}_m} \mathcal{L}^{(t)} = \frac{1}{n} \sum_{i \in [n], p \in [P]} \ell'_{i,t}\pi_{m_{i,t}}(\mathbf{x}_i; \mathbf{\Theta}^{(t)})y_i f_{m_{i,t}}(\mathbf{x}_i; \mathbf{W}^{(t)})\mathbf{x}_i^{(p)}$$

$$- \frac{1}{n} \sum_{i \in [n], p \in [P]} \ell'_{i,t}\pi_{m_{i,t}}(\mathbf{x}_i; \mathbf{\Theta}^{(t)})y_i f_{m_{i,t}}(\mathbf{x}_i, \mathbf{W}^{(t)})\mathbf{x}_i^{(p)}$$

$$= 0.$$

$\square$

Notice that the gradient at iteration $t$ in (E.1) and (E.2) is depend on the random variable $m_{i,t}$, the following lemma shows that it can be approximated by its expectation.

**Lemma E.2.** With probability at least $1 - 1/d$, for all the vector $\mathbf{v} \in \{\mathbf{v}_k\}_{k \in [K]} \cup \{\mathbf{c}_k\}_{k \in [K]}$, $m \in [M]$, $j \in [J]$, we have the following equations hold $|\langle \nabla_{\boldsymbol{\theta}_m} \mathcal{L}^{(t)}, \mathbf{v} \rangle - \mathbb{E}[\langle \nabla_{\boldsymbol{\theta}_m} \mathcal{L}^{(t)}, \mathbf{v} \rangle]| = \widetilde{O}(n^{-1/2}(\sigma_0 + \eta t)^3)$, $|\langle \nabla_{\mathbf{w}_{m,j}} \mathcal{L}^{(t)}, \mathbf{v} \rangle - \mathbb{E}[\langle \nabla_{\mathbf{w}_{m,j}} \mathcal{L}^{(t)}, \mathbf{v} \rangle]| = \widetilde{O}(n^{-1/2}(\sigma_0 + \eta t)^2)$, for all $t \leq d^{100}$. Here $\mathbb{E}[\langle \nabla_{\mathbf{w}_{m,j}} \mathcal{L}^{(t)}, \mathbf{v} \rangle]$ and $\mathbb{E}[\langle \nabla_{\boldsymbol{\theta}_m} \mathcal{L}^{(t)}, \mathbf{v} \rangle]$ can be computed as follows,

$$\mathbb{E}[\langle \nabla_{\boldsymbol{\theta}_m} \mathcal{L}^{(t)}, \mathbf{v} \rangle] = \frac{1}{n} \sum_{i,p} \mathbb{P}(m_{i,t} = m) \ell'_{i,t} \pi_m(\mathbf{x}_i; \boldsymbol{\Theta}^{(t)}) y_i f_m(\mathbf{x}_i; \mathbf{W}^{(t)}) \langle \mathbf{x}_i^{(p)}, \mathbf{v} \rangle$$

$$- \frac{1}{n} \sum_{i,p,m'} \mathbb{P}(m_{i,t} = m') \ell'_{i,t} \pi_{m'}(\mathbf{x}_i; \boldsymbol{\Theta}^{(t)}) \pi_m(\mathbf{x}_i; \boldsymbol{\Theta}^{(t)}) y_i f_{m'}(\mathbf{x}_i; \mathbf{W}^{(t)}) \langle \mathbf{x}_i^{(p)}, \mathbf{v} \rangle$$

$$\mathbb{E}[\langle \nabla_{\mathbf{w}_{m,j}} \mathcal{L}^{(t)}, \mathbf{v} \rangle] = \frac{1}{n} \sum_{i,p} \mathbb{P}(m_{i,t} = m) \ell'_{i,t} \pi_m(\mathbf{x}_i; \boldsymbol{\Theta}^{(t)}) y_i \sigma'(\langle \mathbf{w}_{m,j}^{(t)}, \mathbf{x}_i^{(p)} \rangle) \langle \mathbf{x}_i^{(p)}, \mathbf{v} \rangle.$$

*Proof.* Because we are using normalized gradient descent, $\|\mathbf{w}_{m,j}^{(t)} - \mathbf{w}_{m,j}^{(0)}\|_2 \leq O(\eta t)$ and thus by Lemma D.5 we have $|\langle \mathbf{w}_{m,j}^{(t)}, \mathbf{x}_i^{(p)} \rangle| \leq \widetilde{O}(\sigma_0 + \eta t)$. Therefore,

$$\langle \nabla_{\mathbf{w}_{m,j}} \mathcal{L}^{(t)}, \mathbf{v} \rangle = \frac{1}{n} \sum_i \underbrace{\sum_p \mathbb{1}(m_{i,t} = m) \ell'_{i,t} \pi_m(\mathbf{x}_i; \boldsymbol{\Theta}^{(t)}) y_i \sigma'(\langle \mathbf{w}_{m,j}^{(t)}, \mathbf{x}_i^{(p)} \rangle) \langle \mathbf{x}_i^{(p)}, \mathbf{v} \rangle}_{A_i},$$

where $A_i$ are independent random variables with $|A_i| \leq \widetilde{O}((\sigma_0 + \eta t)^2)$. Applying Hoeffding's inequality gives that with probability at least $1 - 1/(4d^{101}MJK)$ we have that $|\langle \nabla_{\mathbf{w}_{m,j}} \mathcal{L}^{(t)}, \mathbf{v} \rangle - \mathbb{E}[\langle \nabla_{\mathbf{w}_{m,j}} \mathcal{L}^{(t)}, \mathbf{v} \rangle]| = \widetilde{O}(n^{-1/2}(\sigma_0 + \eta t)^2)$. Applying union bound gives that with probability at least $1 - 1/(2d)$, $|\langle \nabla_{\mathbf{w}_{m,j}} \mathcal{L}^{(t)}, \mathbf{v} \rangle - \mathbb{E}[\langle \nabla_{\mathbf{w}_{m,j}} \mathcal{L}^{(t)}, \mathbf{v} \rangle]| = \widetilde{O}(n^{-1/2}(\sigma_0 + \eta t)^2), \forall m \in [M], j \in [J], t \leq d^{100}$. Similarly, we can prove $|\langle \nabla_{\boldsymbol{\theta}_m} \mathcal{L}^{(t)}, \mathbf{v} \rangle - \mathbb{E}[\langle \nabla_{\boldsymbol{\theta}_m} \mathcal{L}^{(t)}, \mathbf{v} \rangle]| = \widetilde{O}(n^{-1/2}(\sigma_0 + \eta t)^3)$. $\qquad \square$

### E.1 Exploration Stage

Denote $T_1 = \lfloor \eta^{-1} \sigma_0^{0.5} \rfloor$. The first stage ends when $t = T_1$. During the first stage training, we can prove that the neural network parameter maintains the following property.

**Lemma E.3.** For all $t \leq T_1$, we have the following properties hold,

- $\langle \mathbf{w}_{m,j}^{(t)}, \mathbf{v}_k \rangle = O(\sigma_0^{0.5}), \langle \mathbf{w}_{m,j}^{(t)}, \mathbf{c}_k \rangle = O(\sigma_0^{0.5}), \langle \mathbf{w}_{m,j}^{(t)}, \boldsymbol{\xi}_{i,p} \rangle = \widetilde{O}(\sigma_0^{0.5})$,

- $f_m(\mathbf{x}_i; \mathbf{W}^{(t)}) = \widetilde{O}(\sigma_0^{1.5})$,

- $|\ell'_{i,t} - 1/2| \leq \widetilde{O}(\sigma_0^{1.5})$,

- $\|\boldsymbol{\theta}_m^{(t)}\|_2 \leq \widetilde{O}(\sigma_0^{1.5})$,

- $\|\mathbf{h}(\mathbf{x}_i; \boldsymbol{\Theta}^{(t)})\|_\infty = \widetilde{O}(\sigma_0^{1.5}), \pi_m(\mathbf{x}_i; \boldsymbol{\Theta}^{(t)}) = M^{-1} + \widetilde{O}(\sigma_0^{1.5})$,

for all $m \in [M], k \in [k], i \in [n], p \geq 4$.

*Proof.* The first property is obvious since $\|\mathbf{w}_{m,j}^{(t)} - \mathbf{w}_{m,j}^{(0)}\|_2 \leq O(\eta T_1) = O(\sigma_0^{0.5})$ and thus

$$|f_m(\mathbf{x}_i; \mathbf{W}^{(t)})| \leq \sum_{p \in [P]} \sum_{j \in [J]} |\sigma(\langle \mathbf{w}_{m,j}^{(t)}, \mathbf{x}_i^{(p)} \rangle)| = \widetilde{O}(\sigma_0^{1.5}).$$

Then we show that the loss derivative is close to $1/2$ during this stage.

Let $s = y_i \pi_{m_{i,t}}(\mathbf{x}_i; \boldsymbol{\Theta}^{(t)}) f_{m_{i,t}}(\mathbf{x}_i, \mathbf{W}^{(t)})$, then we have that $|s| = \widetilde{O}(\sigma_0^{1.5})$ and

$$\left| \ell'_{i,t} - \frac{1}{2} \right| = \left| \frac{1}{e^s + 1} - 1/2 \right| \overset{(i)}{\leq} |s| = \widetilde{O}(\sigma_0^{1.5}),$$

where $(i)$ can be proved by considering $|s| \leq 1$ and $|s| > 1$.

Now we prove the fourth bullet in Lemma E.3. Because $|f_m| = \widetilde{O}(\sigma_0^{1.5})$, we can upper bound the gradient of the gating network by

$$
\begin{aligned}
\|\nabla_{\boldsymbol{\theta}_m}\mathcal{L}^{(t)}\|_2 = \bigg\| & \frac{1}{n}\sum_{i,p} \mathbb{1}(m_{i,t} = m)\ell'_{i,t}\pi_{m_{i,t}}(\mathbf{x}_i;\boldsymbol{\Theta}^{(t)})y_i f_{m_{i,t}}(\mathbf{x}_i;\mathbf{W}^{(t)})\mathbf{x}_i^{(p)} \\
& - \frac{1}{n}\sum_{i,p}\ell'_{i,t}\pi_{m_{i,t}}(\mathbf{x}_i;\boldsymbol{\Theta}^{(t)})\pi_m(\mathbf{x}_i;\boldsymbol{\Theta}^{(t)})y_i f_{m_{i,t}}(\mathbf{x}_i;\mathbf{W}^{(t)})\mathbf{x}_i^{(p)} \bigg\|_2. \\
= & \widetilde{O}(\sigma_0^{1.5}),
\end{aligned}
$$

where the last inequality is due to $|\ell'_{i,t}| \leq 1$, $\pi_m, \pi_{m_{i,t}} \in [0,1]$ and $\|\mathbf{x}_i^{(p)}\|_2 = O(1)$. This further implies that

$$
\|\boldsymbol{\theta}_m^{(t)}\|_2 = \|\boldsymbol{\theta}_m^{(t)} - \boldsymbol{\theta}_m^{(0)}\|_2 \leq \widetilde{O}(\sigma_0^{1.5}t\eta_r) = \widetilde{O}(\sigma_0^{1.5}),
$$

where the last inequality is by $\eta_r = \Theta(M^2)\eta$. The proof of $\|\mathbf{h}(\mathbf{x}_i;\boldsymbol{\Theta}^{(t)})\|_\infty \leq O(\sigma_0^{1.5})$ and $\pi_m(\mathbf{x}_i;\boldsymbol{\Theta}^{(t)}) = M^{-1} + O(\sigma_0^{1.5})$ are straight forward given $\|\boldsymbol{\theta}_m^{(t)}\|_2 = \widetilde{O}(\sigma_0^{1.5})$. $\qquad\square$

We will first investigate the property of the router.

**Lemma E.4.** $\max_{m\in[M]}|\mathbb{P}(m_{i,t} = m) - 1/M| = \widetilde{O}(\sigma_0^{1.5})$ for all $t \leq T_1$, $i \in [n]$ and $m \in [M]$.

*Proof.* By Lemma E.3 we have that $\|\mathbf{h}(\mathbf{x}_i;\boldsymbol{\Theta}^{(t)})\|_\infty \leq \widetilde{O}(\sigma_0^{1.5})$. Lemma 5.1 further implies that

$$
\max_{m\in[M]}|\mathbb{P}(m_{i,t} = m) - 1/M| = \widetilde{O}(\sigma_0^{1.5}).
$$

$\qquad\square$

**Lemma E.5.** We have following gradient update rules hold for the experts,

$$
\begin{aligned}
\langle\nabla_{\mathbf{w}_{m,j}}\mathcal{L}^{(t)}, \mathbf{v}_k\rangle &= -\frac{\mathbb{E}[\alpha^3] + \widetilde{O}(d^{-0.005})}{2KM^2}\sigma'(\langle\mathbf{w}_{m,j}^{(t)}, \mathbf{v}_k\rangle) + \widetilde{O}(\sigma_0^{2.5}), \\
\langle\nabla_{\mathbf{w}_{m,j}}\mathcal{L}^{(t)}, \mathbf{c}_k\rangle &= \widetilde{O}(d^{-0.005})\sigma'(\langle\mathbf{w}_{m,j}^{(t)}, \mathbf{c}_k\rangle) + \widetilde{O}(\sigma_0^{2.5}), \\
\langle\nabla_{\mathbf{w}_{m,j}}\mathcal{L}^{(t)}, \boldsymbol{\xi}_{i,p}\rangle &= \widetilde{O}(d^{-0.005})\sigma'(\langle\mathbf{w}_{m,j}^{(t)}, \boldsymbol{\xi}_{i,p}\rangle) + \widetilde{O}(\sigma_0^{2.5})
\end{aligned}
$$

for all $t \leq T_1, j \in [J], k \in [K], m \in [M], p \geq 4$. Besides, we have the following gradient norm upper bound holds

$$
\begin{aligned}
\|\nabla_{\mathbf{w}_{m,j}}\mathcal{L}^{(t)}\|_2 \leq & \sum_{k\in[K]}\frac{\mathbb{E}[\alpha^3] + \widetilde{O}(d^{-0.005})}{2KM^2}\sigma'(\langle\mathbf{w}_{m,j}^{(t)}, \mathbf{v}_k\rangle) + \sum_{k\in[K]}\widetilde{O}(d^{-0.005})\sigma'(\langle\mathbf{w}_{m,j}^{(t)}, \mathbf{c}_k\rangle) \\
& + \sum_{i\in[n],p\geq 4}\widetilde{O}(d^{-0.005})\sigma'(\langle\mathbf{w}_{m,j}^{(t)}, \boldsymbol{\xi}_{i,p}\rangle) + \widetilde{O}(\sigma_0^{2.5})
\end{aligned}
$$

for all $t \leq T_1, j \in [J], m \in [M]$.

*Proof.* The experts gradient can be computed as follows,

$$
\nabla_{\mathbf{w}_{m,j}}\mathcal{L}^{(t)} = \frac{1}{n}\sum_{i\in[n],p\in[P]}\mathbb{1}(m_{i,t} = m)\ell'_{i,t}f_m(\mathbf{x}_i;\mathbf{W}^{(t)})\pi_m(\mathbf{x}_i;\boldsymbol{\Theta}^{(t)})y_i\sigma'(\langle\mathbf{w}_{m,j}^{(t)}, \mathbf{x}_i^{(p)}\rangle)\mathbf{x}_i^{(p)}.
$$

We first compute the inner product $\langle \nabla_{\mathbf{w}_{m,j}} \mathcal{L}^{(t)}, \mathbf{c}_k \rangle$. By Lemma E.2, we have that $|\langle \nabla_{\mathbf{w}_{m,j}} \mathcal{L}^{(t)}, \mathbf{c}_k \rangle - \mathbb{E}[\langle \nabla_{\mathbf{w}_{m,j}} \mathcal{L}^{(t)}, \mathbf{c}_k \rangle]| = \widetilde{O}(n^{-1/2}\sigma_0) \le \widetilde{O}(\sigma_0^{2.5})$.

$$
\mathbb{E}[\langle \nabla_{\mathbf{w}_{m,j}} \mathcal{L}^{(t)}, \mathbf{c}_k \rangle] = -\frac{1}{n} \sum_{i \in \Omega_k} \mathbb{P}(m_{i,t} = m)\ell'_{i,t}\pi_m(\mathbf{x}_i; \boldsymbol{\Theta}^{(t)})\sigma'(\langle \mathbf{w}_{m,j}^{(t)}, \mathbf{c}_k \rangle)y_i\beta_i^3\|\mathbf{c}_k\|_2^2
$$
$$
- \frac{1}{n} \sum_{i \in [n], p \ge 4} \mathbb{P}(m_{i,t} = m)\ell'_{i,t}\pi_m(\mathbf{x}_i; \boldsymbol{\Theta}^{(t)})\sigma'(\langle \mathbf{w}_{m,j}^{(t)}, \boldsymbol{\xi}_{i,p} \rangle)y_i\langle \mathbf{c}_k, \boldsymbol{\xi}_{i,p} \rangle
$$
$$
= \left[ -\frac{1}{2nM} \sum_{i \in \Omega_k} y_i\beta_i^3\mathbb{P}(m_{i,t} = m) + \widetilde{O}(\sigma_0^{1.5}) \right]\sigma'(\langle \mathbf{w}_{m,j}^{(t)}, \mathbf{c}_k \rangle) + \widetilde{O}(\sigma_0^{2.5})
$$
$$
= \widetilde{O}(n^{-1/2} + \sigma_0^{1.5})\sigma'(\langle \mathbf{w}_{m,j}^{(t)}, \mathbf{c}_k \rangle) + \widetilde{O}(\sigma_0^{2.5})
$$
$$
= \widetilde{O}(d^{-0.005})\sigma'(\langle \mathbf{w}_{m,j}^{(t)}, \mathbf{c}_k \rangle) + \widetilde{O}(\sigma_0^{2.5})
$$

where the second equality is due to Lemma E.3 and D.2, the third equality is due to Lemma E.4, the last equality is by the choice of $n$ and $\sigma_0$. Next we compute the inner product $\langle \nabla_{\mathbf{w}_{m,j}} \mathcal{L}, \mathbf{v}_k \rangle$. By Lemma E.2, we have that $|\langle \nabla_{\mathbf{w}_{m,j}} \mathcal{L}^{(t)}, \mathbf{v}_k \rangle - \mathbb{E}[\langle \nabla_{\mathbf{w}_{m,j}} \mathcal{L}^{(t)}, \mathbf{v}_k \rangle]| = \widetilde{O}(n^{-1/2}\sigma_0) \le \widetilde{O}(\sigma_0^{2.5})$.

$$
\mathbb{E}[\langle \nabla_{\mathbf{w}_{m,j}} \mathcal{L}^{(t)}, \mathbf{v}_k \rangle] = -\frac{1}{n} \sum_{i \in \Omega_k} \mathbb{P}(m_{i,t} = m)\ell'_{i,t}\pi_m(\mathbf{x}_i; \boldsymbol{\Theta}^{(t)})\sigma'(\langle \mathbf{w}_{m,j}^{(t)}, \mathbf{v}_k \rangle)\alpha_i^3\|\mathbf{v}_k\|_2^2
$$
$$
- \frac{1}{n} \sum_{k' \ne k} \sum_{i \in \Omega_{k',k}} \mathbb{P}(m_{i,t} = m)\ell'_{i,t}\pi_m(\mathbf{x}_i; \boldsymbol{\theta}^{(t)})\sigma'(\langle \mathbf{w}_{m,j}^{(t)}, \mathbf{v}_k \rangle)\gamma_i^3 y_i\epsilon_i\|\mathbf{v}_k\|_2^2
$$
$$
- \frac{1}{n} \sum_{i \in [n], p \ge 4} \mathbb{P}(m_{i,t} = m)\ell'_{i,t}\pi_m(\mathbf{x}_i; \boldsymbol{\Theta}^{(t)})\sigma'(\langle \mathbf{w}_{m,j}^{(t)}, \boldsymbol{\xi}_{i,p} \rangle)y_i\langle \mathbf{v}_k, \boldsymbol{\xi}_{i,p} \rangle
$$
$$
= \left[ -\frac{1}{2nM} \sum_{i \in \Omega_k} \mathbb{P}(m_{i,t} = m)\alpha_i^3 - \frac{1}{2nM} \sum_{i \in \Omega_{k',k}} \mathbb{P}(m_{i,t} = m)\gamma_i^3 y_i\epsilon_i + O(\sigma_0^{1.5}) \right] \cdot
$$
$$
\sigma'(\langle \mathbf{w}_{m,j}^{(t)}, \mathbf{c}_k \rangle) + \widetilde{O}(\sigma_0^{2.5})
$$
$$
= \left( \mathbb{E}[\alpha^3] + \widetilde{O}(n^{-1/2} + \sigma_0^{1.5}) \right)\sigma'(\langle \mathbf{w}_{m,j}^{(t)}, \mathbf{v}_k \rangle) + \widetilde{O}(\sigma_0^{2.5})
$$
$$
= \left( \frac{\mathbb{E}[\alpha^3]}{2KM^2} + \widetilde{O}(d^{-0.005}) \right)\sigma'(\langle \mathbf{w}_{m,j}^{(t)}, \mathbf{v}_k \rangle) + \widetilde{O}(\sigma_0^{2.5})
$$

where the second equality is due to Lemma E.3 and D.2, the third equality is due to Lemma E.4, the last equality is by the choice of $n$ and $\sigma_0$. Finally we compute the inner product $\langle \nabla_{\mathbf{w}_{m,j}} \mathcal{L}, \boldsymbol{\xi}_{i,p} \rangle$ as follows

$$
\langle \nabla_{\mathbf{w}_{m,j}} \mathcal{L}^{(t)}, \boldsymbol{\xi}_{i,p} \rangle = -\frac{1}{n} \mathbb{1}(m_{i,t} = m)\ell'_{i,t}\pi_m(\mathbf{x}_i; \boldsymbol{\theta}^{(t)})\sigma'(\langle \mathbf{w}_{m,j}^{(t)}, \boldsymbol{\xi}_{i,p} \rangle)\|\boldsymbol{\xi}_{i,p}\|_2^2 + \widetilde{O}(\sigma_0 d^{-1/2})
$$
$$
= \widetilde{O}\left( \frac{\|\boldsymbol{\xi}_{i,p}\|_2^2}{n} \right)\sigma'(\langle \mathbf{w}_{m,j}^{(t)}, \boldsymbol{\xi}_{i,p} \rangle) + \widetilde{O}(\sigma_0 d^{-1/2})
$$
$$
= \widetilde{O}(d^{-0.005})\sigma'(\langle \mathbf{w}_{m,j}^{(t)}, \boldsymbol{\xi}_{i,p} \rangle) + \widetilde{O}(\sigma_0^{2.5}),
$$

where the first equality is due to Lemma D.2, second equality is due to $|\ell'_{i,t}| \le 1, \pi_m \in [0, 1]$ and the third equality is due to Lemma D.2 and our choice of $n, \sigma_0$. Based on previous results, let $B$ be the projection matrix on the linear space spanned by $\{\mathbf{v}_k\}_{k \in [K]} \cup \{\mathbf{c}_k\}_{k \in [K]}$. We can verify that

$$
\|\nabla_{\mathbf{w}_{m,j}} \mathcal{L}^{(t)}\|_2 \le \|B\nabla_{\mathbf{w}_{m,j}} \mathcal{L}^{(t)}\|_2 + \|(I - B)\nabla_{\mathbf{w}_{m,j}} \mathcal{L}^{(t)}\|_2
$$
$$
\le \sum_{k \in [K]} \frac{\mathbb{E}[\alpha^3] + \widetilde{O}(d^{-0.005})}{2KM^2}\sigma'(\langle \mathbf{w}_{m,j}^{(t)}, \mathbf{v}_k \rangle) + \sum_{k \in [K]} \widetilde{O}(d^{-0.005})\sigma'(\langle \mathbf{w}_{m,j}^{(t)}, \mathbf{c}_k \rangle)
$$
$$
+ \sum_{i \in [n], p \ge 4} \widetilde{O}(d^{-0.005})\sigma'(\langle \mathbf{w}_{m,j}^{(t)}, \boldsymbol{\xi}_{i,p} \rangle) + \widetilde{O}(\sigma_0^{2.5}).
$$

$\square$

Because we use normalized gradient descent, all the experts get trained at the same speed. Following lemma shows that expert $m$ will focus on the signal $\mathbf{v}_{k_m^*}$.

**Lemma E.6.** For all $m \in [M]$ and $t \leq T_1$, we have following inequalities hold,

$$\langle \mathbf{w}_{m,j_m^*}^{(t)}, \mathbf{v}_{k_m^*} \rangle = O(\sigma_0^{0.5}),$$

$$\langle \mathbf{w}_{m,j}^{(t)}, \mathbf{v}_k \rangle = \widetilde{O}(\sigma_0), \forall (j,k) \neq (j_m^*, k_m^*),$$

$$\langle \mathbf{w}_{m,j}^{(t)}, \mathbf{c}_k \rangle = \widetilde{O}(\sigma_0), \forall j \in [J], k \in [K],$$

$$\langle \mathbf{w}_{m,j}^{(t)}, \boldsymbol{\xi}_{i,p} \rangle = \widetilde{O}(\sigma_0), \forall j \in [J], i \in [n], p \geq 4.$$

*Proof.* For $t \leq T_1$, the update rule of every expert could be written as,

$$\langle \mathbf{w}_{m,j}^{(t+1)}, \mathbf{v}_k \rangle = \langle \mathbf{w}_{m,j}^{(t)}, \mathbf{v}_k \rangle + \frac{\eta}{\|\nabla_{\mathbf{W}_m}\mathcal{L}^{(t)}\|_F}\left[\frac{3\mathbb{E}[\alpha^3] + \widetilde{O}(d^{-0.005})}{2KM^2}\langle \mathbf{w}_{m,j}^{(t)}, \mathbf{v}_k \rangle^2 + \widetilde{O}(\sigma_0^{2.5})\right],$$

$$\langle \mathbf{w}_{m,j}^{(t+1)}, \boldsymbol{\xi}_{i,p} \rangle = \langle \mathbf{w}_{m,j}^{(t)}, \boldsymbol{\xi}_{i,p} \rangle + \frac{\eta}{\|\nabla_{\mathbf{W}_m}\mathcal{L}^{(t)}\|_F}\left[\widetilde{O}(d^{-0.005})\langle \mathbf{w}_{m,j}^{(t)}, \boldsymbol{\xi}_{i,p} \rangle^2 + \widetilde{O}(\sigma_0^{2.5})\right],$$

$$\langle \mathbf{w}_{m,j}^{(t+1)}, \mathbf{c}_k \rangle = \langle \mathbf{w}_{m,j}^{(t)}, \mathbf{c}_k \rangle + \frac{\eta}{\|\nabla_{\mathbf{W}_m}\mathcal{L}^{(t)}\|_F}\left[\widetilde{O}(d^{-0.005})\langle \mathbf{w}_{m,j}^{(t)}, \mathbf{c}_k \rangle^2 + \widetilde{O}(\sigma_0^{2.5})\right]. \tag{E.3}$$

For $t \leq T_1$, we have that $\langle \mathbf{w}_{m,j}^{(t)}, \mathbf{v}_{k_m^*} \rangle \leq O(\sigma_0^{0.5})$. By comparing the update rule of $\langle \mathbf{w}_{m,j}^{(t)}, \mathbf{v}_{k_m^*} \rangle$ and other inner product presented in (E.3) , We can prove that $\langle \mathbf{w}_{m,j}^{(t)}, \mathbf{v}_{k_m^*} \rangle$ will grow to $\sigma_0^{0.5}$ while other inner product still remain nearly unchanged.

**Comparison with** $\langle \mathbf{w}_{m,j}^{(t)}, \mathbf{v}_k \rangle$. Consider $k \neq k_m^*$. We want to get an upper bound of $\langle \mathbf{w}_{m,j}^{(t)}, \mathbf{v}_k \rangle$, so without loss of generality we can assume $\langle \mathbf{w}_{m,j}^{(t)}, \mathbf{v}_k \rangle = \Omega(\sigma_0)$. Since $\sigma_0 \leq d^{-0.01}$, we have that $\langle \mathbf{w}_{m,j}^{(t)}, \mathbf{v}_k \rangle^2 + \widetilde{O}(\sigma_0^{2.5}) = (1 + \widetilde{O}(d^{-0.005}))\langle \mathbf{w}_{m,j}^{(t)}, \mathbf{v}_k \rangle^2$. Therefore, we have that

$$\langle \mathbf{w}_{m,j}^{(t+1)}, \mathbf{v}_{k_m^*} \rangle = \langle \mathbf{w}_{m,j}^{(t)}, \mathbf{v}_{k_m^*} \rangle + \frac{\eta}{\|\nabla_{\mathbf{W}_m}\mathcal{L}^{(t)}\|_F}\frac{3\mathbb{E}[\alpha^3] + \widetilde{O}(d^{-0.005})}{2KM^2}\langle \mathbf{w}_{m,j}^{(t)}, \mathbf{v}_{k_m^*} \rangle^2, \tag{E.4}$$

$$\langle \mathbf{w}_{m,j}^{(t+1)}, \mathbf{v}_k \rangle = \langle \mathbf{w}_{m,j}^{(t)}, \mathbf{v}_k \rangle + \frac{\eta}{\|\nabla_{\mathbf{W}_m}\mathcal{L}^{(t)}\|_F}\frac{3\mathbb{E}[\alpha^3] + \widetilde{O}(d^{-0.005})}{2KM^2}\langle \mathbf{w}_{m,j}^{(t)}, \mathbf{v}_k \rangle^2. \tag{E.5}$$

Applying Lemma F.2 by choosing $C_t = (3\mathbb{E}[\alpha^3] + \widetilde{O}(d^{-0.005}))/(2KM^2\|\nabla_{\mathbf{W}_m}\mathcal{L}^{(t)}\|_F)$, $S = 1 + \widetilde{O}(d^{-0.005})$, $G = 1/(3\log(d)M^2)$ and verifying $\langle \mathbf{w}_m^{(0)}, \mathbf{v}_{k_m^*} \rangle \geq S(1 + G^{-1})\langle \mathbf{w}_m^{(0)}, \mathbf{v}_k \rangle$ (events in Section D hold), we have that $\langle \mathbf{w}_{m,j}^{(t)}, \mathbf{v}_k \rangle \leq O(G^{-1}\sigma_0) = \widetilde{O}(\sigma_0)$.

**Comparison with** $\langle \mathbf{w}_{m,j}^{(t)}, \mathbf{c}_k \rangle$. We want to get an upper bound of $\langle \mathbf{w}_{m,j}^{(t)}, \mathbf{c}_k \rangle$, so without loss of generality we can assume $\langle \mathbf{w}_{m,j}^{(t)}, \mathbf{v}_k \rangle = \Omega(\sigma_0)$. Because $\sigma_0 \leq d^{-0.01}$, one can easily show that

$$\langle \mathbf{w}_{m,j}^{(t+1)}, \mathbf{v}_{k_m^*} \rangle = \langle \mathbf{w}_{m,j}^{(t)}, \mathbf{v}_{k_m^*} \rangle + \frac{\eta}{\|\nabla_{\mathbf{W}_m}\mathcal{L}^{(t)}\|_F}\frac{3\mathbb{E}[\alpha^3] + \widetilde{O}(d^{-0.005})}{2KM^2}\langle \mathbf{w}_{m,j}^{(t)}, \mathbf{v}_{k_m^*} \rangle^2,$$

$$\langle \mathbf{w}_{m,j}^{(t+1)}, \mathbf{c}_k \rangle \leq \langle \mathbf{w}_{m,j}^{(t)}, \mathbf{c}_k \rangle + \frac{\eta}{\|\nabla_{\mathbf{W}_m}\mathcal{L}^{(t)}\|_F}\widetilde{O}(d^{-0.01})\langle \mathbf{w}_{m,j}^{(t)}, \mathbf{c}_k \rangle^2.$$

Again, applying Lemma F.2 by choosing $C_t = (3\mathbb{E}[\alpha^3] + \widetilde{O}(d^{-0.005}))/(2KM^2\|\nabla_{\mathbf{W}_m}\mathcal{L}^{(t)}\|_F)$, $S = \widetilde{O}(d^{-0.01})$, $G = 2$ and verifying $\langle \mathbf{w}_m^{(0)}, \mathbf{v}_{k_m^*} \rangle \geq S(1 + G^{-1})\langle \mathbf{w}_m^{(0)}, \mathbf{c}_k \rangle$ (events in Section D hold), we have that $\langle \mathbf{w}^{(t)}, \mathbf{v}_k \rangle \leq O(G^{-1}\sigma_0) = \widetilde{O}(\sigma_0)$.

**Comparison with** $\langle \mathbf{w}_{m,j}^{(t)}, \boldsymbol{\xi}_{i,p} \rangle$. The proof is exact the same as the one with $\mathbf{c}_k$. $\qquad\square$

Denote the iteration $T^{(m)}$ as the first time that $\|\nabla_{\mathbf{W}_m}\mathcal{L}^{(t)}\|_F \geq \sigma_0^{1.8}$. Then Following lemma gives an upper bound of $T^{(m)}$ for all $m \in \mathcal{M}$.

**Lemma E.7.** For all $m \in [M]$, we have that $T^{(m)} = \widetilde{O}(\eta^{-1}\sigma_0^{0.8})$ and thus $T^{(m)} < 0.01 T_1$. Besides, for all $T_m < t \leq T_1$ we have that

$$\langle \nabla_{\mathbf{w}_{m,j_m^*}} \mathcal{L}^{(t)}, \mathbf{v}_{k_m^*} \rangle \geq (1 - \sigma_0^{0.1}) \|\nabla_{\mathbf{W}_m} \mathcal{L}^{(t)}\|_F.$$

*Proof.* Let projection matrix $B = \mathbf{v}_{k_m^*} \mathbf{v}_{k_m^*}^\top \in \mathbb{R}^{d \times d}$, then we can divide the gradient into two orthogonal part

$$\|\nabla_{\mathbf{w}_{m,j_m^*}} \mathcal{L}^{(t)}\|_2 = \|B \nabla_{\mathbf{w}_{m,j_m^*}} \mathcal{L}^{(t)} + (I - B) \nabla_{\mathbf{w}_{m,j_m^*}} \mathcal{L}^{(t)}\|_2$$
$$\leq \|B \nabla_{\mathbf{w}_{m,j_m^*}} \mathcal{L}^{(t)}\|_2 + \|(I - B) \nabla_{\mathbf{w}_{m,j_m^*}} \mathcal{L}^{(t)}\|_2$$

Recall that

$$\nabla_{\mathbf{w}_{m,j_m^*}} \mathcal{L}^{(t)} = \frac{1}{n} \sum_{i,p} \mathbb{1}(m_{i,t} = m) \ell_{i,t}' \pi_m(\mathbf{x}_i; \boldsymbol{\Theta}^{(t)}) y_i \sigma'(\langle \mathbf{w}_{m,j_m^*}^{(t)}, \mathbf{x}_i^{(p)} \rangle) \mathbf{x}_i^{(p)},$$

So we have that

$$\|(I - B) \nabla_{\mathbf{w}_{m,j_m^*}} \mathcal{L}^{(t)}\|_2 = \left\| \frac{1}{n} \sum_{i,p} \mathbb{1}(m_{i,t} = m) \ell_{i,t}' \pi_m(\mathbf{x}_i; \boldsymbol{\Theta}^{(t)}) y_i \sigma'(\langle \mathbf{w}_{m,j_m^*}^{(t)}, \mathbf{x}_i^{(p)} \rangle)(I - B) \mathbf{x}_i^{(p)} \right\|_2$$
$$\leq \frac{1}{n} \sum_{i,p} \left\| \sigma'(\langle \mathbf{w}_{m,j_m^*}^{(t)}, \mathbf{x}_i^{(p)} \rangle)(I - B) \mathbf{x}_i^{(p)} \right\|_2$$
$$\leq \widetilde{O}(\sigma_0^2),$$

where the first inequality is by $|\ell_{i,t}'| \leq 1, \pi_m \in [0, 1]$ and the second equality is because

1. when $\mathbf{x}_i^{(p)}$ align with $\mathbf{v}_{k_m^*}$, $(I - B)\mathbf{x}_i^{(p)} = \mathbf{0}$.

2. when $\mathbf{x}_i^{(p)}$ doesn't align with $\mathbf{v}_{k_m^*}$, $\langle \mathbf{w}_{m,j_m^*}^{(t)}, \mathbf{x}_i^{(p)} \rangle = \widetilde{O}(\sigma_0)$.

Therefore, we have that

$$\|\nabla_{\mathbf{w}_{m,j_m^*}} \mathcal{L}^{(t)}\|_2 \leq \|B \nabla_{\mathbf{w}_{m,j_m^*}} \mathcal{L}^{(t)}\|_2 + \widetilde{O}(\sigma_0^2) = \langle \nabla_{\mathbf{w}_{m,j_m^*}} \mathcal{L}^{(t)}, \mathbf{v}_{k_m^*} \rangle + \widetilde{O}(\sigma_0^2).$$

We next compute the gradient of the neuron $\mathbf{w}_{m,j}, j \neq j_m^*$,

$$\|\nabla_{\mathbf{w}_{m,j}} \mathcal{L}^{(t)}\|_2 = \left\| \frac{1}{n} \sum_{i,p} \mathbb{1}(m_{i,t} = m) \ell_{i,t}' \pi_m(\mathbf{x}_i; \boldsymbol{\Theta}^{(t)}) y_i \sigma'(\langle \mathbf{w}_{m,j}^{(t)}, \mathbf{x}_i^{(p)} \rangle) \mathbf{x}_i^{(p)} \right\|_2 = \widetilde{O}(\sigma_0^2), \tag{E.6}$$

where the inequality is by $\langle \mathbf{w}_{m,j}^{(t)}, \mathbf{x}_i^{(p)} \rangle = \widetilde{O}(\sigma_0), \forall j \neq j_m^*$ which is due to Lemma E.6. Now we can upper bound the gradient norm,

$$\|\nabla_{\mathbf{W}_m} \mathcal{L}^{(t)}\|_F \leq \sum_{j \in [J]} \|\nabla_{\mathbf{w}_{m,j}} \mathcal{L}^{(t)}\|_2 \leq \|\nabla_{\mathbf{w}_{m,j_m^*}} \mathcal{L}^{(t)}\|_2 + \widetilde{O}(\sigma_0^2). \tag{E.7}$$

When $\|\nabla_{\mathbf{W}_m} \mathcal{L}^{(t)}\|_F \geq \sigma_0^{1.8}$, it is obviously that

$$\langle \nabla_{\mathbf{w}_{m,j_m^*}} \mathcal{L}, \mathbf{v}_{k_m^*} \rangle \geq \|\nabla_{\mathbf{w}_{m,j_m^*}} \mathcal{L}^{(t)}\|_2 - \widetilde{O}(\sigma_0^2) \geq \|\nabla_{\mathbf{W}_m} \mathcal{L}^{(t)}\|_F - \widetilde{O}(\sigma_0^2) \geq (1 - \sigma_0^{0.1}) \|\nabla_{\mathbf{W}_m} \mathcal{L}^{(t)}\|_F,$$

where the first inequality is by (E.6) and the second inequality is by (E.7). Now let us give an upper bound for $T^{(m)}$. During the period $t \leq T^{(m)}$, $\|\nabla_{\mathbf{W}_m} \mathcal{L}^{(t)}\|_F < \sigma_0^{1.8}$. On the one hand, by Lemma E.5 we have that

$$\|\nabla_{\mathbf{W}_m} \mathcal{L}^{(t)}\|_2 \geq -\langle \nabla_{\mathbf{w}_{m,j}} \mathcal{L}^{(t)}, \mathbf{v}_{k_m^*} \rangle = \frac{3\mathbb{E}[\alpha^3] - \widetilde{O}(d^{-0.005})}{2KM^2} [\langle \mathbf{w}_{m,j_m^*}^{(t)}, \mathbf{v}_{k_m^*} \rangle]^2 - \widetilde{O}(\sigma_0^{2.5})$$

which implies that the inner product $\langle \mathbf{w}_{m,j_m^*}^{(t)}, \mathbf{v}_{k_m^*} \rangle \leq \widetilde{O}(\sigma_0^{0.9})$. On the other hand, by Lemma E.6 we have that

$$\langle \mathbf{w}_{m,j_m^*}^{(t+1)}, \mathbf{v}_{k_m^*} \rangle \geq \langle \mathbf{w}_{m,j_m^*}^{(t)}, \mathbf{v}_{k_m^*} \rangle + \frac{\eta}{\|\nabla_{\mathbf{W}_m} \mathcal{L}^{(t)}\|_F} \Theta(\frac{1}{KM^2}) \langle \mathbf{w}_{m,j_m^*}^{(t)}, \mathbf{v}_{k_m^*} \rangle^2$$

$$\geq \langle \mathbf{w}_{m,j_m^*}^{(t)}, \mathbf{v}_{k_m^*} \rangle + \Theta\left(\frac{\eta}{KM^2\sigma_0^{1.8}}\right) \langle \mathbf{w}_{m,j_m^*}^{(t)}, \mathbf{v}_{k_m^*} \rangle^2$$

$$\geq \langle \mathbf{w}_{m,j_m^*}^{(t)}, \mathbf{v}_{k_m^*} \rangle + \Theta\left(\frac{\eta}{KM^2\sigma_0^{0.8}}\right) \langle \mathbf{w}_{m,j_m^*}^{(t)}, \mathbf{v}_{k_m^*} \rangle,$$

where last inequality is by $\langle \mathbf{w}_{m,j_m^*}^{(t)}, \mathbf{v}_{k_m^*} \rangle \geq 0.1\sigma_0$. Therefore, we have that the inner product $\langle \mathbf{w}_{m,j}^{(t)}, \mathbf{v}_{k_m^*} \rangle$ grows exponentially and will reach $\widetilde{O}(\sigma_0^{0.9})$ within $\widetilde{O}(\eta^{-1}\sigma_0^{0.8})$ iterations.

$\square$

Recall that $T_1 = \lfloor \eta^{-1}\sigma_0^{0.5} \rfloor$, following Lemma shows that the expert $m \in [M]$ only learns one feature during the first stage,

**Lemma E.8.** For all $t \leq T_1, m \in [M]$, we have that

$$\langle \mathbf{w}_{m,j_m^*}^{(t)}, \mathbf{v}_{k_m^*} \rangle = O(\sigma_0^{0.5}),$$

$$\langle \mathbf{w}_{m,j}^{(t)}, \mathbf{v}_k \rangle = \widetilde{O}(\sigma_0), \forall (j,k) \neq (j_m^*, k_m^*),$$

$$\langle \mathbf{w}_{m,j}^{(t)}, \mathbf{c}_k \rangle = \widetilde{O}(\sigma_0), \forall j \in [J], k \in [K],$$

$$\langle \mathbf{w}_{m,j}^{(t)}, \boldsymbol{\xi}_{i,p} \rangle = \widetilde{O}(\sigma_0), \forall j \in [J], i \in [n], p \geq 4.$$

Besides $\langle \mathbf{w}_{m,j_m^*}^{(t)}, \mathbf{v}_{k_m^*} \rangle \geq (1 - \sigma_0^{0.1})\eta t$, for all $t \geq T_1/2$.

*Proof.* By Lemma E.7, we have $T^{(m)} = \widetilde{O}(\eta^{-1}\sigma_0^{0.8}) < \sigma_0^{0.2} \cdot T_1$. Notice that $\langle \nabla_{\mathbf{w}_{m,j_m^*}} \mathcal{L}^{(t)}, \mathbf{v}_{k^*} \rangle \geq (1 - \sigma_0^{0.1})\|\nabla_{\mathbf{W}_m} \mathcal{L}^{(t)}\|_F$, for all $T_m \leq t \leq T_1$. Therefore, we have that

$$\langle \mathbf{w}_{m,j_m^*}^{(t+1)}, \mathbf{v}_{k_m^*} \rangle \geq \langle \mathbf{w}_{m,j_m^*}^{(t)}, \mathbf{v}_{k_m^*} \rangle + (1 - \sigma_0^{0.1})\eta, \forall T_m \leq t \leq T_1,$$

which implies $\langle \mathbf{w}_{m,j_m^*}^{(t)}, \mathbf{v}_{k_m^*} \rangle \geq (1 - O(\sigma_0^{0.1}))\eta t, \forall t \geq T_1/2$. Finally, applying Lemma E.6 completes the proof.

$\square$

## E.2 Router Learning Stage

Denote $T_2 = \lfloor \eta^{-1}M^{-2} \rfloor$, The second stage ends when $t = T_2$. Given $\mathbf{x} = [\alpha y\mathbf{v}_k, \beta\mathbf{c}_k, \gamma\epsilon\mathbf{v}_{k'}, \boldsymbol{\xi}]$, we denote by $\bar{x} = [\mathbf{0}, \beta\mathbf{c}_k, \mathbf{0}, \ldots, \mathbf{0}]$ the one only keeps cluster-center signal and denote by $\widehat{x} = [\alpha y\mathbf{v}_k, \mathbf{0}, \gamma\epsilon\mathbf{v}_{k'}, \mathbf{0}]$ the one that only keeps feature signal and feature noise.

For all $T_1 \leq t \leq T_2$, we will show that the router only focuses on the cluster-center signals and the experts only focus on the feature signals, i.e., we will prove that $|f_m(\mathbf{x}_i; \mathbf{W}^{(t)}) - f_m(\widehat{\mathbf{x}}_i; \mathbf{W}^{(t)})|$ and $\|\mathbf{h}(\mathbf{x}_i; \boldsymbol{\Theta}^{(t)}) - \mathbf{h}(\bar{\mathbf{x}}_i; \boldsymbol{\Theta}^{(t)})\|_\infty$ are small. In particular, We claim that for all $T_1 \leq t \leq T_2$, following proposition holds.

**Proposition E.9.** For all $T_1 \leq t \leq T_2$, following inequalities hold,

$$|f_m(\mathbf{x}_i; \mathbf{W}^{(t)}) - f_m(\widehat{\mathbf{x}}_i; \mathbf{W}^{(t)})| \leq O(d^{-0.001}), \forall m \in [M], i \in [n], \tag{E.8}$$

$$\|\mathbf{h}(\mathbf{x}_i; \boldsymbol{\Theta}^{(t)}) - \mathbf{h}(\bar{\mathbf{x}}_i; \boldsymbol{\Theta}^{(t)})\|_\infty \leq O(d^{-0.001}), \forall i \in [n], \tag{E.9}$$

$$\mathbb{P}(m_{i,t} = m), \pi_m(\mathbf{x}_i; \boldsymbol{\Theta}^{(t)}) = \Omega(1/M), \forall m \in [M], i \in \Omega_{k_m^*}. \tag{E.10}$$

Proposition E.9 implies that expert will only focus on the label signal and router will only focus on the cluster-center signal. We will prove Proposition E.9 by induction. Before we move into the detailed proof of Proposition E.9, we will first prove some important lemmas.

**Lemma E.10.** For all $T_1 \leq t \leq T_2$, the neural network parameter maintains following property.

- $|f_m(\mathbf{x}_i; \mathbf{W}^{(t)})| = O(1), \forall m \in [M],$

- $\pi_{m_{i,t}}(\mathbf{x}_i; \mathbf{\Theta}^{(t)}) = \Omega(1/M), \forall i \in [n].$

*Proof.* Because we use normalized gradient descent, the first bullet would be quite straight forward.
$$|f_m(\mathbf{x}_i, \mathbf{W}^{(t)})| = \sum_{j \in [J]} \sum_{p \in [P]} \sigma(\langle \mathbf{w}_{m,j}^{(t)}, \mathbf{x}_i^{(p)} \rangle) \overset{(i)}{=} O(1),$$
where (i) is by $\|\mathbf{w}_{m,j}^{(t)} - \mathbf{w}_{m,j}^{(0)}\|_2 = O(\eta T_2) = O(M^{-2})$ and $\mathbf{x}_i^{(p)} = O(1)$.

Now we prove the second bullet. By Lemma C.4, we have that $h_{m_{i,t}}(\mathbf{x}; \mathbf{\Theta}) \geq \max_m h_m(\mathbf{x}; \mathbf{\Theta}) - 1$, which implies that
$$\pi_{m_{i,t}}(\mathbf{x}_i; \mathbf{\Theta}^{(t)}) = \frac{\exp(h_{m_{i,t}}(\mathbf{x}_i; \mathbf{\Theta}^{(t)}))}{\sum_m \exp(h_m(\mathbf{x}; \mathbf{\Theta}^{(t)}))} \geq \frac{\exp(h_{m_{i,t}}(\mathbf{x}_i; \mathbf{\Theta}^{(t)}))}{M \max_m \exp(h_m(\mathbf{x}; \mathbf{\Theta}^{(t)}))} \geq \frac{1}{eM}.$$
$\square$

**Lemma E.11.** Denote $\delta_{\mathbf{\Theta}} = \max_i \|\mathbf{h}(\bar{\mathbf{x}}_i; \mathbf{\Theta}) - \mathbf{h}(\mathbf{x}_i; \mathbf{\Theta})\|_\infty$ and let the random variable $\bar{m}_{i,t}$ be expert that get routed if we use the gating network output $\mathbf{h}(\bar{\mathbf{x}}_i; \mathbf{\Theta}^{(t)})$ instead. Then we have following inequalities,
$$|\pi_m(\mathbf{x}_i; \mathbf{\Theta}) - \pi_m(\bar{\mathbf{x}}_i; \mathbf{\Theta})| = O(\delta_{\mathbf{\Theta}}), \forall m \in [M], i \in [n], . \tag{E.11}$$
$$|\mathbb{P}(m_{i,t} = m) - \mathbb{P}(\bar{m}_{i,t} = m)| = O(M^2 \delta_{\mathbf{\Theta}}), \forall m \in [M], i \in [n]. \tag{E.12}$$

*Proof.* By definition of $\delta_{\mathbf{\Theta}}$, we have that $\|\mathbf{h}(\mathbf{x}_i; \mathbf{\Theta}^{(t)}) - \mathbf{h}(\bar{\mathbf{x}}_i; \mathbf{\Theta}^{(t)})\|_\infty \leq \delta_{\mathbf{\Theta}}$. Then applying Lemma 5.1 gives $|\mathbb{P}(m_{i,t} = m) - \mathbb{P}(\bar{m}_{k,t} = m)| = \widetilde{O}(\delta_{\mathbf{\Theta}}), \forall m \in [M], i \in [n]$, which completes the proof for (E.12).

Next we prove (E.11), which needs more effort. For all $i \in [n]$, we have
$$\pi_m(\mathbf{x}_i; \mathbf{\Theta}) = \frac{\pi_m(\bar{\mathbf{x}}_i; \mathbf{\Theta}) \exp(h_m(\mathbf{x}_i; \mathbf{\Theta}) - h_m(\bar{\mathbf{x}}_i; \mathbf{\Theta}))}{\sum_{m'} \pi_{m'}(\bar{\mathbf{x}}_i; \mathbf{\Theta}) \exp(h_{m'}(\mathbf{x}_i; \mathbf{\Theta}) - h_{m'}(\bar{\mathbf{x}}_i; \mathbf{\Theta}))}.$$
Let $\delta_{m'} = \exp(h_{m'}(\mathbf{x}_i; \mathbf{\Theta}) - h_{m'}(\bar{\mathbf{x}}_i; \mathbf{\Theta})) = 1 + O(\delta_{\mathbf{\Theta}})$. Then for sufficiently small $\delta_{\mathbf{\Theta}}$, we have that $\delta_{m'} \geq 0.5$. Then we can further compute
$$\begin{aligned}
|\pi_m(\mathbf{x}_i; \mathbf{\Theta}^{(t)}) - \pi_m(\bar{\mathbf{x}}_i; \mathbf{\Theta})| &= \pi_m(\bar{\mathbf{x}}_i; \mathbf{\Theta}) \left| \frac{\delta_m}{\sum_{m'} \pi_{m'}(\bar{\mathbf{x}}_i; \mathbf{\Theta})\delta_{m'}} - 1 \right| \\
&= \pi_m(\bar{\mathbf{x}}_i; \mathbf{\Theta}) \frac{|\sum_{m'} \pi_{m'}(\bar{\mathbf{x}}_i; \mathbf{\Theta})(\delta_{m'} - \delta_m)|}{\sum_{m'} \pi_{m'}(\bar{\mathbf{x}}_i; \mathbf{\Theta})\delta_{m'}} \\
&\leq \pi_m(\bar{\mathbf{x}}_i; \mathbf{\Theta}) \frac{\sum_{m'} \pi_{m'}(\bar{\mathbf{x}}_i; \mathbf{\Theta})|\delta_{m'} - \delta_m|}{\sum_{m'} \pi_{m'}(\bar{\mathbf{x}}_i; \mathbf{\Theta})\delta_{m'}} \\
&\leq O(\delta_{\mathbf{\Theta}}),
\end{aligned}$$
where the last inequality is by $|\delta_{m'} - \delta_m| \leq O(\delta_{\mathbf{\Theta}})$, $\pi_m(\bar{\mathbf{x}}_i; \mathbf{\Theta}) \leq 1$ and $\sum_{m'} \pi_{m'}(\bar{\mathbf{x}}_i; \mathbf{\Theta})\delta_{m'} \geq [\sum_{m'} \pi_{m'}(\bar{\mathbf{x}}_i; \mathbf{\Theta})]/2 = 0.5$. $\square$

Following Lemma implies that the pattern learned by experts during the first stage won't change in the second stage.

**Lemma E.12.** Suppose (E.8), (E.9), (E.10) hold for all $t \in [T_1, T] \subseteq [T_1, T_2 - 1]$, then we have following inequalities hold for all $t \in [T_1, T + 1]$,
$$\begin{aligned}
\langle \mathbf{w}_{m,j_m^*}^{(t)}, \mathbf{v}_{k_m^*} \rangle &\geq (1 - O(\sigma_0^{0.1}))\eta t, \\
\langle \mathbf{w}_{m,j}^{(t)}, \mathbf{v}_k \rangle &= \widetilde{O}(\sigma_0), \forall (j, k) \neq (j_m^*, k_m^*), \\
\langle \mathbf{w}_{m,j}^{(t)}, \mathbf{c}_k \rangle &= \widetilde{O}(\sigma_0), \forall j \in [J], k \in [K], \\
\langle \mathbf{w}_{m,j}^{(t)}, \boldsymbol{\xi}_{i,p} \rangle &= \widetilde{O}(\sigma_0), \forall j \in [J], k \in [K], i \in [n], p \geq 4.
\end{aligned}$$

*Proof.* Most of the proof exactly follows the proof in the first stage, so we only list some key steps here. Recall that

$$\nabla_{\mathbf{w}_{m,j}}\mathcal{L}^{(t)} = \frac{1}{n}\sum_{i,p} \mathbb{1}(m_{i,t} = m)\ell'_{i,t}\pi_m(\mathbf{x}_i; \boldsymbol{\Theta}^{(t)})y_i\sigma'(\langle \mathbf{w}_{m,j}^{(t)}, \mathbf{x}_i^{(p)}\rangle)\mathbf{x}_i^{(p)}.$$

In the proof of Lemma E.5, we do Taylor expansion at the zero point. Now we will do Taylor expansion at $f_m(\widehat{\mathbf{x}}_i; \mathbf{W})$ and $\pi(\bar{\mathbf{x}}_i; \boldsymbol{\Theta})$ as follows,

$$|\pi_m(\mathbf{x}_i; \boldsymbol{\Theta}^{(t)})f_m(\mathbf{x}_i; \mathbf{W}^{(t)}) - \pi_m(\bar{\mathbf{x}}_i; \boldsymbol{\Theta}^{(t)})f_m(\widehat{\mathbf{x}}_i; \mathbf{W}^{(t)})|$$
$$\leq |\pi_m(\bar{\mathbf{x}}_i; \boldsymbol{\Theta}^{(t)})[f_m(\mathbf{x}_i; \mathbf{W}^{(t)}) - f_m(\widehat{\mathbf{x}}_i; \mathbf{W}^{(t)})]| + |[\pi_m(\mathbf{x}_i; \boldsymbol{\Theta}^{(t)}) - \pi_m(\bar{\mathbf{x}}_i; \boldsymbol{\Theta}^{(t)})]f_m(\mathbf{x}_i; \mathbf{W}^{(t)})|$$
$$\leq |f_m(\mathbf{x}_i; \mathbf{W}^{(t)}) - f_m(\widehat{\mathbf{x}}_i; \mathbf{W}^{(t)})| + O(|\pi_m(\mathbf{x}_i; \boldsymbol{\Theta}^{(t)}) - \pi_m(\bar{\mathbf{x}}_i; \boldsymbol{\Theta}^{(t)})|)$$
$$\leq O(d^{-0.001}),$$

where the first inequality is by triangle inequality, the second inequality is by $\pi_m(\bar{\mathbf{x}}_i; \boldsymbol{\Theta}^{(t)}) \leq 1$ and $|f_m(\mathbf{x}_i; \mathbf{W}^{(t)})| = O(1)$ in Lemma E.10, the third inequality is by (E.8), (E.9) and (E.11).

Then follow the proof of Lemma E.5, we have that

$$\mathbb{E}[\langle \nabla_{\mathbf{w}_{m,j}}\mathcal{L}^{(t)}, \mathbf{v}_{k_m^*}\rangle] = -\frac{1}{n}\sum_{i\in\Omega_{k_m^*}}\mathbb{P}(m_{i,t}=m)\ell'_{i,t}\pi_m(\mathbf{x}_i; \boldsymbol{\Theta}^{(t)})\sigma'(\langle\mathbf{w}_{m,j}^{(t)}, \mathbf{v}_{k_m^*}\rangle)\alpha_i^3\|\mathbf{v}_{k_m^*}\|_2^2$$

$$-\frac{1}{n}\sum_{i\in\Omega_{k',k_m^*}}\mathbb{P}(m_{i,t}=m)\ell'_{i,t}\pi_m(\mathbf{x}_i; \boldsymbol{\Theta}^{(t)})\sigma'(\langle\mathbf{w}_{m,j}^{(t)}, \mathbf{v}_{k_m^*}\rangle)\gamma_i^3 y_i\epsilon_i\|\mathbf{v}_{k_m^*}\|_2^2$$

$$-\frac{1}{n}\sum_{i,p}\mathbb{P}(m_{i,t}=m)\ell'_{i,t}\pi_m(\mathbf{x}_i; \boldsymbol{\Theta}^{(t)})\sigma'(\langle\mathbf{w}_{m,j}^{(t)}, \boldsymbol{\xi}_{i,p}\rangle)y_i\langle\mathbf{v}_{k_m^*}, \boldsymbol{\xi}_{i,p}\rangle$$

$$= \left[-\widetilde{\Theta}\left(\frac{1}{n}\right)\sum_{i\in\Omega_{k_m^*}}\mathbb{P}(m_{i,t}=m)\alpha_i^3 - \widetilde{\Theta}\left(\frac{1}{n}\right)\sum_{i\in\Omega_{k',k_m^*}}\mathbb{P}(m_{i,t}=m)\gamma_i^3 y_i\epsilon_i\right.$$

$$\left.+ O(d^{-0.001})\right]\cdot\sigma'(\langle\mathbf{w}_{m,j}^{(t)}, \mathbf{v}_{k_m^*}\rangle) + \widetilde{O}(d^{-1/2})$$

$$\overset{(i)}{=} -\widetilde{\Theta}(1)\sigma'(\langle\mathbf{w}_{m,j}^{(t)}, \mathbf{v}_{k_m^*}\rangle),$$

where (i) is due to (E.10): $\mathbb{P}(m_{i,t}=m) \geq \Theta(1/M), \forall i \in \Omega_{k_m^*}, m \in [M]$. Again follow Lemma E.5 and Lemma E.6, we further have that

$$\langle\nabla_{\mathbf{w}_{m,j}}\mathcal{L}^{(t)}, \mathbf{v}_k\rangle = -\widetilde{\Theta}(1)[\langle\mathbf{w}_{m,j}^{(t)}, \mathbf{v}_k\rangle]^2,$$
$$\langle\nabla_{\mathbf{w}_{m,j}}\mathcal{L}^{(t)}, \mathbf{c}_k\rangle = \widetilde{O}(1)[\langle\mathbf{w}_{m,j}^{(t)}, \mathbf{c}_k\rangle]^2,$$
$$\langle\nabla_{\mathbf{w}_{m,j}}\mathcal{L}^{(t)}, \boldsymbol{\xi}_{i,p}\rangle = \widetilde{O}(1)[\langle\mathbf{w}_{m,j}^{(t)}, \boldsymbol{\xi}_{i,p}\rangle]^2.$$

Thus for all $T_1 \leq t \leq T$, the update rule of every expert could be written as,

$$\langle\mathbf{w}_{m,j}^{(t+1)}, \mathbf{v}_{k_m^*}\rangle = \langle\mathbf{w}_{m,j}^{(t)}, \mathbf{v}_{k_m^*}\rangle + \widetilde{\Theta}(1)\frac{\eta}{\|\nabla_{\mathbf{w}_m}\mathcal{L}^{(t)}\|_F}\langle\mathbf{w}_{m,j}^{(t)}, \mathbf{v}_{k_m^*}\rangle^2$$

$$\langle\mathbf{w}_{m,j}^{(t+1)}, \mathbf{v}_k\rangle = \langle\mathbf{w}_{m,j}^{(t)}, \mathbf{v}_k\rangle + \widetilde{O}(1)\frac{\eta}{\|\nabla_{\mathbf{w}_m}\mathcal{L}^{(t)}\|_F}\langle\mathbf{w}_{m,j}^{(t)}, \mathbf{v}_k\rangle^2$$

$$\langle\mathbf{w}^{(t+1)}, \boldsymbol{\xi}_{i,p}\rangle = \langle\mathbf{w}^{(t)}, \boldsymbol{\xi}_{i,p}\rangle + \widetilde{O}(1)\frac{\eta}{\|\nabla_{\mathbf{w}_m}\mathcal{L}^{(t)}\|_F}\langle\mathbf{w}^{(t)}, \boldsymbol{\xi}_{i,p}\rangle^2$$

$$\langle\mathbf{w}_{m,j}^{(t+1)}, \mathbf{c}_k\rangle = \langle\mathbf{w}_{m,j}^{(t)}, \mathbf{c}_k\rangle + \widetilde{O}(1)\frac{\eta}{\|\nabla_{\mathbf{w}_m}\mathcal{L}^{(t)}\|_F}\langle\mathbf{w}_{m,j}^{(t)}, \mathbf{c}_k\rangle^2.$$

By the first stage of training we have that $\langle \mathbf{w}_{m,j}^{(T_1)}, \mathbf{v}_{k_m^*} \rangle = \Theta(\sigma_0^{0.5})$, while others remains $\widetilde{O}(\sigma_0)$. Then we can use Lemma F.2, by choosing $S = \widetilde{\Theta}(1)$ and $G = 2$, then we have that

$$\langle \mathbf{w}_{m,j}^{(t)}, \mathbf{v}_{k_m^*} \rangle = O(1).$$
$$\langle \mathbf{w}_{m,j}^{(t)}, \mathbf{v}_k \rangle = \widetilde{O}(\sigma_0), \forall k \neq k_m^*.$$
$$\langle \mathbf{w}_{m,j}^{(t)}, \mathbf{c}_k \rangle = \widetilde{O}(\sigma_0).$$
$$\langle \mathbf{w}^{(t)}, \boldsymbol{\xi}_{i,p} \rangle = \widetilde{O}(\sigma_0).$$

Then following Lemma E.7 and E.8, we can prove that for all $T_1 \leq t \leq T + 1$, $m \in [M]$,

$$\langle \mathbf{w}_{m,j_m^*}^{(t)}, \mathbf{v}_{k_m^*} \rangle \geq (1 - O(\sigma_0^{0.1}))\eta t,$$
$$\langle \mathbf{w}_{m,j}^{(t)}, \mathbf{v}_k \rangle = \widetilde{O}(\sigma_0), \forall (j,k) \neq (j_m^*, k_m^*),$$
$$\langle \mathbf{w}_{m,j}^{(t)}, \mathbf{c}_k \rangle = \widetilde{O}(\sigma_0), \forall j \in [J], k \in [K],$$
$$\langle \mathbf{w}_{m,j}^{(t)}, \boldsymbol{\xi}_{i,p} \rangle = \widetilde{O}(\sigma_0), \forall j \in [J], i \in [n], p \geq 4.$$

$\square$

By the result of expert training we have following results

**Lemma E.13.** Suppose (E.8), (E.9), (E.10) hold for all $t \in [T_1, T] \subseteq [T_1, T_2 - 1]$, then we have that $|f_m(\mathbf{x}_i; \mathbf{W}^{(t)}) - f_m(\widehat{\mathbf{x}}_i; \mathbf{W}^{(t)})| = \widetilde{O}(\sigma_0^3)$ for all $m \in [M]$ and $i \in [n]$, $t \in [T_1, T + 1]$. Besides,

$$y_i f_m(\widehat{\mathbf{x}}_i; \mathbf{W}^{(t)}) = \sum_{j \in [J]} \left[ \alpha_i^3 \sigma(\langle \mathbf{w}_{m,j}^{(t)}, \mathbf{v}_k \rangle) + \gamma_i^3 \sigma(\langle \mathbf{w}_{m,j}^{(t)}, \mathbf{v}_{k'} \rangle) \right], \forall i \in \Omega_{k,k'}^+, m \in [M],$$
$$y_i f_m(\widehat{\mathbf{x}}_i; \mathbf{W}^{(t)}) = \sum_{j \in [J]} \left[ \alpha_i^3 \sigma(\langle \mathbf{w}_{m,j}^{(t)}, \mathbf{v}_k \rangle) - \gamma_i^3 \sigma(\langle \mathbf{w}_{m,j}^{(t)}, \mathbf{v}_{k'} \rangle) \right], \forall i \in \Omega_{k,k'}^-, m \in [M].$$

*Proof.* For all $i \in \Omega_k$, we have that

$$\left| f_m(\mathbf{x}_i; \mathbf{W}^{(t)}) - f_m(\widehat{\mathbf{x}}_i; \mathbf{W}^{(t)}) \right| \leq \left| \sum_{j \in [J]} \sigma(\langle \mathbf{w}_{m,j}^{(t)}, \mathbf{c}_k \rangle) \right| + \left| \sum_{j \in [J], p \geq 4} \sigma(\langle \mathbf{w}_{m,j}^{(t)}, \boldsymbol{\xi}_{i,p} \rangle) \right|$$
$$\leq O(J) \cdot \max_{k,j} \sigma(\langle \mathbf{w}_{m,j}^{(t)}, \mathbf{c}_k \rangle) + O(J) \cdot \max_{i,j,p} |\sigma(\langle \mathbf{w}_{m,j}^{(t)}, \boldsymbol{\xi}_{i,p} \rangle)|$$
$$= \widetilde{O}(\sigma_0^3),$$

where the first inequality is by triangle inequality and the last equality is by Lemma E.12. $\square$

Next we will show that router only focus on the cluster-center signal rather than the label signal during the router training.

**Lemma E.14.** Suppose (E.8), (E.9), (E.10) hold for all $t \in [T_1, T] \subseteq [T_1, T_2 - 1]$, then we have that $\|\mathbf{h}(\bar{\mathbf{x}}_i; \boldsymbol{\Theta}^{(t)}) - \mathbf{h}(\mathbf{x}_i; \boldsymbol{\Theta}^{(t)})\|_\infty = \widetilde{O}(d^{-0.005})$ hold for all $i \in [n]$ and $t \in [T_1, T + 1]$. Besides, we have that $\max_{m,k} |\langle \boldsymbol{\theta}_m^{(t)}, \mathbf{v}_k \rangle|, \max_{m,i,p} |\langle \boldsymbol{\theta}_m^{(t)}, \boldsymbol{\xi}_{i,p} \rangle| = \widetilde{O}(d^{-0.005})$ for all $t \in [T_1, T + 1]$.

*Proof.* Recall the definition of $\delta_{\boldsymbol{\Theta}}$ in Lemma E.11, we need to show that $\delta_{\boldsymbol{\Theta}^{(t)}} = \widetilde{O}(d^{-0.005})$ for all $t \in [T_1, T + 1]$. We first prove following router parameter update rules,

$$\langle \nabla_{\boldsymbol{\theta}_m} \mathcal{L}^{(t)}, \mathbf{v}_k \rangle = O(\delta_{\boldsymbol{\Theta}^{(t)}} K^2) + \widetilde{O}(d^{-0.005}), \langle \nabla_{\boldsymbol{\theta}_m} \mathcal{L}^{(t)}, \boldsymbol{\xi}_{i,p} \rangle = \widetilde{O}(d^{-0.005}), \tag{E.13}$$

for all $T_1 \leq t \leq T$, $m \in [M]$, $k \in [K]$, $i \in [n]$ and $p \geq 4$.

Consider the inner product of the router gradient and the feature vector and we have

$$
\mathbb{E}[\langle \nabla_{\boldsymbol{\theta}_m} \mathcal{L}^{(t)}, \mathbf{v}_k \rangle]
$$

$$
= \underbrace{\frac{1}{n} \sum_{i \in \Omega_k} \mathbb{P}(m_{i,t} = m) \ell'_{i,t} y_i \pi_m(\mathbf{x}_i; \boldsymbol{\Theta}^{(t)}) f_m(\mathbf{x}_i; \mathbf{W}^{(t)}) y_i \alpha_i}_{I_1}
$$

$$
+ \underbrace{\frac{1}{n} \sum_{i \in \Omega_{k',k}} \mathbb{P}(m_{i,t} = m) \ell'_{i,t} y_i \pi_m(\mathbf{x}_i; \boldsymbol{\Theta}^{(t)}) f_m(\mathbf{x}_i; \mathbf{W}^{(t)}) \epsilon_i \gamma_i}_{I_2}
$$

$$
- \underbrace{\frac{1}{n} \sum_{i \in \Omega_k, m' \in [M]} \mathbb{P}(m_{i,t} = m') \ell'_{i,t} y_i \pi_{m'}(\mathbf{x}_i; \boldsymbol{\Theta}^{(t)}) \pi_m(\mathbf{x}_i; \boldsymbol{\Theta}^{(t)}) f_{m'}(\mathbf{x}_i, \mathbf{W}^{(t)}) y_i \alpha_i}_{I_3}
$$

$$
- \underbrace{\frac{1}{n} \sum_{i \in \Omega_{k',k}, m' \in [M]} \mathbb{P}(m_{i,t} = m') \ell'_{i,t} \pi_{m'}(\mathbf{x}_i; \boldsymbol{\Theta}^{(t)}) y_i \pi_m(\mathbf{x}_i; \boldsymbol{\Theta}^{(t)}) f_{m'}(\mathbf{x}_i, \mathbf{W}^{(t)}) \epsilon_i \gamma_i}_{I_4}
$$

$$
+ \underbrace{\frac{1}{n} \sum_{i \in [n], p \geq 4} \mathbb{P}(m_{i,t} = m) \ell'_{i,t} y_i \pi_m(\mathbf{x}_i; \boldsymbol{\Theta}^{(t)}) f_m(\mathbf{x}_i; \mathbf{W}^{(t)}) \langle \mathbf{x}_i^{(p)}, \mathbf{v}_k \rangle}_{I_5}
$$

$$
- \underbrace{\frac{1}{n} \sum_{i \in [n], p \geq 4, m' \in [M]} \mathbb{P}(m_{i,t} = m') \ell'_{i,t} y_i \pi_{m'}(\mathbf{x}_i; \boldsymbol{\Theta}^{(t)}) \pi_m(\mathbf{x}_i; \boldsymbol{\Theta}^{(t)}) f_{m'}(\mathbf{x}_i; \mathbf{W}^{(t)}) \langle \mathbf{x}_i^{(p)}, \mathbf{v}_k \rangle}_{I_6} .
$$

$$
(E.14)
$$

Denote $y_i \pi_m(\bar{\mathbf{x}}_i; \boldsymbol{\Theta}^{(t)}) f_m(\widehat{\mathbf{x}}_i; \mathbf{W}^{(t)}), \forall i \in \Omega_{k,k'}^+$ by $\bar{F}_{k,k'}^+$. We next show that the output of the MoE multiplied by label: $y_i \pi_m(\mathbf{x}_i; \boldsymbol{\Theta}^{(t)}) f_m(\mathbf{x}_i; \mathbf{W}), \forall i \in \Omega_{k,k'}^+$ can be approximated by $\bar{F}_{k,k'}^+$.

$$
|\pi_m(\mathbf{x}_i; \boldsymbol{\Theta}^{(t)}) f_m(\mathbf{x}_i; \mathbf{W}^{(t)}) - \pi_m(\bar{\mathbf{x}}_i; \boldsymbol{\Theta}^{(t)}) f_m(\widehat{\mathbf{x}}_i; \mathbf{W}^{(t)})|
$$

$$
\leq |[\pi_m(\mathbf{x}_i; \boldsymbol{\Theta}^{(t)}) - \pi_m(\bar{\mathbf{x}}_i; \boldsymbol{\Theta}^{(t)})] f_m(\mathbf{x}_i; \mathbf{W}^{(t)})| + |\pi_m(\bar{\mathbf{x}}_i; \boldsymbol{\Theta}^{(t)})[f_m(\mathbf{x}_i; \mathbf{W}^{(t)}) - f_m(\widehat{\mathbf{x}}_i; \mathbf{W}^{(t)})]|
$$

$$
\leq O(|\pi_m(\mathbf{x}_i; \boldsymbol{\Theta}^{(t)}) - \pi_m(\bar{\mathbf{x}}_i; \boldsymbol{\Theta}^{(t)})|) + |f_m(\mathbf{x}_i; \mathbf{W}^{(t)}) - f_m(\widehat{\mathbf{x}}_i; \mathbf{W}^{(t)})|
$$

$$
\leq O(\delta_{\boldsymbol{\Theta}^{(t)}}) + \widetilde{O}(\sigma_0^3),
$$

where the first inequality is by triangle inequality, the second inequality is by $\pi_m(\bar{\mathbf{x}}_i; \boldsymbol{\Theta}^{(t)}) \leq 1$ and $|f_m(\mathbf{x}_i; \mathbf{W}^{(t)})| = O(1)$ in Lemma E.10, the third inequality is by (E.11) and Lemma E.13.

Similarly, denote $y_i \pi_m(\bar{\mathbf{x}}_i; \boldsymbol{\Theta}^{(t)}) f_m(\widehat{\mathbf{x}}_i; \mathbf{W}^{(t)}), i \in \Omega_{k,k'}^-$ by $\bar{F}_{k,k'}^-$ and we can show that value $y_i \pi_m(\mathbf{x}_i; \boldsymbol{\Theta}^{(t)}) f_m(\mathbf{x}_i; \mathbf{W}^{(t)}), \forall i \in \Omega_{k,k'}^-$ can be approximated by $\bar{F}_{k,k'}^-$. Now we can bound $I_1$ as

follows,

$$I_1 = \sum_{k'\neq k} \frac{\ell'(\bar{F}_{k,k'+})\bar{F}_{k,k'}^+}{n} \sum_{i\in\Omega_{k,k'}^+} \left[\mathbb{P}(m_{i,t}=m)y_i\alpha_i + O(\delta_{\boldsymbol{\Theta}^{(t)}})\right] + \widetilde{O}(\sigma_0^3)$$

$$+ \sum_{k'\neq k} \frac{\ell'(\bar{F}_{k,k'-})\bar{F}_{k,k'}^-}{n} \sum_{i\in\Omega_{k,k'}^-} \left[\mathbb{P}(m_{i,t}=m)y_i\alpha_i + O(\delta_{\boldsymbol{\Theta}^{(t)}})\right] + \widetilde{O}(\sigma_0^3)$$

$$\overset{(i)}{=} \sum_{k'\neq k} \frac{\ell'(\bar{F}_{k,k'+})\bar{F}_{k,k'}^+}{n} \sum_{i\in\Omega_{k,k'}^+} \left[\mathbb{P}(\bar{m}_{i,t}=m)y_i\alpha_i + O(M^2\delta_{\boldsymbol{\Theta}^{(t)}})\right] + \widetilde{O}(\sigma_0^3)$$

$$+ \sum_{k'\neq k} \frac{\ell'(\bar{F}_{k,k'-})\bar{F}_{k,k'}^-}{n} \sum_{i\in\Omega_{k,k'}^-} \left[\mathbb{P}(\bar{m}_{i,t}=m)y_i\alpha_i + O(M^2\delta_{\boldsymbol{\Theta}^{(t)}})\right] + \widetilde{O}(\sigma_0^3)$$

$$\overset{(ii)}{=} O(M^2\delta_{\boldsymbol{\Theta}^{(t)}}) + \widetilde{O}(n^{-1/2} + \sigma_0^3)$$

$$= O(M^2\delta_{\boldsymbol{\Theta}^{(t)}}) + \widetilde{O}(d^{-0.005})$$

where (i) is due to (E.12) and (ii) is by $\sum_{i\in\Omega_{k,k'}^+} y_i\alpha = \widetilde{O}(\sqrt{n})$ and $\sum_{i\in\Omega_{k,k'}^-} y_i\alpha = \widetilde{O}(\sqrt{n})$ in Lemma D.1. Similarly we can prove that $I_2, I_3, I_4 = O(M^2\delta_{\boldsymbol{\Theta}^{(t)}}) + \widetilde{O}(d^{-0.005})$. Since $\langle \mathbf{x}_i^{(p)}, \mathbf{v}_i \rangle = \widetilde{O}(d^{-1/2}), \forall p \geq 4$, $\pi_m, \pi_{m_{i,t}} \leq 1$ and $f_{m_{i,t}} = O(1)$, we can upper bound $I_5, I_6$ by $\widetilde{O}(d^{-1/2})$. Plugging those bounds into the gradient computation (E.14) gives

$$\mathbb{E}[\langle \nabla_{\boldsymbol{\theta}_m}\mathcal{L}^{(t)}, \mathbf{v}_k \rangle] = O(M^2\delta_{\boldsymbol{\Theta}^{(t)}}) + \widetilde{O}(d^{-0.005}).$$

We finally consider the alignment between router gradient and noise

$$\langle \nabla_{\boldsymbol{\theta}_m}\mathcal{L}^{(t)}, \boldsymbol{\xi}_{i',p'} \rangle = \frac{1}{n} \sum_{i\in[n],p\geq 4} \mathbb{1}(m_{i,t}=m)\ell_{i,t}' y_i \pi_{m_{i,t}}(\mathbf{x}_i;\boldsymbol{\Theta}^{(t)})f_{m_{i,t}}(\mathbf{x}_i;\mathbf{W}^{(t)})\langle \mathbf{x}_i^{(p)}, \boldsymbol{\xi}_{i',p'} \rangle$$

$$- \frac{1}{n} \sum_{i\in[n],p\geq 4} \ell_{i,t}' y_i \pi_{m_{i,t}}(\mathbf{x}_i;\boldsymbol{\Theta}^{(t)})\pi_m(\mathbf{x}_i;\boldsymbol{\Theta}^{(t)})f_{m_{i,t}}(\mathbf{x}_i;\mathbf{W}^{(t)})\langle \mathbf{x}_i^{(p)}, \boldsymbol{\xi}_{i',p'} \rangle.$$

$$\overset{(i)}{=} \widetilde{O}\left(\frac{1}{n}\right) + \widetilde{O}(d^{-1/2})$$

$$\overset{(ii)}{=} \widetilde{O}(d^{-1/2}),$$

where the (i) is by considering the cases $(i',p') = \boldsymbol{\xi}_{i,p}$ and $\boldsymbol{\xi}_{i',p'} \neq \boldsymbol{\xi}_{i,p}$ respectively and (ii) is due to our choice of $n$. Now, we have completed the proof of (E.13).

Plugging the gradient estimation (E.13) in to the gradient update rule for the gating network (3.5) gives

$$\max_{m,k}|\langle \boldsymbol{\theta}_m^{(t+1)}, \mathbf{v}_k \rangle| \leq \max_{m,k}|\langle \boldsymbol{\theta}_m^{(t)}, \mathbf{v}_k \rangle| + O(\eta_r M^2 \delta_{\boldsymbol{\Theta}^{(t)}}) + \widetilde{O}(\eta_r d^{-0.005}) \tag{E.15}$$

$$\max_{m,i,p}|\langle \boldsymbol{\theta}_m^{(t+1)}, \boldsymbol{\xi}_{i,p} \rangle| \leq \max_{m,i,p}|\langle \boldsymbol{\theta}_m^{(t)}, \boldsymbol{\xi}_{i,p} \rangle| + \widetilde{O}(\eta_r d^{-0.005}) \tag{E.16}$$

Combining (E.15) and (E.16), we have that there exist $C_1 = O(M^2)$ and $C_2 = \widetilde{O}(d^{-0.005})$ such that $\delta_{\boldsymbol{\Theta}^{(t+1)}} \leq \delta_{\boldsymbol{\Theta}^{(t)}} + C_1\eta_r\delta_{\boldsymbol{\Theta}^{(t)}} + C_2\eta_r$. Therefore, we have that

$$\delta_{\boldsymbol{\Theta}^{(t+1)}} + C_1^{-1}C_2 \leq (1 + C_1\eta_r)[\delta_{\boldsymbol{\Theta}^{(t)}} + C_1^{-1}C_2]$$
$$\leq \exp(C_1\eta_r)[\delta_{\boldsymbol{\Theta}^{(t)}} + C_1^{-1}C_2],$$

where the last inequality is due to $\exp(z) \geq 1 + z$ for all $z \in \mathbb{R}$. Then we further have that

$$\delta_{\boldsymbol{\Theta}^{(t)}} \leq \exp(C_1\eta_r t)[\delta_{\boldsymbol{\Theta}^{(0)}} + C_1^{-1}C_2] \leq \exp(C_1\eta_r\eta^{-1}M^{-2})[\delta_{\boldsymbol{\Theta}^{(0)}} + C_1^{-1}C_2] = \widetilde{O}(d^{-0.005}),$$

where the last equality is by $\eta_r = \Theta(M^2)\eta$. $\qquad\qquad\square$

Define $\Delta_{\Theta} := \max_{k \in [K]} \max_{m,m' \in \mathcal{M}_k} \max_{(\mathbf{x}_i, y_i) \in \Omega_k} |h_m(\mathbf{x}_i; \Theta) - h_{m'}(\mathbf{x}_i; \Theta)|$, which measures the bias of the router towards different experts in the same $\mathcal{M}_k$. Following Lemma shows that the router will treats professional experts equally when $\Delta_{\theta}$ is small.

**Lemma E.15.** For all $t \geq 0$, we have that following inequality holds,

$$\max_{k \in [K]} \max_{m,m' \in \mathcal{M}_k} \max_{(\mathbf{x}_i, y_i) \in \Omega_k} |\pi_{m'}(\mathbf{x}_i; \Theta^{(t)}) - \pi_m(\mathbf{x}_i; \Theta^{(t)})| \leq 2\Delta_{\Theta^{(t)}},$$

$$\max_{k \in [K]} \max_{m,m' \in \mathcal{M}_k} \max_{(\mathbf{x}_i, y_i) \in \Omega_k} |\mathbb{P}(m_{i,t} = m) - \mathbb{P}(m_{i,t} = m')| = O(M^2)\Delta_{\Theta^{(t)}}.$$

*Proof.* By Lemma C.3, we directly have that

$$|\mathbb{P}(m_{i,t} = m) - \mathbb{P}(m_{i,t} = m')| \leq O(M^2)|h_m(\mathbf{x}_i; \Theta^{(t)}) - h_{m'}(\mathbf{x}_i; \Theta^{(t)})|.$$

Then, we prove that

$$|\pi_{m'}(\mathbf{x}_i; \Theta) - \pi_m(\mathbf{x}_i; \Theta)| \leq 2|h_m(\mathbf{x}_i; \Theta^{(t)}) - h_{m'}(\mathbf{x}_i; \Theta^{(t)})|. \tag{E.17}$$

When $|h_m(\mathbf{x}_i; \Theta^{(t)}) - h_{m'}(\mathbf{x}_i; \Theta^{(t)})| \geq 1$, it is obvious that (E.17) is true. When $|h_m(\mathbf{x}_i; \Theta^{(t)}) - h_{m'}(\mathbf{x}_i; \Theta^{(t)})| \leq 1$ we have that

$$
\begin{aligned}
|\pi_{m'}(\mathbf{x}_i; \Theta) - \pi_m(\mathbf{x}_i; \Theta)| &= \left| \frac{\exp(h_m(\mathbf{x}_i; \Theta^{(t)})) - \exp(h_{m'}(\mathbf{x}_i; \Theta^{(t)}))}{\sum_{m''} \exp\left(h_{m''}(\mathbf{x}_i; \Theta^{(t)})\right)} \right| \\
&= \left| \frac{\exp(h_{m'}(\mathbf{x}_i; \Theta^{(t)}))}{\sum_{m''} \exp\left(h_{m''}(\mathbf{x}_i; \Theta^{(t)})\right)} \right| \cdot |\exp(h_m(\mathbf{x}_i; \Theta^{(t)}) - h_{m'}(\mathbf{x}_i; \Theta^{(t)})) - 1| \\
&\leq 2|h_m(\mathbf{x}_i; \Theta^{(t)}) - h_{m'}(\mathbf{x}_i; \Theta^{(t)})|,
\end{aligned}
$$

which completes the proof of (E.17). $\square$

Notice that the gating network is initialized to be zero, so we have $\Delta_{\Theta} = 0$ at initialization. We can further show that $\Delta_{\Theta} = O(1/\text{poly}(d))$ during the training up to time $T = \widetilde{O}(\eta^{-1})$.

**Lemma E.16.** Suppose (E.8), (E.9), (E.10) hold for all $t \in [T_1, T] \subseteq [T_1, T_2 - 1]$, then we have that $\Delta_{\Theta^{(t)}} \leq \widetilde{O}(d^{-0.001})$ holds for all $t \in [T_1, T + 1]$.

*Proof.* One of the key observation is the similarity of the m-th and the $m'$-th expert in the same expert class $\mathcal{M}_k$. Lemma E.12 implies that $\max_{i \in \Omega_k} |f_m(\mathbf{x}_i, \mathbf{W}^{(t)}) - f_{m'}(\mathbf{x}_i, \mathbf{W}^{(t)})| = \widetilde{O}(\sigma_0^{0.1}) \leq \widetilde{O}(d^{-0.001})$.

Another key observe is that, we only need to focus on the $k - th$ cluster-center signal. Lemma E.14 implies that,

$$
\begin{aligned}
\Delta_{\Theta^{(t)}} &= \max_{k \in [K]} \max_{m,m' \in \mathcal{M}_k} \max_{(\mathbf{x}_i, y_i) \in \Omega_k} |h_m(\mathbf{x}_i; \Theta) - h_{m'}(\mathbf{x}_i; \Theta^{(t)})| \\
&\leq \max_{k \in [K]} \max_{m,m' \in \mathcal{M}_k} \max_{(\mathbf{x}_i, y_i) \in \Omega_k} |h_m(\bar{\mathbf{x}}_i; \Theta^{(t)}) - h_{m'}(\bar{\mathbf{x}}_i; \Theta^{(t)})| + 2\delta_{\Theta^{(t)}} \\
&= \max_{k \in [K]} \max_{m,m' \in \mathcal{M}_k} |\langle \boldsymbol{\theta}_m - \boldsymbol{\theta}_{m'}, \beta_i \mathbf{c}_k \rangle| + 2\delta_{\Theta^{(t)}} \\
&\leq C_2 \max_{k \in [K]} \max_{m,m' \in \mathcal{M}_k} |\langle \boldsymbol{\theta}_m - \boldsymbol{\theta}_{m'}, \mathbf{c}_k \rangle| + 2\delta_{\Theta^{(t)}},
\end{aligned}
$$

where the first inequality is by Lemma E.14 and the second inequality is by $\beta_i \leq C_2$. We now prove that following gradient difference is small

$$\langle \nabla_{\boldsymbol{\theta}_m} \mathcal{L}^{(t)} - \nabla_{\boldsymbol{\theta}_{m'}} \mathcal{L}^{(t)}, \mathbf{c}_k \rangle$$

$$\overset{(i)}{=} \frac{1}{n} \sum_{i \in [n]} \sum_{p \in [P]} \mathbb{P}(m_{i,t} = m) \ell'_{i,t} \pi_m(\mathbf{x}_i; \boldsymbol{\Theta}^{(t)}) y_i f_m(\mathbf{x}_i; \mathbf{W}^{(t)}) \langle \mathbf{x}_i^{(p)}, \mathbf{c}_k \rangle$$

$$- \frac{1}{n} \sum_{i \in [n]} \sum_{p \in [P]} \mathbb{P}(m_{i,t} = m') \ell'_{i,t} \pi_{m'}(\mathbf{x}_i; \boldsymbol{\Theta}^{(t)}) y_i f_{m'}(\mathbf{x}_i; \mathbf{W}^{(t)}) \langle \mathbf{x}_i^{(p)}, \mathbf{c}_k \rangle$$

$$+ \frac{1}{n} \sum_{i \in \Omega_k} \sum_{p \in [P]} \sum_{m'' \in [M]} [\pi_{m'}(\mathbf{x}_i; \boldsymbol{\Theta}^{(t)}) - \pi_m(\mathbf{x}_i; \boldsymbol{\Theta}^{(t)})] \mathbb{P}(m_{i,t} = m'') \ell'_{i,t} \pi_{m''}(\mathbf{x}_i; \boldsymbol{\Theta}^{(t)}) \cdot$$

$$y_i f_{m''}(\mathbf{x}_i, \mathbf{W}) \langle \mathbf{x}_i^{(p)}, \mathbf{c}_k \rangle + \widetilde{O}(d^{-0.001})$$

$$= O\Big(\frac{1}{n}\Big) \sum_{i \in \Omega_k} [\mathbb{P}(m_{i,t} = m') - \mathbb{P}(m_{i,t} = m)] |\ell'_{i,t} \pi_m(\mathbf{x}_i; \boldsymbol{\Theta}) \beta_i y_i f_m(\mathbf{x}_i; \mathbf{W}^{(t)})| + \widetilde{O}(d^{-0.001})$$

$$+ O(1) \max_{i \in \Omega_k} |\pi_{m'}(\mathbf{x}_i; \boldsymbol{\Theta}^{(t)}) - \pi_m(\mathbf{x}_i; \boldsymbol{\Theta}^{(t)})| + O(1) \max_{i \in \Omega_k} |f_m(\mathbf{x}_i, \mathbf{W}^{(t)}) - f_{m'}(\mathbf{x}_i, \mathbf{W}^{(t)})|$$

$$= O(1) |\mathbb{P}(m_{i,t} = m') - \mathbb{P}(m_{i,t} = m)| + O(1) \max_{i \in \Omega_k} |\pi_{m'}(\mathbf{x}_i; \boldsymbol{\Theta}^{(t)}) - \pi_m(\mathbf{x}_i; \boldsymbol{\Theta}^{(t)})|$$

$$+ O(1) \max_{i \in \Omega_k} |f_m(\mathbf{x}_i, \mathbf{W}^{(t)}) - f_{m'}(\mathbf{x}_i, \mathbf{W}^{(t)})| + \widetilde{O}(d^{-0.001})$$

$$\overset{(ii)}{=} O(M^2 \Delta_{\boldsymbol{\Theta}^{(t)}}) + \widetilde{O}(d^{-0.001}),$$

where the (i) is by Lemma E.2 and (ii) is by Lemma E.15. It further implies that $\Delta_{\boldsymbol{\Theta}^{(t+1)}} \leq O(\eta_r M^2) \Delta_{\boldsymbol{\Theta}^{(t)}} + \widetilde{O}(\eta_r d^{-0.001})$. Following previous proof of $\delta_{\boldsymbol{\Theta}}$, we have that $\Delta_{\boldsymbol{\Theta}^{(T+1)}} = \widetilde{O}(d^{-0.001})$. $\square$

Together with the key technique 1, we can infer that each expert $m \in \mathcal{M}_k$ will get nearly the same load as other experts in $\mathcal{M}_k$. Since $\Delta_{\boldsymbol{\Theta}}$ keeps increasing during the training, it cannot be bounded if we allow the total number of iterations goes to infinity in Algorithm 1. This is the reason that we require early stopping in Theorem 4.2, which we believe can be waived by adding load balancing loss (Eigen et al., 2013; Shazeer et al., 2017; Fedus et al., 2021), or advanced MoE layer structure such as BASE Layers (Lewis et al., 2021; Dua et al., 2021) and Hash Layers (Roller et al., 2021).

**Lemma E.17.** Suppose (E.8), (E.9), (E.10) hold for all $t \in [T_1, T] \subseteq [T_1, T_2 - 1]$, then for $m \notin \mathcal{M}_k$ and $t \in [T_1, T]$, if $\langle \boldsymbol{\theta}_m^{(t)}, \mathbf{c}_k \rangle \geq \max_{m'} \langle \boldsymbol{\theta}_{m'}^{(t)}, \mathbf{c}_k \rangle - 1$ we have that

$$\langle \nabla_{\boldsymbol{\theta}_m} \mathcal{L}^{(t)}, \mathbf{c}_k \rangle \geq \Omega\Big(\frac{\eta^3 t^3}{K M^3}\Big) + \widetilde{O}(d^{-0.005}).$$

*Proof.* The expectation of the inner product $\langle \nabla_{\boldsymbol{\theta}_m} \mathcal{L}^{(t)}, \mathbf{c}_k \rangle$ can be computed as follows,

$$\mathbb{E}[\langle \nabla_{\boldsymbol{\theta}_m} \mathcal{L}^{(t)}, \mathbf{c}_k \rangle] = \frac{1}{n} \sum_{i,p} \mathbb{P}(m_{i,t} = m) \ell'_{i,t} \pi_m(\mathbf{x}_i; \boldsymbol{\Theta}^{(t)}) y_i f_m(\mathbf{x}_i; \mathbf{W}^{(t)}) \langle \mathbf{x}_i^{(p)}, \mathbf{c}_k \rangle$$

$$- \frac{1}{n} \sum_{i,p,m'} \mathbb{P}(m_{i,t} = m') \ell'_{i,t} \pi_{m'}(\mathbf{x}_i; \boldsymbol{\Theta}^{(t)}) \pi_m(\mathbf{x}_i; \boldsymbol{\Theta}^{(t)}) y_i f_{m'}(\mathbf{x}_i, \mathbf{W}^{(t)}) \langle \mathbf{x}_i^{(p)}, \mathbf{c}_k \rangle$$

$$\overset{(i)}{=} \frac{1}{n} \sum_{i \in \Omega_k} \mathbb{P}(m_{i,t} = m) \ell'_{i,t} \pi_m(\mathbf{x}_i; \boldsymbol{\Theta}^{(t)}) \beta_i y_i f_m(\mathbf{x}_i; \mathbf{W}^{(t)}) + \widetilde{O}(d^{-0.005})$$

$$- \frac{1}{n} \sum_{i \in \Omega_k} \sum_{m' \in [M]} \mathbb{P}(m_{i,t} = m') \ell'_{i,t} \pi_{m'}(\mathbf{x}_i; \boldsymbol{\Theta}^{(t)}) \pi_m(\mathbf{x}_i; \boldsymbol{\Theta}^{(t)}) \beta_i y_i f_{m'}(\mathbf{x}_i, \mathbf{W}).$$

(E.18)

where (i) is due to $|\langle \boldsymbol{\xi}_{i,p}, \mathbf{c}_k \rangle| = \widetilde{O}(d^{-0.5})$.

We can rewrite the inner product (E.18) as follows,

$$
\mathbb{E}[\langle \nabla_{\boldsymbol{\theta}_m} \mathcal{L}^{(t)}, \mathbf{c}_k \rangle] = \frac{1}{n} \sum_{i \in \Omega_k} \mathbb{P}(m_{i,t} = m) \ell'_{i,t} \pi_m(\mathbf{x}_i; \boldsymbol{\Theta}^{(t)}) \beta_i y_i f_m(\mathbf{x}_i; \mathbf{W}^{(t)}) + \widetilde{O}(d^{-0.005})
$$

$$
- \frac{1}{n} \sum_{i \in \Omega_k} \sum_{m' \in [M]} \mathbb{P}(m_{i,t} = m') \ell'_{i,t} \pi_{m'}(\mathbf{x}_i; \boldsymbol{\Theta}^{(t)}) \pi_m(\mathbf{x}_i; \boldsymbol{\Theta}^{(t)}) \beta_i y_i f_{m'}(\mathbf{x}_i, \mathbf{W})
$$

$$
= \underbrace{\frac{1}{n} \sum_{i \in \Omega_k} \mathbb{P}(m_{i,t} = m) \ell'_{i,t} \pi_m(\mathbf{x}_i; \boldsymbol{\Theta}^{(t)}) y_i \beta_i f_m(\mathbf{x}_i; \mathbf{W}^{(t)}) + \widetilde{O}(d^{-0.005})}_{I_1}
$$

$$
\underbrace{- \frac{1}{n} \sum_{i \in \Omega_k, m' \in \mathcal{M}_k} \mathbb{P}(m_{i,t} = m') \ell'_{i,t} \pi_{m'}(\mathbf{x}_i; \boldsymbol{\Theta}^{(t)}) \pi_m(\mathbf{x}_i; \boldsymbol{\Theta}^{(t)}) \beta_i y_i f_{m'}(\mathbf{x}_i, \mathbf{W}^{(t)})}_{I_2}
$$

(E.19)

$$
\underbrace{- \frac{1}{n} \sum_{i \in \Omega_k, m' \notin \mathcal{M}_k} \mathbb{P}(m_{i,t} = m') \ell'_{i,t} \pi_{m'}(\mathbf{x}_i; \boldsymbol{\Theta}^{(t)}) \pi_m(\mathbf{x}_i; \boldsymbol{\Theta}^{(t)}) \beta_i y_i f_{m'}(\mathbf{x}_i, \mathbf{W}^{(t)})}_{I_3}.
$$

(E.20)

To calculate $I_1, I_2, I_3$, let's first lower bound $I_2$. We now consider the case that $m \notin \mathcal{M}_k, m' \in \mathcal{M}_k$. Because $\langle \boldsymbol{\theta}_m^{(t)}, \mathbf{c}_k \rangle \geq \max_{m'} \langle \boldsymbol{\theta}_m^{(t)}, \mathbf{c}_k \rangle - 1$, we can easily prove that $\pi_m(\mathbf{x}_i; \boldsymbol{\Theta}^{(t)}) = \Omega(1/M), \forall i \in \Omega_k$. Then we have that

$$
I_2 = -\frac{1}{n} \sum_{i \in \Omega_k, m' \in \mathcal{M}_k} \mathbb{P}(m_{i,t} = m') \ell'_{i,t} \pi_{m'}(\mathbf{x}_i; \boldsymbol{\Theta}^{(t)}) \pi_m(\mathbf{x}_i; \boldsymbol{\Theta}^{(t)}) \beta_i y_i f_{m'}(\mathbf{x}_i, \mathbf{W}^{(t)})
$$

$$
\geq \Omega\Big(\frac{\eta^3 t^3}{nM^3}\Big) \sum_{i \in \Omega_k, m' \in \mathcal{M}_k} \beta_i
$$

$$
\geq \Omega\Big(\frac{\eta^3 t^3}{KM^3}\Big),
$$

where the first inequality is by $\pi_{m'}(\mathbf{x}_i; \boldsymbol{\Theta}^{(t)}) = \Omega(1/M)$, $\mathbb{P}(m_{i,t} = m') \geq \Theta(1/M)$, $\forall i \in \Omega_{k_m^*}, m \in [M]$, $y_i f_{m'}(\mathbf{x}_i; \mathbf{W}^{(t)}) = \eta^3 t^3 (1 - O(\sigma_0^{0.1}))$ and $\ell' = -\Theta(1)$ for all $i \in \Omega_k, m' \in \mathcal{M}_k$ due to Proposition E.9 and Lemma E.12, and the last inequality is by $|\mathcal{M}_k| \geq 1$ in Lemma D.4 and $\sum_{i \in \Omega_k} \beta_i = \Omega(n/K)$ in Lemma D.1.

Then we consider the case that $m, m' \notin \mathcal{M}_k$. Applying Taylor expansion of $\ell'_{i,t} = 1/2 + O(J\eta^3 t^3)$ gives

$$
\frac{1}{n} \sum_{i \in \Omega_k} \mathbb{P}(m_{i,t} = m) \ell'_{i,t} \pi_m(\mathbf{x}_i; \boldsymbol{\Theta}^{(t)}) y_i \beta_i f_m(\mathbf{x}_i; \mathbf{W}^{(t)})
$$

$$
= \frac{1}{2n} \sum_{i \in \Omega_k} \mathbb{P}(m_{i,t} = m) \pi_m(\mathbf{x}_i; \boldsymbol{\Theta}^{(t)}) y_i \beta_i f_m(\mathbf{x}_i; \mathbf{W}^{(t)}) + O(J^2 \eta^6 t^6)
$$

$$
= \frac{1}{2n} \sum_{k'} \sum_{i \in \Omega_{k,k'}^+} \mathbb{P}(m_{i,t} = m) \pi_m(\mathbf{x}_i; \boldsymbol{\Theta}^{(t)}) y_i \beta_i f_m(\mathbf{x}_i; \mathbf{W}^{(t)}) + O(J^2 \eta^6 t^6)
$$

$$
+ \frac{1}{2n} \sum_{k'} \sum_{i \in \Omega_{k,k'}^-} \mathbb{P}(m_{i,t} = m) \pi_m(\mathbf{x}_i; \boldsymbol{\Theta}^{(t)}) y_i \beta_i f_m(\mathbf{x}_i; \mathbf{W}^{(t)})
$$

$$
= O(J^2 \eta^6 t^6) + \widetilde{O}(d^{-0.005}).
$$

(E.21)

where the last inequality is by the technique we have used before in Lemma E.16. By (E.21), we can get upper bound $|I_1|, |I_3|$ by $O(J^2 \eta^6 t^6) + \widetilde{O}(d^{-0.005})$.

Plugging the bound of $I_1, I_2, I_3$ into (E.20) gives,

$$\langle \nabla_{\boldsymbol{\theta}_m} \mathcal{L}^{(t)}, \mathbf{c}_k \rangle \geq \Omega\left(\frac{\eta^3 t^3}{KM^3}\right) + O(J^2 \eta^6 t^6) + \widetilde{O}(d^{-0.005})$$

$$\leq \Omega\left(\frac{\eta^3 t^3}{KM^3}\right) + \widetilde{O}(d^{-0.005}),$$

where the last inequality is by $t \leq T_2 = \lfloor \eta^{-1} M^{-2} \rfloor$. $\qquad \square$

Now we can claim that Proposition E.9 is true and we summarize the results as follow lemma.

**Lemma E.18.** For all $T_1 \leq t \leq T_2$, we have Proposition E.9 holds. Besides, we have that $\langle \boldsymbol{\theta}_m^{(T_2)}, \mathbf{c}_k \rangle \leq \max_{m' \in [M]} \langle \boldsymbol{\theta}_{m'}^{(T_2)}, \mathbf{c}_k \rangle - \Omega(K^{-1}M^{-9})$ for all $m \notin \mathcal{M}_k$..

*Proof.* We will first use induction to prove Proposition E.9. It is worth noting that proposition E.9 is true at the beginning of the second stage $t = T_1$. Suppose (E.8), (E.9), (E.10) hold for all $t \in [T_1, T] \subseteq [T_1, T_2 - 1]$, we next verify that they also hold for $t \in [T_1, T+1]$. Lemma E.13 shows that (E.8) holds for $t \in [T_1, T+1]$. Lemma E.14 further shows that (E.8) holds for $t \in [T_1, T+1]$. Therefore, we only need to verify whether (E.10) holds for $t \in [T_1, T+1]$. Therefore, for each pair $i \in \Omega_k$, $m \in \mathcal{M}_k$, we need to estimate the gap between expert $m$ and the expert with best performance $h_m(\mathbf{x}_i; \boldsymbol{\Theta}^{(t)}) - \max_{m'} h_{m'}(\mathbf{x}_i; \boldsymbol{\Theta}^{(t)})$. By Lemma E.17 and Lemma E.14, we can induce that $h_m(\mathbf{x}_i; \boldsymbol{\Theta}^{(t)})$ is small therefore cannot be the largest one. Thus $h_m(\mathbf{x}_i; \boldsymbol{\Theta}^{(t)}) - \max_{m'} h_{m'}(\mathbf{x}_i; \boldsymbol{\Theta}^{(t)}) = h_m(\mathbf{x}_i; \boldsymbol{\Theta}^{(t)}) - \max_{m'} h_{m'}(\mathbf{x}_i; \boldsymbol{\Theta}^{(t)}) \leq \Delta_{\boldsymbol{\Theta}^{(t)}} \leq \widetilde{O}(d^{-0.001})$. Therefore, by Lemma C.3 we have (E.10) holds. Now we have verified that (E.10) also holds for $t \in [T_1, T+1]$, which completes the induction for Lemma E.9.

Finally, we carefully characterize the value of $\langle \boldsymbol{\theta}_m^{(t)}, \mathbf{c}_k \rangle$, for $\eta_r \eta^{-1} = \Theta(M^2)$ and $m \notin \mathcal{M}_k$. If $\langle \boldsymbol{\theta}_m^{(t)}, \mathbf{c}_k \rangle \geq \max_{m'} \langle \boldsymbol{\theta}_{m'}^{(t)}, \mathbf{c}_k \rangle - 1$, by Lemma E.17 we have that

$$\langle \boldsymbol{\theta}_m^{(t+1)}, c_k \rangle \leq \langle \boldsymbol{\theta}_m^{(t)}, c_k \rangle - \Theta\left(\frac{\eta_r \eta^3 t^3}{KM^3}\right) + \widetilde{O}(\eta_r d^{-0.005}) \leq 0. \qquad (E.22)$$

If there exists $t \leq T_2 - 1$ such that $\langle \boldsymbol{\theta}_m^{(t+1)}, c_k \rangle \leq \max_{m'} \langle \boldsymbol{\theta}_{m'}^{(t)}, \mathbf{c}_k \rangle - 1$, clearly we have that $\langle \boldsymbol{\theta}_m^{(T_2)}, c_k \rangle \leq -\Omega(K^{-1}M^{-9})$ since $\langle \boldsymbol{\theta}_m^{(t)}, c_k \rangle$ will keep decreasing as long as $\langle \boldsymbol{\theta}_m^{(t+1)}, c_k \rangle \geq -1$ and our step size $\eta_r = \Theta(M^2)\eta$ is small enough. If $\langle \boldsymbol{\theta}_m^{(t+1)}, c_k \rangle \geq \max_{m'} \langle \boldsymbol{\theta}_{m'}^{(t)}, \mathbf{c}_k \rangle - 1$ holds for all $t \leq T_2 - 1$, take telescope sum of (E.22) from $t = 0$ to $t = T_2 - 1$ gives that

$$\langle \boldsymbol{\theta}_m^{(T_2)}, c_k \rangle \leq \langle \boldsymbol{\theta}_m^{(0)}, c_k \rangle - \sum_{s=0}^{T_2-1} \Theta\left(\frac{\eta_r \eta^3 s^3}{KM^3}\right) + \widetilde{O}(d^{-0.005})$$

$$\overset{(i)}{=} -\sum_{s=0}^{T_2-1} \Theta\left(\frac{\eta_r \eta^3 s^3}{KM^3}\right) + \widetilde{O}(d^{-0.005})$$

$$\overset{(ii)}{=} -\Theta\left(\frac{\eta_r \eta^3 T_2^4}{KM^3}\right) + \widetilde{O}(d^{-0.005})$$

$$\leq -\Omega(K^{-1}M^{-9}),$$

where the (i) is by $\boldsymbol{\theta}_m^{(0)} = 0$ and (ii) is by $\sum_{i=0}^{n-1} i^3 = n^2(n-1)^2/4$ and the last inequality is due to $T_2 = \lfloor \eta^{-1} M^{-2} \rfloor$ and $\eta_r = \Theta(M^2)\eta$. Now we have proved that $\langle \boldsymbol{\theta}_m^{(T_2)}, c_k \rangle \leq -\Omega(K^{-1}M^{-9})$ for all $m \notin \mathcal{M}_k$. Finally, by Lemma E.1 we have that

$$\max_{m' \in [M]} \langle \boldsymbol{\theta}_{m'}^{(T_2)}, \mathbf{c}_k \rangle \geq \frac{1}{m} \sum_{m' \in [M]} \langle \boldsymbol{\theta}_{m'}^{(T_2)}, \mathbf{c}_k \rangle = 0.$$

Therefore, we have that $\langle \boldsymbol{\theta}_m^{(T_2)}, \mathbf{c}_k \rangle \leq -\Omega(K^{-1}M^{-9}) \leq \max_{m' \in [M]} \langle \boldsymbol{\theta}_{m'}^{(T_2)}, \mathbf{c}_k \rangle - \Omega(K^{-1}M^{-9})$, which completes the proof.

$\qquad \square$

## E.3 Generalization Results

In this section, we will present the detailed proof of Lemma 5.2 and Theorem 4.2 based on analysis in the previous stages.

*Proof of Lemma 5.2.* We consider the $m$-th expert in the MoE layer, suppose that $m \in \mathcal{M}_k$. Then if we draw a new sample $(\mathbf{x}, y) \in \Omega_k$. Without loss of generality, we assume $\mathbf{x} = [\alpha y \mathbf{v}_k, \beta \mathbf{c}_k, \gamma \epsilon \mathbf{v}_{k'}, \boldsymbol{\xi}]$. By Lemma E.8, we have already get the bound for inner product between weights and feature signal, cluster-center signal and feature noise. However, we need to recalculate the bound of the inner product between weights and random noises because we have fresh random noises i.i.d drawn from $\mathcal{N}(0, (\sigma_p^2/d) \cdot I_d)$. Notice that we use normalized gradient descent for expert with step size $\eta$, so we have that

$$\|\mathbf{w}_{m,j}^{(T_1)} - \mathbf{w}_{m,j}^{(0)}\|_2 \leq \eta T_1 = O(\sigma_0^{0.5}).$$

Therefore, by triangle inequality we have that $\|\mathbf{w}_{m,j}^{(T_1)}\|_2 \leq \|\mathbf{w}_{m,j}^{(0)}\|_2 + O(\sigma_0^{0.5}) \leq \widetilde{O}(\sigma_0 \sqrt{d})$. Because the inner product $\langle \mathbf{w}_{m,j}^{(t)}, \boldsymbol{\xi}_p \rangle$ follows the distribution $\mathcal{N}(0, (\sigma_p^2/d) \cdot \|\mathbf{w}_{m,j}^{(T_1)}\|_2^2)$, we have that with probability at least $1 - 1/(dPMJ)$,

$$|\langle \mathbf{w}_{m,j}^{(T_1)}, \boldsymbol{\xi}_p \rangle| = O(\sigma_p d^{-1/2} \|\mathbf{w}_{m,j}^{(t)}\|_2 \log(dPMJ)) \leq \widetilde{O}(\sigma_0).$$

Applying Union bound for $m \in [M], j \in [J], p \geq 4$ gives that, with probability at least $1 - 1/d$,

$$|\langle \mathbf{w}_{m,j}^{(T_1)}, \boldsymbol{\xi}_p \rangle| = \widetilde{O}(\sigma_0), \forall m \in [M], j \in [J], p \geq 4. \tag{E.23}$$

Now under the event that (E.23) holds, we have that

$$
\begin{aligned}
y f_m(\mathbf{x}, \mathbf{W}^{(t)}) &= y \sum_{j \in [J]} \sum_{p \in [P]} \sigma(\langle \mathbf{w}_{m,j}, \mathbf{x}^{(p)} \rangle) \\
&= y \sigma(\langle \mathbf{w}_{m,j_m^*}, \alpha y \mathbf{v}_k \rangle) + y \sum_{(j,p) \neq (j_m^*, 1)} \sigma(\langle \mathbf{w}_{m,j}, \mathbf{x}^{(p)} \rangle) \\
&\geq C_1^3 (1 - \sigma_0^{0.1})^3 \sigma_0^{1.5} - \widetilde{O}(\sigma_0^3) \\
&\geq \Omega(\sigma_0^{1.5}),
\end{aligned}
$$

where the first inequality is due to (E.3). Because (E.23) holds holds with probability at least $1 - 1/d$, so we have prove that

$$\mathbb{P}_{(\mathbf{x}, y) \sim \mathcal{D}} \big( y f_m(\mathbf{x}; \mathbf{W}^{(T_1)}) \leq 0 \big| (\mathbf{x}, y) \in \Omega_k \big) \leq 1/d.$$

On the other hand, if we draw a new sample $(\mathbf{x}, y) \in \Omega_{k'}, k' \neq k$. Then we consider the special set $\Omega_{k',k}^- \subseteq \Omega_{k'}$ where feature noise is $\mathbf{v}_k$ and the sign of the feature noise $\epsilon$ is not equal to the label $y$. Without loss of generality, we assume it as $\mathbf{x} = [\alpha y \mathbf{v}_{k'}, \beta \mathbf{c}_{k'}, -\gamma y \mathbf{v}_k, \boldsymbol{\xi}]$. Then under the event that (E.23) holds, we have that

$$
\begin{aligned}
y f_m(\mathbf{x}, \mathbf{W}^{(t)}) &= y \sum_{j \in [J]} \sum_{p \in [P]} \sigma(\langle \mathbf{w}_{m,j}, \mathbf{x}^{(p)} \rangle) \\
&= y \sigma(\langle \mathbf{w}_{m,j_m^*}, -\gamma y \mathbf{v}_k \rangle) + y \sum_{(j,p) \neq (j_m^*, 3)} \sigma(\langle \mathbf{w}_{m,j}, \mathbf{x}^{(p)} \rangle) \\
&\leq -C_1^3 (1 - \sigma_0^{0.1})^3 \sigma_0^{1.5} + \widetilde{O}(\sigma_0^3) \\
&\leq -\Omega(\sigma_0^{1.5}),
\end{aligned}
$$

where the first inequality is due to (E.3). Because (E.23) holds holds with probability at least $1 - 1/d$, so we have prove that

$$\mathbb{P}_{(\mathbf{x}, y) \sim \mathcal{D}} \big( y f_m(\mathbf{x}; \mathbf{W}^{(T_1)}) \leq 0 \big| (\mathbf{x}, y) \in \Omega_{k',k}^- \big) \geq 1 - 1/d.$$

Then we further have that

$$\mathbb{P}_{(\mathbf{x},y)\sim\mathcal{D}}\big(yf_m(\mathbf{x};\mathbf{W}^{(T_1)}) \leq 0\big|(\mathbf{x},y) \in \Omega_{k'}\big)$$
$$\geq \mathbb{P}_{(\mathbf{x},y)\sim\mathcal{D}}\big(yf_m(\mathbf{x};\mathbf{W}^{(T_1)}) \leq 0\big|(\mathbf{x},y) \in \Omega_{k',k}^-\big) \cdot \mathbb{P}_{(\mathbf{x},y)\sim\mathcal{D}}\big((\mathbf{x},y) \in \Omega_{k',k}^-\big|(\mathbf{x},y) \in \Omega_{k'}\big)$$
$$\geq \Omega(1/K),$$

which completes the proof.

$\square$

*Proof of Theorem 4.2.* We will give the prove for $T = T_2$, i.e., at the end of the second stage.

**Test Error is small.** We first prove the following result for the experts. For all expert $m \in \mathcal{M}_k$, we have that

$$\mathbb{P}_{(\mathbf{x},y)\sim\mathcal{D}}\big(yf_m(\mathbf{x};\mathbf{W}^{(T)}) \leq 0\big|(\mathbf{x},y) \in \Omega_k\big) = o(1). \tag{E.24}$$

The proof of is similar to the proof of Lemma 5.2. We consider the $m$-th expert in the MoE layer, suppose that $m \in \mathcal{M}_k$. Then if we draw a new sample $(\mathbf{x},y) \in \Omega_k$. Without loss of generality, we assume $\mathbf{x} = [\alpha y\mathbf{v}_k, \beta\mathbf{c}_k, \gamma\epsilon\mathbf{v}_{k'}, \boldsymbol{\xi}]$. By Lemma E.8, we have already get the bound for inner product between weights and feature signal, cluster-center signal and feature noise. However, we need to recalculate the bound of the inner product between weights and random noises because we have fresh random noises i.i.d drawn from $\mathcal{N}(0, (\sigma_p^2/d) \cdot I_d)$. Notice that we use normalized gradient descent with step size $\eta$, so we have that

$$\|\mathbf{w}_{m,j}^{(T)} - \mathbf{w}_{m,j}^{(0)}\|_2 \leq \eta T = \widetilde{O}(1).$$

Therefore, by triangle inequality we have that $\|\mathbf{w}_{m,j}^{(T)}\|_2 \leq \|\mathbf{w}_{m,j}^{(0)}\|_2 + \widetilde{O}(1) \leq \widetilde{O}(\sigma_0\sqrt{d})$. Because the inner product $\langle\mathbf{w}_{m,j}^{(t)}, \boldsymbol{\xi}_p\rangle$ follows the distribution $\mathcal{N}(0, (\sigma_p^2/d) \cdot \|\mathbf{w}_{m,j}^{(T)}\|_2^2)$, with probability at least $1 - 1/(dPMJ)$ we have that ,

$$|\langle\mathbf{w}_{m,j}^{(T)}, \boldsymbol{\xi}_p\rangle| = O(\sigma_p d^{-1/2}\|\mathbf{w}_{m,j}^{(t)}\|_2 \log(dPMJ)) \leq \widetilde{O}(\sigma_0).$$

Applying Union bound for $m \in [M], j \in [J], p \geq 4$ gives that, with probability at least $1 - 1/d$,

$$|\langle\mathbf{w}_{m,j}^{(T)}, \boldsymbol{\xi}_p\rangle| = \widetilde{O}(\sigma_0), \forall m \in [M], j \in [J], p \geq 4. \tag{E.25}$$

Now, under the event that (E.25) holds, we have that

$$yf_m(\mathbf{x},\mathbf{W}^{(T)}) = y \sum_{j\in[J]} \sum_{p\in[P]} \sigma(\langle\mathbf{w}_{m,j}^{(T)}, \mathbf{x}^{(p)}\rangle)$$
$$= y\sigma(\langle\mathbf{w}_{m,j_m^*}^{(T)}, \alpha y\mathbf{v}_k\rangle) + y \sum_{(j,p)\neq(j_m^*,1)} \sigma(\langle\mathbf{w}_{m,j}^{(T)}, \mathbf{x}^{(p)}\rangle)$$
$$\geq C_1^3(1 - \sigma_0^{0.1})^3 M^{-4} - \widetilde{O}(\sigma_0^3)$$
$$= \widetilde{\Omega}(1),$$

where the first inequality is by Lemma E.12. Because (E.25) holds with probability at least $1 - 1/d$, so we have prove that

$$\mathbb{P}_{(\mathbf{x},y)\sim\mathcal{D}}\big(yf_m(\mathbf{x};\mathbf{W}^{(T)}) \leq 0\big|(\mathbf{x},y) \in \Omega_k\big) \leq 1/d.$$

We then prove that, with probability at least $1 - o(1)$, an example $\mathbf{x} \in \Omega_k$ will be routed to one of the experts in $\mathcal{M}_k$. For $\mathbf{x} = [\alpha y\mathbf{v}_k, \beta\mathbf{c}_k, \gamma\epsilon\mathbf{v}_{k'}, \boldsymbol{\xi}]$, we need to check that $h_m(\mathbf{x};\boldsymbol{\Theta}^{(T)}) < \max_{m'} h_{m'}(\mathbf{x};\boldsymbol{\Theta}^{(T)}), \forall m \notin \mathcal{M}_k$. By Lemma E.18, we know that $\langle\boldsymbol{\theta}_m^{(T)}, \mathbf{c}_k\rangle \leq \max_{m'}\langle\boldsymbol{\theta}_{m'}^{(T)}, \mathbf{c}_k\rangle - \Omega(K^{-1}M^{-9})$. Further by Lemma E.14, we have that $\max_{m,k}|\langle\boldsymbol{\theta}_m^{(T)}, \mathbf{v}_k\rangle| = O(d^{-0.001})$. Again to calculate test error, we need to give an upper bound $\langle\boldsymbol{\theta}_m^{(T)}, \boldsymbol{\xi}_p\rangle$, where $\boldsymbol{\xi}_p$ is a fresh noise drawn from $\mathcal{N}(0, (\sigma_p^2/d) \cdot I_d)$. We can upper bound the gradient of the gating network by

$$\|\nabla_{\boldsymbol{\theta}_m}\mathcal{L}^{(t)}\|_2 = \left\|\frac{1}{n}\sum_{i,p}\mathbb{1}(m_{i,t}=m)\ell'_{i,t}\pi_{m_{i,t}}(\mathbf{x}_i;\boldsymbol{\Theta}^{(t)})y_i f_{m_{i,t}}(\mathbf{x}_i;\mathbf{W}^{(t)})\mathbf{x}_i^{(p)}\right.$$

$$\left.-\frac{1}{n}\sum_{i,p}\ell'_{i,t}\pi_{m_{i,t}}(\mathbf{x}_i;\boldsymbol{\Theta}^{(t)})\pi_m(\mathbf{x}_i;\boldsymbol{\Theta}^{(t)})y_i f_{m_{i,t}}(\mathbf{x}_i;\mathbf{W}^{(t)})\mathbf{x}_i^{(p)}\right\|_2.$$

$$=\widetilde{O}(1),$$

where the last inequality is due to $|\ell'_{i,t}|\le 1$, $\pi_m,\pi_{m_{i,t}}\in[0,1]$ and $\|\mathbf{x}_i^{(p)}\|_2 = O(1)$. This further implies that

$$\|\boldsymbol{\theta}_m^{(T)}\|_2 = \|\boldsymbol{\theta}_m^{(T)}-\boldsymbol{\theta}_m^{(0)}\|_2 \le \widetilde{O}(t\eta_r) \le \widetilde{O}(\eta^{-1}\eta_r) = \widetilde{O}(1),$$

where the last inequality is by $\eta_r = \Theta(M^2)\eta$. Because the inner product $\langle\boldsymbol{\theta}_m^{(T)},\boldsymbol{\xi}_p\rangle$ follows the distribution $\mathcal{N}(0,(\sigma_p^2/d)\cdot\|\boldsymbol{\theta}_m^{(T)}\|_2^2)$, we have that with probability at least $1-1/(dPM)$,

$$|\langle\boldsymbol{\theta}_m^{(T)},\boldsymbol{\xi}_p\rangle| = O(\sigma_p d^{-1/2}\|\boldsymbol{\theta}_m^{(T)}\|_2 \log(dPM)) \le \widetilde{O}(d^{-1/2}).$$

Applying Union bound for $m\in[M], p\ge 4$ gives that, with probability at least $1-1/d$,

$$|\langle\boldsymbol{\theta}_m^{(T)},\boldsymbol{\xi}_p\rangle| = \widetilde{O}(d^{-1/2}), \forall m\in[M], p\ge 4. \tag{E.26}$$

Now, under the event that (E.26) holds, we have that

$$h_m(\mathbf{x};\boldsymbol{\Theta}^{(T)}) - \max_{m'}h_{m'}(\mathbf{x};\boldsymbol{\Theta}^{(T)})$$

$$\le \langle\boldsymbol{\theta}_m^{(T)},\mathbf{c}_k\rangle - \max_{m'}\langle\boldsymbol{\theta}_{m'}^{(T)},\mathbf{c}_k\rangle + 4\max_{m,k}|\langle\boldsymbol{\theta}_m^{(T)},\mathbf{v}_k\rangle| + 4P\max_{m,p}|\langle\boldsymbol{\theta}_m^{(T)},\boldsymbol{\xi}_p\rangle|$$

$$\le -\Omega(K^{-1}M^{-9}) + \widetilde{O}(d^{-0.001})$$

$$< 0.$$

Because (E.26) holds holds with probability at least $1-1/d$, so we have prove that with probability at least $1-1/d$, an example $\mathbf{x}\in\Omega_k$ will be routed to one of the experts in $\mathcal{M}_k$.

**Training Error is zero.** The prove for training error is much easier, because we no longer need to deal with the fresh noises and we no longer need to use high probability bound for those inner products with fresh noises. That's the reason we can get exactly zero training error. We first prove the following result for the experts. For all expert $m\in\mathcal{M}_k$, we have that

$$y_i f_m(\mathbf{x}_i;\mathbf{W}^{(T)}) \le 0, \forall i\in\Omega_k.$$

Without loss of generality, we assume that the feature patch appears in $\mathbf{x}_i^{(1)}$. By Lemma E.12, we have that for all $i\in\Omega_k$

$$y_i f_m(\mathbf{x}_i,\mathbf{W}^{(T)}) = y_i\sum_{j\in[J]}\sum_{p\in[P]}\sigma(\langle\mathbf{w}_{m,j}^{(T)},\mathbf{x}_i^{(p)}\rangle)$$

$$= y_i\sigma(\langle\mathbf{w}_{m,j_m^*}^{(T)},\alpha y_i\mathbf{v}_k\rangle) + y_i\sum_{(j,p)\ne(j_m^*,1)}\sigma(\langle\mathbf{w}_{m,j}^{(T)},\mathbf{x}^{(p)}\rangle)$$

$$\ge C_1^3(1-\sigma_0^{0.1})^3 M^{-4} - \widetilde{O}(\sigma_0^3)$$

$$> 0,$$

where the first inequality is Lemma E.12. We then prove that, and example $(\mathbf{x}_i,y_i)\in\Omega$ will be routed to one of the experts in $\mathcal{M}_k$. Suppose the $m$-th expert is not in $\mathcal{M}_k$. We only need to check the value of $h_m(\mathbf{x}_i;\boldsymbol{\Theta}^{(T)}) < \max_{m'}h_{m'}(\mathbf{x}_i;\boldsymbol{\Theta}^{(T)})$, which is straight forward by Lemma E.18 and Lemma E.14.

$\square$

# F   Auxiliary Lemmas

**Lemma F.1.** Let $\{a_m\}_{m=1}^M$ are the random variable i.i.d. drawn from $\mathcal{N}(0,1)$. Define the non-increasing sequence of $\{a_m\}_{m=1}^M$ as $a^{(1)} \geq \ldots \geq a^{(M)}$. Then we have that

$$\mathbb{P}(a^{(2)} \geq (1-G)a^{(1)}) \leq GM^2$$

*Proof.* Let $\Psi$ be the CDF of $\mathcal{N}(0,1)$ and let $\rho$ be the PDF of $\mathcal{N}(0,\sigma_0^2)$. Then we have that,

$$
\begin{aligned}
&\mathbb{P}(a^{(2)} \geq (1-G)a^{(1)}) \\
&= \int_{a^{(1)} \geq \ldots \geq a^{(M)}} \mathbb{1}(a^{(2)} \geq (1-G)a^{(1)})M!\Pi_m \rho(a^{(m)})d\mathbf{a} \\
&= \int_{a^{(1)} \geq a^{(2)}} \mathbb{1}(a^{(2)} \geq (1-G)a^{(1)})M(M-1)\rho(a^{(1)})\rho(a^{(2)})\Psi(a^{(2)})^{M-2}da^{(1)}da^{(2)} \\
&\leq \int_{a^{(1)} \geq a^{(2)}} \mathbb{1}(a^{(2)} \geq (1-G)a^{(1)})M(M-1)\rho(a^{(1)})\frac{1}{\sqrt{2\pi}}da^{(1)}da^{(2)} \\
&= \int_{a^{(1)} \geq 0} \frac{GM(M-1)}{\sqrt{2\pi}}a^{(1)}\rho(a^{(1)})da^{(1)} \\
&\leq GM^2.
\end{aligned}
$$

$\square$

For normalized gradient descent we have following lemma,

**Lemma F.2** (Lemma C.19 Allen-Zhu and Li 2020c). Let $\{x_t, y_t\}_{t=1,\ldots}$ be two positive sequences that satisfy

$$
\begin{aligned}
x_{t+1} &\geq x_t + \eta \cdot C_t x_t^2 \\
y_{t+1} &\leq y_t + S\eta \cdot C_t y_t^2,
\end{aligned}
$$

and $|x_{t+1} - x_t|^2 + |y_{t+1} - y_t|^2 \leq \eta^2$. Suppose $x_0, y_0 = o(1), x_0 \geq y_0 S(1+G)$,

$$\eta \leq \min\{\frac{G^2 x_0}{\log(A/x_0)}, \frac{G^2 y_0}{\log(1/G)}\}.$$

Then we have for all $A > x_0$, let $T_x$ be the first iteration such that $x_t \geq A$, then we have $y_{T_x} \leq O(y_0 G^{-1})$.

*Proof.* We only need to replace $O(\eta A^{q-1})$ in the proof of Lemma C.19 by $O(\eta)$, because we use normalized gradient descent, i.e, $C_t \mathbf{x}_t^2 \leq 1$. For completeness, we present the whole poof here.

for all $g = 0, 1, 2, \ldots,$, let $\mathcal{T}_g$ be the first iteration such that $x_t \geq (1+\delta)^g x_0$, let $b$ be the smallest integer such that $(1+\delta)^b x_0 \geq A$. For simplicity of notation, we replace $x_t$ with $A$ whenever $x_t \geq A$. Then by the definition of $\mathcal{T}_g$, we have that

$$\sum_{t \in [\mathcal{T}_g, \mathcal{T}_{g+1})} \eta C_t[(1+\delta)^g x_0]^2 \leq x_{\mathcal{T}_{g+1}} - x_{\mathcal{T}_g} \leq \delta(1+\delta)^g x_0 + O(\eta),$$

where the last inequality holds because we are using normalized gradient descent, i.e., $\max_t |x_{t+1} - x_t| \leq \eta$. This implies that

$$\sum_{t \in [\mathcal{T}_g, \mathcal{T}_{g+1})} \eta C_t \leq \frac{\delta}{(1+\delta)^g}\frac{1}{x_0} + \frac{O(\eta)}{x_0^2}.$$

Recall that $b$ is the smallest integer such that $(1+\delta)^b x_0 \geq A$, so we can calculate

$$\sum_{t \geq 0, x_t \leq A} \eta C_t \leq \left[\sum_{g=0}^{b-1} \frac{\delta}{(1+\delta)^g}\frac{1}{x_0}\right] + \frac{O(\eta)}{x_0^2}b = \frac{1+\delta}{x_0} + \frac{O(\eta)b}{x_0^2} \leq \frac{1+\delta}{x_0} + \frac{O(\eta)\log(A/x_0)}{x_0^2 \log(1+\delta)}$$

Let $T_x$ be the first iteration $t$ in which $x_t \geq A$. Then we have that

$$\sum_{t=0}^{T_x} \eta C_t \leq \frac{1+\delta}{x_0} + \frac{O(\eta)\log(A/x_0)}{\delta x_0^2}. \tag{F.1}$$

On the other hand, let $A' = G^{-1}y_0$ and b' be the smallest integer such that $(1+\delta)b'x_0 \geq A'$. For simplicity of notation, we replace $y_t$ with $A'$ when $y_t \geq A'$. Then let $\mathcal{T}'_g$ be the first iteration such that $y_t \geq (1+\delta)^g y_0$, then we have that

$$\sum_{t\in[\mathcal{T}'_g,\mathcal{T}'_{g+1})} \eta S C_t[(1+\delta)^{g+1}x_0]^{(q-1)} \geq y_{\mathcal{T}'_{g+1}} - y_{\mathcal{T}'_g} \geq \delta(1+\delta)^g y_0 - O(\eta).$$

Therefore, we have that

$$\sum_{t\in[\mathcal{T}'_g,\mathcal{T}'_{g+1})} S\eta C_t \geq \frac{\delta}{(1+\delta)^g(1+\delta)^2} \frac{1}{y_0} - \frac{O(\eta)}{y_0^2}.$$

Recall that $b'$ is the smallest integer such that $(1+\delta)^{b'}y_0 \geq A'$. wo we have that

$$\sum_{t\geq 0, x_t \leq A} \eta S C_t \geq \sum_{g=0}^{b'-2} \frac{\delta}{(1+\delta)^g(1+\delta)^2} \frac{1}{y_0} - \frac{O(\eta)b'}{y_0^2}$$

Let $T_y$ be the first iteration $t$ in which $y_t \geq A'$, so we can calculate

$$\sum_{t=0}^{T_y} \eta S C_t \geq \frac{1 - O(\delta+G)}{y_0} - \frac{O(\eta)\log(A'/y_0)}{y_0^2\delta}. \tag{F.2}$$

Compare (F.1) and (F.2). Choosing $\delta = G$ and $\eta \leq \min\{\frac{G^2 x_0}{\log(A/x_0)}, \frac{G^2 y_0}{\log(1/G)}\}$, together with $x_0 \geq y_0 S(1+G)$

$\square$