# OpenReview forum: "Towards Understanding the Mixture-of-Experts Layer in Deep Learning"
_NeurIPS.cc/2022/Conference — NeurIPS 2022 Accept_

### Official Review · Reviewer_nUCK · 2022-07-03

**Rating:** 7
**Confidence:** 3
**Soundness:** 3 good
**Presentation:** 3 good
**Contribution:** 3 good

**Summary:**

The presented paper aims to crack open the blackbox of sparsely activated mixture-of-expert models and tries to understand what leads to the improved performance on many neural network tasks. Through experiments on a toy task, CIFAR-10, as well as a new CIFAR-10-Rotate dataset, the key-findings include that performance improvements are highly connected to the intrinsic cluster structure of the data  and dependent on the non-linear activation function in the experts.

**Questions:**

see above

**Limitations:**

Limitations are appropriately addressed in the conclusion and future extensions are given.

**Strengths And Weaknesses:**

### Strengths

- Intuitive visualisations on a toy task to understand the behaviour when training sparsely activated models and additional experiments on a created dataset named CIFAR-10-Rotate with strong underlying cluster structure.
- Empirical evidence that MoE's are heavily dependent on the non-linear activation function to learn the target distribution backed by visualisations for the router dispatch entropy in Figure 3.
-  Great effort of putting theory behind some of the empirical observations found in the field. Theory papers like this are essential for effectively moving forward in the empirical-driven field of deep learning.

---

### Weaknesses:

### [Update]: Weaknesses have been addressed in the rebuttal and I've increased the score from 6 to 7.

- **(medium)**: Table 1 + 2 only compare Single with linear + nonlinear activation function to the MoE models and shows that the Single model isn't able to learn the problem at hand to the same extend as the MoE variants. Since effectively MoE models have more parameters to play with (albeit them being in parallel branches), I believe adding total + effective parameter counts and adding one more line where we extend the Single model to the total number of parameters in the MoE model would be beneficial. This would verify that MoE's aren't just finding winning lottery tickets in their parameter initialization which allows them to perform better.
- **(small)**: The paper does consider other non-linear activation functions in the Appendix but here the setting is different than in the main paper. I would like to see Table 6 for the settings in Table 1, i.e. with $\alpha \in(0.5,2), \beta \in(1,2), \gamma \in(0.5,3), \sigma_{n}=1$ and  $\alpha \in(0.5,2), \beta \in(1,2), \gamma \in(0.5,3), \sigma_{n}=2$.
- **(large)**: Comparing the results of CIFAR-10 and CIFAR-10-Rotate, I find it surprising that MoE's don't seem to benefit CIFAR-10 although it should also have intrinsic clusters (similar to CIFAR-10-Rotate) which different experts should be able to utilize. In a paper that tries to understand MoE's and states that the cluster structure is important, I was hoping for a better explanation (or at least a hypothesis) in l.340 -- l.345 that would help to assess when MoE's are beneficial and what properties to look for in the dataset. In its current state, I do not know how I would transfer some of the findings in the paper to other real-world datasets e.g. multilingual machine translation. Maybe adding additional datasets from other domains or other computer vision datasets would help to solve this problem.
- **(large)**: Continuing from the previous point, a nice analysis would be to visualise the latent embeddings on CIFAR-10 & CIFAR-10-Rotate similar to Figure 1 using t-SNE and see if we can distill any inherent characteristics about the clustering behaviour. My assumption would be that in CIFAR-10-Rotate the inter-cluster separation/distance is a lot higher than in regular CIFAR-10 which enables better expert learning. Thinking this further, maybe we could add additional regularization during the the training of MOE's on CIFAR-10 to enforce better separation and simplify the training for the different experts.


---

### Minor Improvement Suggestions:

- Related literature that seems relevant enough to cite:
    - "*BASE Layers: Simplifying Training of Large, Sparse Models*" (Lewis et al., ICML 2021)
    - "*Hash Layers For Large Sparse Models*" (Roller et al., NeurIPS 2021)
    - "*Tricks for Training Sparse Translation Models*" (Dua et al., arXiv 2021)
- Typos & Formatting:
    - Table 1 can be made more easy to read and parse by using the $\LaTeX$ `booktabs` package with proper spacing. Similarly, Table 2--6 would benefit from it as well.
    - The definition the empirical loss after l. 175 & dispatch Entropy after l. 316 should have an Equation number
    -  l. 121 Radamacher -> Rademacher
- l. 165 refer to the experiments with other activation functions in Table 6 in the appendix where you consider more commonly used choices such as $\mathrm{ReLU}$ / $\mathrm{GeLU}$.
- l. 193 cite Adam paper
- Generally refer more to the Appendix in the main paper. It is really hard to follow the reading flow without it.

---

Apart from the mentioned points, the paper is well written and presentation, structure, and general layout of thoughts are up to NeurIPS standard. Related work is mentioned and cited accordingly apart from the previously mentioned papers.

---

> ### Author Response · Authors · 2022-08-02
> **Response to Reviewer nUCK**
>
> We thank the reviewer for the valuable feedback and improvement suggestions. Minor issues and suggestions have been addressed accordingly in the revision. We would respectively address the major concerns below, and we are happy to discuss any follow-up questions.
>
> ***
>
> **Response to Weakness 1:** We agree that it is necessary to show that MoE is not performing better just because it has more parameters to play with. In Appendix G Table 7, we added a column indicating the number of filters (which corresponds to J, number of neurons, in Eq 3.1) for single models and MoEs, while the filter size is consistent for all models. In the original synthetic experiment, we let $J=128$ for all single models and $J=16$ for all experts in MoE. As we let the number of experts $M=8$, the total number of filters/neurons for MoE would be 128 and equivalent to single models. We additionally added synthetic experiments for a single model with much more filters ($J=512$). Consistent with our theorem, a single model still fails under our data distributions. When the noise level is high, a single nonlinear CNN with 512 filters suffers from overfitting and performs even worse.
>
> ***
>
> **Response to Weakness 2:**  We have added the results under setting 1 and 2 in Appendix G Table 8, where single models with other non-linear activation functions would also fail in these settings.
>
> ***
>
> **Response to Weakness 3:**  In any k-class classification problem, it is natural to think that the data with the same label can compose a cluster. However, such a clustering structure can not help solve the classification problem since learning the labeling function is the ultimate goal of the task. Therefore, the underlying clustering structure considered in the mixture of classification problems should be independent of the labeling function. Take the data distribution in Definition 3.1 as an example, which is a 2-class classification problem, the clusters are $\Omega_{k}, k \in [K]$ instead of $\set{(x,y)|y=1}$ and $\set{(x,y)|y=-1}$.  For the CIFAR-10-Rotate data set, since ten classes are independent of the label (Rotate/Not-Rotate), each class becomes an underlying cluster for the classification problem. But this is not true for the CIFAR-10 data set because class information is the ultimate goal we want to learn.
>
>
> Extending to the general NLP domain, the data structure discussed above could be found in tasks with multiple linguistic sources, multiple topics, or multiple backgrounds. In Appendix G.2, we provide a simple example of how MoE would work for multilingual tasks. We combine sentiment analysis datasets with linguistic sources of English, French, and Russian as a multilingual sentiment analysis dataset. The goal is then to determine whether the sentiment of a text is positive or negative. In Figure 8, we provide visualization of the pre-trained text embeddings via t-SNE, where we could observe that each different source of language forms an underlying cluster. Table 13 shows that MoEs could improve over single models on this simple multilingual sentiment analysis task. In Table 14, we report the final router dispatch of MoE to each expert concerning each language. It is distinguishable that MoE managed to dispatch data with regard to the language/cluster to the experts.
>
> Regarding that MoEs do not seem to benefit CIFAR-10, similar results can also be found in [1] Table 4, where Gated CNN (MoE) could not provide an improvement over a single CNN.
>
> [1] Kalyan, Sai Krishna. Designing mixture of deep experts. MS thesis. Universitat Politècnica de Catalunya, 2017.
>
> ***
>
>
> **Response to Weakness 4:** Thanks for the advice. Visualizing the latent embeddings would help better illustrate our concept and theory. In Appendix G Figure 9, we added the visualization of the latent embeddings on CIFAR-10 and CIFAR-10-Rotate. As discussed in the previous question, the underlying clustering structure should be independent of the labeling function. An equivalent hypothesis is that if we fix the label $y$, the clustering structure should still exist. To verify that, we fix the label $y$ when visualizing the data distributions. In CIFAR-10, if we fix $y$, no clustering structure would be observed. However, in CIFAR-10-Rotate, fixing $y$ still reveals clustering structures. And if we investigate further, we could observe that the clusters in CIFAR-10-Rotate roughly consist of the classes in CIFAR-10: for example, images of the frog class mostly fall into one cluster, and the images of the ship class mostly fall into the other.

---

> > ### Comment · Reviewer_nUCK · 2022-08-02
> > **Thanks for the rebuttal**
> >
> > Thanks for the additional effort, I've increased the score from 6 to 7 as I believe all my weaknesses have been addressed.
> >
> > Regarding the additional experiments for Weaknesses 3 & 4, I believe that expanding the findings here in the main paper would be very beneficial for readers. I do understand that space is very limited but I consider this to be very insightful so try and see if you can squeeze it in somehow.
> >
> > I believe a slight restructuring of the Appendix can also be helpful here. If you move Appendix F & G before the proofs i.e. at position A, more readers will actually go through them and benefit from the additional experiments.

---

> > > ### Author Response · Authors · 2022-08-02
> > > **Thank you!**
> > >
> > > Thank you for raising the score and for your further suggestion. Given the extra page in the final version, we will move some of the new findings to the main text. We will also move Appendices F and G to the front of the appendix in the final version.

---

### Official Review · Reviewer_By4k · 2022-07-10

**Rating:** 4
**Confidence:** 5
**Soundness:** 3 good
**Presentation:** 3 good
**Contribution:** 2 fair

**Summary:**

The paper analyzes several features of mixture of experts (MoE) and derive a few conclusions such as the superiority of MoE in terms of the classification accuracy and it good loading balance.

**Questions:**

The key limitation is the analysis is not deep enough. If the analysis is empirical, more experiments are needed. If the analysis is based on math, then concrete prove should be added. Most of the math shown in this paper is used to describe the design of the experiments and does not help me understand the working mechanisms of MoE. Then more analysis on larger model and datasets are also necessary.

**Ethics Review Area:**

["I don’t know"]

**Limitations:**

Yes

**Strengths And Weaknesses:**

Strength: The paper is well written and clear motivation. The results that show MoE is more effective on clustered data is interesting.

Weakness

The analysis about MoE is too superficial. In the experimental, the author shows MoE can beat single expert, which is actually a common sense. It is also not surprising that a non-linear MoE is better than a plain MoE. For most of MoE papers, they use the MLP as the expert, which is a non-linear module.

The experiments are too toy to reflect the real working mechanisms behind MoE. Nowadays, MoE is often deployed on ViT architectures that are trained with JFT 300M or ImageNet 22k data. It is really hard to say the analysis built on synthetic data and cifar 10 is still applicable to large models. For instance, in L34-35, the author posted a question about why experts are not collapsed into a single expert. From my view, it is due to the loading balance loss that is used during training, otherwise it will indeed convert to one expert. I hope the author should at least further clarify his/her point about this.

---

> ### Author Response · Authors · 2022-08-02
> **Response to Reviewer By4k (2/2)**
>
>
> **Q3.** For instance, in L34-35, the author posted a question about why experts are not collapsed into a single expert. From my view, it is due to the loading balance loss that is used during training, otherwise it will indeed convert to one expert. I hope the author should at least further clarify his/her point about this.
>
> **A3.** There is a huge misunderstanding. Load balancing loss is not the reason why experts are not collapsed into a single expert. The purpose of adding load balancing loss is to help the router dispatch the same amount of data to each expert, which can’t directly affect the behavior of each expert. Actually, even if all the experts receive the same amount of data, they can still collapse into a single expert by learning the same function i.e., $f_{1} = …= f_{k} = … = f_{M}$ where $f_{k}$ represents the $k-th$ expert. In this case, no matter how clever the router is, the MoE will behave like a single model. According to our key technique 2, the divergence of experts is based on their functionality and comes from random initialization and the nonlinearity of the experts.
>
> For linear MoE, even with load balancing loss, its experts would not diverge to learn different feature signals. In Appendix G Table 9, we provide the empirical results for linear MoE with load balancing loss under setting 1-4 and directly compare it to nonlinear MoE without load balancing loss. We note that load balancing loss tries to guarantee that all experts receive a similar amount of data, but it is not the key element for the success of MoE on the data distribution that we study.
>
> ***
>
>
> **Q4.** The key limitation is the analysis is not deep enough … If the analysis is based on math, then concrete prove should be added .. Then more analysis on larger model and datasets are also necessary.
>
>
> **A4.**  We want to highlight that our paper is based on deep theoretical analysis, and we have concrete and solid proof presented in the appendix from page 15 to page 37. In the main paper, we also provide a lot of experiments to support our theorem. More supporting empirical results can be found in appendix F and G.

---

> > ### Comment · Reviewer_By4k · 2022-08-03
> > **follow up question**
> >
> > Thanks for the author's detailed response. I have one question about loading balance loss. I agree with the author that the random initialization can somehow prevent the router to choose the same expert for all tokens. However, I still think the loading balance loss is more critical. Its purpose is of course to distribute tokens in a more balance manner and optimize the speed but I disagree that **it can’t directly affect the behavior of each expert**. I think if you use some initialization but use the balance loss, it can still prevent the experts collapse. But just using random initialization cannot prevent this issue. Actually, in most of the experiments about MoE I did, the balance loss is necessary without which MoE will be likely to be a single MLP. But my experiments are large-scale with at least 32 experts. In conclusion, I do not argue the finding given in this paper is incorrect but it is too toy and contradicts to my own MoE experiences.

---

> > > ### Author Response · Authors · 2022-08-03
> > > **Response to Follow-up Question**
> > >
> > > Thank you for your follow-up question. We’re glad that we have addressed your other questions.
> > >
> > > Now we understand where the disagreement might come from. We think this is because we’re talking about different notions of “Collapse”.
> > >
> > > **Your notion of “Collapse”**: When you say “MoE collapses to MLP”, you are referring to the case where the router will route most of the data into one expert, and only this expert will be fully trained, while the other experts will receive few or even no data, and they will not be sufficiently trained.  In fact, we discussed your notion of “Collapse” in Main Difficulty 3, and there we proposed to use normalized gradient descent for expert training to tackle the difficulty.
> > >
> > > **Our notion of “Collapse”**: When we say “Experts in MoE collapse”, we are referring to the case when all the experts will become similar to each other. In other words, these experts are not diverse enough. The router will still route different data to different experts, but due to the similar behavior of all the experts, there is no “real expert” for each data point. We discussed this notion of “Collapse” in Main Difficulty 2, and we use random initialization and nonlinear activation functions for experts to overcome this difficulty.
> > >
> > > Back to load balancing loss, it can indeed alleviate your notion of “Collapse”, while in our paper we found that normalized gradient descent on expert training can also help mitigate your notion of “Collapse”. However, our point is that load balancing loss is not sufficient to resolve our notion of “Collapse”. Instead, we show in our paper that random initialization and nonlinear activation functions are the key to resolve our notion of “Collapse”.
> > >
> > > Please let us know if you have any further questions. Thanks.

---

> > > ### Author Response · Authors · 2022-08-07
> > > **Response to Follow-up Question (Part 2)**
> > >
> > > We’d like to follow up with you to see if our previous reply has addressed your question.
> > >
> > > To further address your question, we have conducted two more experiments in the past two days, and added these additional experiment results to Appendix G. In Appendix G Table 12, we report the average test accuracy for nonlinear MoE by trying different combinations of load balancing loss and normalized gradient descent. The synthetic data is from setting 1 and we choose the number of experts to be $M=32$. We observe that using normalized gradient descent or load balancing loss (or both) can lead to successful learning of the data distribution. However, without using normalized gradient or load balancing loss will result in failure of learning the data distribution. This experiment indicates that both normalized gradient descent and load balancing loss can mitigate your notion of “Collapse” (our main difficulty 3 in Section  5).  This confirms the effectiveness of using normalized gradient descent to address your notion of “Collapse”.
> > >
> > > In addition, in Appendix G Table 10, we investigate nonlinear MoE with the exact **same** initialization for different experts and load balancing loss. The synthetic data is from setting 1 and we report the performances of MoEs with 8 experts as well as 32 experts. Recall that the data distribution we study cannot be learned by any single model: experts must diverge to learn different labeling functions. We observe that the performances of MoEs drop to the similar performance of single models. Meanwhile, as we report in Table 11, all the experts of the MoE are in use and each expert gets a similar amount of data. This indicates that load balancing loss successfully ensures that MoEs do not use only one of its experts. However, since all the experts have similar performance and fail to learn different labeling functions, MoE will still fail even with load balancing loss. In other words, using the same initialization for different experts and loading balancing loss cannot resolve our notation of “Collapse”. But according to our previous experiments, using random initialization for different nonlinear experts can resolve our notation of “Collapse” without using load balancing loss.
> > >
> > > Please let us know if you have any further questions. We will try our best to address them.

---

> ### Author Response · Authors · 2022-08-02
> **Response to Reviewer By4k (1/2)**
>
> We thank the reviewer for the feedback and comments. We address the concerns and questions respectively below and are happy to clarify any follow-up questions/confusions.
> ***
> **Q1.** The analysis about MoE is too superficial. In the experimental, the author shows MoE can beat single expert, which is actually a common sense. It is also not surprising that a non-linear MoE is better than a plain MoE. For most of MoE papers, they use the MLP as the expert, which is a non-linear module.
>
> **A1.** We respectfully disagree that our analysis is superficial. We guess the reviewer may think the two-layer neural networks-based expert is simple compared with more complex networks. However, even in this simplest possible neural network setting, there is no existing analysis that can tell when and how MoE works.  In this paper, we provide the first theoretical analysis toward formally understanding the mechanism of the MoE layer.
>
> Regarding the performances of MoEs and single models, we point out that MoE does not unconditionally improve the performance of a single expert on any dataset. In fact, the improvement of MoE over a single model varies with respect to different tasks. For example, as shown in [1] and [2], MoEs significantly improve the performances of single models only on some datasets while providing a minor or no improvement on other datasets. Notably, in Figure 1 of [1], MoE could be worse than the single model on Winograd-Style Tasks. Empirically, we also found that MoE does not outperform single models on datasets like CIFAR-10. Similar results can be found in other literature like [3] (see their Table 4), where Gated CNN (MoE) does not beat a single CNN on CIFAR10. As the empirical results indicated a possible dependency of MoE on the task, we make an attempt to theoretically understand when and how MoE works better.
>
> We agree that most of the MoE papers use nonlinear experts. That is the reason we study nonlinear MoE in our paper. The purpose of including the linear MoE in the experiment is not to show the superior performance of nonlinear MoE but to support our theory and demonstrate why nonlinear MoEs perform better than linear MoEs.
>
> [1] Du, Nan, et al. "Glam: Efficient scaling of language models with mixture-of-experts." International Conference on Machine Learning. PMLR, 2022.
>
> [2] Zuo, Simiao, et al. "MoEBERT: from BERT to Mixture-of-Experts via Importance-Guided Adaptation." arXiv preprint arXiv:2204.07675 (2022).
>
> [3] Kalyan, Sai Krishna. Designing mixture of deep experts. MS thesis. Universitat Politècnica de Catalunya, 2017.
>
> ***
>
>
> **Q2.** The experiments are too toy to reflect the real working mechanisms behind MoE … It is really hard to say the analysis built on synthetic data and cifar 10 is still applicable to large models.
>
> **A2.** In this work, we do not focus on reproducing the success of the current works of MoE on large model architectures or datasets. Instead, we are interested in providing empirical support for our theorem and helping better understand the mechanism of MoE training that can be extended to real-world applications. We study CIFAR-10 and CIFAR-10-Rotate because they only differ in the clustering structure, which helps demonstrate how the MoE works on the mixture of the classification problem. Such mixture problems can be extended to tasks on large datasets, for example in the NLP domain, with different linguistic sources or topics (and also multi-task learning). In the revision (Appendix G.2), we include a simple example of the mixture of the classification problem in the NLP domain and how MoE would work in such a scenario.

---

### Official Review · Reviewer_RKSY · 2022-07-11

**Rating:** 6
**Confidence:** 3
**Soundness:** 4 excellent
**Presentation:** 3 good
**Contribution:** 3 good

**Summary:**

This paper engages in a formal study of a mixture-of-experts model (MOE) by considering a specific synthetic dataset with multiple clusters and a particular choice of the MOE model architecture. It outlines several theoretical and empirical results that show the importance of having multiple experts to properly fit the training data and the importance of nonlinearity in the architecture of an individual expert. It also mentions several typical problems that occur when training MOE models and discusses a set of techniques that address these issues. Finally, it extends empirical results to a more realistic rotated-CIFAR-10 dataset.


**Questions:**

1. While Theorem 4.1 requires the $\alpha$ and the $\gamma$ distributions to match, empirical results discussed in Section 6.1 were obtained in the $\mathcal{D}_\alpha \ne \mathcal{D}_\gamma$ case. Are there reasons to think that it would be interesting to confirm expected results for a single-expert model and more importantly verify that mixture-of-experts models perform sufficiently well in this case (where the distributions of the signal and the noise multipliers are effectively indistinguishable)?
2. It is not clear if the result of Theorem 4.2 actually requires the experts to be nonlinear functions, can this question be clarified? If Theorem 4.2 also holds for linear experts, then perhaps it should not be summarized as "Nonlinear MOE performs well"? Currently it is a little confusing to first see authors mention that both linear and nonlinear MOEs are expected to fit the training data, yet only see the result seemingly only holding for nonlinear functions.


**Limitations:**

I did not identify any potential negative societal impacts of this work. While also being true for a vast majority of other theoretical studies, the proposed framework is somewhat limited by a number of simplifying choices that make it difficult to extend most results beyond the studied scenario.


**Strengths And Weaknesses:**

### Strengths

1. This paper outlines a simple synthetic data distribution and a MOE model architecture that make it possible to conduct theoretical analysis of this problem and show a conceptual difference between a single- and multiple-expert scenarios. The proposed system is a simple model showing the essence of the advantage of the mixture-of-experts networks versus their simpler single-expert counterparts.
2. The empirical study goes beyond the simple synthetic setup and is extended to a rotated CIFAR-10 dataset showing the improvement provided by the mixture-of-experts model in this simple yet realistic case.
3. The paper outlines several practical techniques that altogether allow one to train mixture-of-experts models that are otherwise prone to various training issues.

### Weaknesses

1. The paper has multiple minor misprints and inaccuracies and may need another careful pass (numerous minor misprints, missing or unnecessary articles, missing parentheses in equations, for example in Lemma 5.2, etc.).
2. Several aspects of the theoretical analysis including the importance of expert nonlinearity for specific theoretical results may currently confuse the reader.
3. The problem setup is somewhat simplistic and cannot be easily extended to more realistic setups. But this limitation can of course be difficult to avoid if one wants to obtain meaningful theoretical predictions.

---

> ### Author Response · Authors · 2022-08-02
> **Response to Reviewer RKSY**
>
> We thank the reviewer for the insightful feedback and helpful suggestions. We have fixed the misprints and inaccuracies and are happy to answer additional follow-up questions.
> ***
> **Answer to Question 1:** Thanks for raising this point. We agree that a comprehensive empirical analysis is essential and can help better support our theoretical results. We have updated the empirical results and added settings 3&4 to confirm our expected results when $\alpha$ and $\gamma$ follow the same distribution. Please see our updated results in Appendix G Table 7. Under settings 3 and 4, the empirical results continue to confirm our finding that MoE significantly outperforms single models on such data distributions and that nonlinear MoE successfully uncovers the underlying cluster structure while linear MoE fails to do so.
>
> ***
>
> **Question 2 (Part 1):** It is not clear if the result of Theorem 4.2 actually requires the experts to be nonlinear functions … perhaps it should not be summarized as "Nonlinear MOE performs well"?
>
> **Answer to Question 2 (Part 1)：** As we introduced before Theorem 4.2, the theorem only works for the nonlinear experts that we defined in Line 158-165 and can’t work for linear experts. Our experiment also indicates that linear MoE trained by gradient descent does not perform very well. As we showed in Table 1, the linear MoE can only achieve approximately 90% accuracy compared to 99% accuracy achieved by nonlinear MoE.
>
> Technically, Theorem 4.2 requires the experts to be nonlinear functions because experts with linear activation functions can’t diverge. In particular,  Lemma 5.2, the key technical lemma for the proof of Theorem 4.2, will no longer hold for linear experts. In the revision, we have made it more clear by emphasizing the conditions in the statement of the theorem.
> ***
> **Question 2 (Part 2):** Currently, it is a little confusing to first see authors mention that both linear and nonlinear MOEs are expected to fit the training data, yet only see the result seemingly only holding for nonlinear functions.
>
> **Answer to Question 2 (Part 2):**  We believe there is a misunderstanding here. Although linear MoE can fit the training data from the function approximation point of view, there is no guarantee that we can find such an MoE efficiently. In particular, let $\mathcal{F} = \{f(W)\}$ denote the function class of all the linear MoE parameterized by weight $W$. Though there exists a linear MoE $f(W^*)$ that performs well, finding such a $W^*$ is very hard since it’s a nonconvex optimization problem. As shown in Table 1 and Figure 1, the linear expert can’t be trained efficiently by gradient descent. It may be possible to find $W^*$ for linear MoE via some sophisticated designed algorithm. But designing such an algorithm is beyond this paper’s scope since the focus of our paper is to understand the MoE trained by a practical optimization algorithm (gradient descent).

---

> > ### Comment · Reviewer_RKSY · 2022-08-08
> > **Thank you!**
> >
> > I would like to thank the authors for their detailed reply! The answers clarified my understanding of the paper and its main results.

---

### Official Review · Reviewer_TAGw · 2022-07-17

**Rating:** 6
**Confidence:** 3
**Soundness:** 3 good
**Presentation:** 3 good
**Contribution:** 3 good

**Summary:**

The authors provide theoretical and empirical study on the mixture of expert layer, which has recently become increasingly used and for which there is limited theoretical understanding. They show that cluster structure in the data is important for success of MoE, that under specific data distribution with cluster structure a single expert 2 layer CNN cannot generalize while a linear MoE can. They also show the advantages of non-linear MoE.

**Questions:**

none

**Ethics Review Area:**

["I don’t know"]

**Strengths And Weaknesses:**


*Strengths*

- Analysis on an important class of model/network components that are not well understood
- Theoretical analysis appears sound
- The writing is clear

*Weakness*
- The distribution studied is limited, i would like to see a more in depth discussion regarding why the distribution (with K centroids) is representative of real data. Indeed the experiments hint naturally many data does not have this underlying structure (e.g. CIFAR-10 performance is not improved by using MoE)
- (minor) There is some slight disconnect between the original question posed "Why do the experts in MoE diversify instead of collapsing into a single model? And how can the router learn to dispatch the data to the right expert?" and the work done, would be goo to make this flow smoother.

---

> ### Author Response · Authors · 2022-08-02
> **Response to Reviewer TAGw**
>
> We thank the reviewer for the feedback and valuable suggestions and are happy to discuss any follow-up questions.
> ***
>
> **Q1.** I would like to see a more in-depth discussion regarding why the distribution (with K centroids) is representative of real data. Indeed the experiments hint naturally many data does not have this underlying structure (e.g. CIFAR-10 performance is not improved by using MoE)
>
> **A1.**   Our experiment on CIFAR-10 reveals that MoE cannot unconditionally improve the performance of a single expert. Such results can also be found in Figure 1 of [1], where the MoE gives a minor improvement in some categories of the dataset. Since this paper aims to theoretically understand the mechanism of MoE training, we need to analyze a data distribution that MoE can make a significant improvement. We agree that the clustering structure can not represent all the real datasets, but it would be one of the most representative datasets that MoE can make a great improvement.
>
> The mixture of the classification problems with clustering structure has been widely applied in traditional machine learning, including the tasks we summarized in Lines 88 to 92. Furthermore, such a mixture of classification problems with clustering structure can also be found in the natural language processing domain. The clustering structure may come from multiple background information, multiple types of topics, and multiple linguistic sources.
>
> In Appendix G.2, we provide an example of multilingual sentiment analysis from different linguistic sources of English, French, and Russian. The visualization (Figure 8) of the pre-trained text embeddings via t-SNE indicates that each source of language forms a cluster. Table 13 shows that MoEs could improve over single models on this simple multilingual sentiment analysis task. We further report the final router dispatch to each expert in Table 14, which shows that the router manages to dispatch data with respect to the underlying clustering structures.
>
> [1] Du, Nan, et al. "Glam: Efficient scaling of language models with mixture-of-experts." International Conference on Machine Learning. PMLR, 2022.
>
>
>
>
>
>
>
>
>
>
>
> ***
>
>
>
> **Q2.** There is some slight disconnect between the original question … and the work done, would be good to make this flow smoother.
>
> **A2.**  Thank you for the valuable suggestion. We answer the first question in Section 5, key technique 2. During the early stage of training (exploration stage), the expert will diverge into several groups that specialize in different clusters $[M] = \sqcup_{k} \mathcal{M}_{k}$. The diverge process will only depend on the initialization of the expert weights and benefit from the nonlinearity of the activation function.
>
> We answer the second question in Section 5 via all the key techniques. The router will wait for experts to diverge in the exploration stage and keep the load balance of different experts. After that, the router will learn to dispatch the data to the right expert by learning the cluster-center signal based on the diversity of the experts (See Lemma D.18 for details).
>
> We will make the flow smoother and highlight the answers in the revision.

---

### Author Response · Authors · 2022-08-02
**Response to All Reviewers**

We want to thank all the reviewers for their valuable comments. We are glad that reviewers recognize the soundness of our proofs (Rev.TAGw) and that our theoretical contributions are “essential for effectively moving forward in the empirical-driven field of deep learning” (Rev. nUCK). We are also thankful to Rev.RKSY for acknowledging that simple synthetic data distribution assumption would be difficult to avoid for a theory paper like ours.

We will reply to the reviewers separately and address all the raised points. A rebuttal revision of our paper with the supplementary material has been uploaded, and the main changes in the body will be highlighted in blue. In the revision, we have addressed the improvement suggestion (Rev. nUCK) and fixed several minor misprints and inaccuracies (Rev. RKSY). Due to the strict 9-page limit, we try to make minimum changes to the main body of the paper. In Appendix G, we include additional empirical results that address reviewers’ concerns on both synthetic and real data experiments and provide a simple example of how linguistic data would fall within the scope of our theory.

---

### Meta-Review · Area_Chair_ceyt · 2022-08-26

**Recommendation:** Accept
**Confidence:** Certain

**Metareview:**

This paper analyzes mixtures of layers applied to DNNs, in order to understand their limits and how they benefit to DNNs. A significant theory and experiments support the claims of this paper, despite the doubts of one review on the experimental protocol. Consequently, I recommend the acceptance of this paper.

**Award:**

No

---

### Decision · Program_Chairs · 2022-09-14

Accept